# Maximum Likelihood Training of Implicit Nonlinear Diffusion Models

**Dongjun Kim**[*]
KAIST
dongjoun57@kaist.ac.kr

**Byeonghu Na**[*]
KAIST
wp03052@kaist.ac.kr

**Se Jung Kwon**
NAVER CLOVA

**Dongsoo Lee**
NAVER CLOVA

**Wanmo Kang**
KAIST

**Il-Chul Moon**
KAIST / Summary.AI

## Abstract

Whereas diverse variations of diffusion models exist, extending the linear diffusion into a nonlinear diffusion process is investigated by very few works. The nonlinearity effect has been hardly understood, but intuitively, there would be promising diffusion patterns to efficiently train the generative distribution towards the data distribution. This paper introduces a data-adaptive nonlinear diffusion process for score-based diffusion models. The proposed Implicit Nonlinear Diffusion Model (INDM) learns by combining a normalizing flow and a diffusion process. Specifically, INDM implicitly constructs a nonlinear diffusion on the *data space* by leveraging a linear diffusion on the *latent space* through a flow network. This flow network is key to forming a nonlinear diffusion, as the nonlinearity depends on the flow network. This flexible nonlinearity improves the learning curve of INDM to nearly Maximum Likelihood Estimation (MLE) against the non-MLE curve of DDPM++, which turns out to be an inflexible version of INDM with the flow fixed as an identity mapping. Also, the discretization of INDM shows the sampling robustness. In experiments, INDM achieves the state-of-the-art FID of 1.75 on CelebA. We release our code at https://github.com/byeonghu-na/INDM.

## 1 Introduction

Diffusion models have recently achieved success on the task of sample generation, and some works [1, 2] claim state-of-the-art performance over Generative Adversarial Networks (GAN) [3]. This success is highlighted particularly in likelihood-based models, including normalizing flows [4], autoregressive models [5], and Variational Auto-Encoders (VAE) [6]. Moreover, this success is noteworthy because it is achieved merely using linear diffusion processes, such as Variance Exploding (VE) Stochastic Differential Equation (SDE) [7], and Variance Preserving (VP) SDE [8].

This paper extends linear diffusions of VE/VP SDEs to a data-adaptive trainable nonlinear diffusion. To motivate the extension, though there are structural similarities between diffusion models and VAEs, the inference part of a linear diffusion process has not been trained while its counterpart of VAE (the encoder) has been trained. We introduce Implicit Nonlinear Diffusion Models (INDM) to train its *forward* SDE, the inference part in diffusion models. INDM constructs the nonlinearity of the data diffusion by transforming a linear *latent* diffusion back to the data space.

We implement the transformation between the data and latent spaces with a normalizing flow. The invertibility of the flow mapping is key to learning a nonlinear inference part. Invertibility is necessary

---

[*]Equal contribution

| SDE | $f(x_t,t)$ | $G(x_t,t)$ | $x_0$ | $x_{0.1}$ | $x_{0.2}$ | $x_{0.3}$ | $x_{0.4}$ | $x_{0.5}$ | $x_{0.6}$ | $x_{0.7}$ | $x_{0.8}$ | $x_{0.9}$ | $x_1$ |
|---|---|---|---|---|---|---|---|---|---|---|---|---|---|
| Linear (VE/VP) | Linear $f(x_t,t) \propto x_t$ | Linear $G(x_t,t) = g(t)$ | | | | | | | | | | | |
| Nonlinear | Nonlinear | Linear $G(x_t,t) = g(t)$ | | | | | | | | | | | |
| | Semi-linear (Rotating) | Nonlinear | | | | | | | | | | | |

Figure 1: Examples of linear (top row) and nonlinear (middle/bottom rows) diffusion processes.

for constructing the nonlinearity, and we clarify this by comparing INDM with LSGM [9], a latent diffusion model with VAE. Altogether, INDM provides the following advantages over the existing models.

- INDM extends the scope of diffusion models from linear SDEs to *implicit nonlinear* SDEs.
- INDM learns not only drift but *volatility* coefficients of the forward (inference) SDE.
- INDM trains its network with *Maximum Likelihood Estimation* (MLE).
- INDM is *robust* on the sampling discretization.

## 2 Preliminary

A diffusion model is constructed with bidirectional *forward* and *reverse* stochastic processes.

**Forward and Reverse Diffusions** A forward diffusion process diffuses an input data variable, $\mathbf{x}_0 \sim p_r$, to a noise variable, and the corresponding reverse diffusion process [10] of this forward diffusion denoises a noise variable to regenerate the input variable. The forward diffusion is fully described by an SDE of $\mathrm{d}\mathbf{x}_t = \mathbf{f}(\mathbf{x}_t,t)\,\mathrm{d}t + \mathbf{G}(\mathbf{x}_t,t)\,\mathrm{d}\mathbf{w}_t$, and the corresponding reverse SDE becomes $\mathrm{d}\mathbf{x}_t = \left[\mathbf{f}(\mathbf{x}_t,t) - \mathrm{div}(\mathbf{G}\mathbf{G}^T)(\mathbf{x}_t,t) - (\mathbf{G}\mathbf{G}^T)(\mathbf{x}_t,t)\nabla_{\mathbf{x}_t}\log p_t(\mathbf{x}_t)\right]\mathrm{d}\bar{t} + \mathbf{G}(\mathbf{x}_t,t)\,\mathrm{d}\bar{\mathbf{w}}_t$. Here, $\mathbf{w}_t \in \mathbb{R}^d$ is an abstraction of a random walk process with independent increments, where $d$ is the data dimension, and $\mathrm{d}\bar{\mathbf{w}}_t$ is the standard Wiener processes with backwards in time.

**Generative Diffusion** Having that the drift ($\mathbf{f} \in \mathbb{R}^d$) and the volatility ($\mathbf{G} \in \mathbb{R}^{d\times d}$) terms are given a-priori, diffusion models [1] estimate the data score, $\nabla_{\mathbf{x}_t}\log p_t(\mathbf{x}_t)$, with the score network, $\mathbf{s}_{\boldsymbol{\theta}}(\mathbf{x}_t,t)$. By plugging the score network in the data score, we obtain another diffusion process, called the *generative* SDE, described by $\mathrm{d}\mathbf{x}_t^{\boldsymbol{\theta}} = \left[\mathbf{f}(\mathbf{x}_t^{\boldsymbol{\theta}},t) - \mathrm{div}(\mathbf{G}\mathbf{G}^T)(\mathbf{x}_t^{\boldsymbol{\theta}},t) - (\mathbf{G}\mathbf{G}^T)(\mathbf{x}_t^{\boldsymbol{\theta}},t)\mathbf{s}_{\boldsymbol{\theta}}(\mathbf{x}_t^{\boldsymbol{\theta}},t)\right]\mathrm{d}\bar{t} + \mathbf{G}(\mathbf{x}_t^{\boldsymbol{\theta}},t)\,\mathrm{d}\bar{\mathbf{w}}_t$. Starting from a prior distribution of $\mathbf{x}_T^{\boldsymbol{\theta}} \sim \pi$ and solving the SDE time backwards, Song et al. [1] construct the generative stochastic process of $\{\mathbf{x}_t^{\boldsymbol{\theta}}\}_{t=0}^T$ that perfectly reconstructs the reverse process of $\{\mathbf{x}_t\}_{t=0}^T$ under two conditions: 1) $\mathbf{s}_{\boldsymbol{\theta}}(\mathbf{x}_t,t) = \nabla_{\mathbf{x}_t}\log p_t(\mathbf{x}_t)$ and 2) $\mathbf{x}_T \sim \pi$. We define a generative distribution, $p_{\boldsymbol{\theta}}$, as the distribution of $\mathbf{x}_0^{\boldsymbol{\theta}}$.

**Score Estimation** The diffusion model estimates the data score with the score network by minimizing the denoising score loss [1], given by $\mathcal{L}(\{\mathbf{x}_t\}_{t=0}^T, \lambda; \boldsymbol{\theta}) = \int_0^T \lambda(t)\mathbb{E}_{\mathbf{x}_0,\mathbf{x}_t}[\|\mathbf{s}_{\boldsymbol{\theta}}(\mathbf{x}_t,t) - \nabla_{\mathbf{x}_t}\log p_{0t}(\mathbf{x}_t|\mathbf{x}_0)\|_2^2]\,\mathrm{d}t$, where $p_{0t}(\mathbf{x}_t|\mathbf{x}_0)$ is a transition probability of $\mathbf{x}_t$ given $\mathbf{x}_0$; and $\lambda$ is the weighting function that determines the level of contribution for each diffusion time. When $\mathbf{G}(\mathbf{x}_t,t) = g(t)$, Song et al. [11], Huang et al. [12] proved that this loss with the likelihood weighting ($\lambda = g^2$) turns out to be a variational bound of the negative log-likelihood: $\mathbb{E}_{\mathbf{x}_0}[-\log p_{\boldsymbol{\theta}}(\mathbf{x}_0)] \leq \mathcal{L}(\{\mathbf{x}_t\}_{t=0}^T, g^2; \boldsymbol{\theta}) - \mathbb{E}_{\mathbf{x}_T}[\log \pi(\mathbf{x}_T)]$, up to a constant, see Appendix A.1 for a detailed discussion.

**Choice of Drift (f) and Volatility (G) Terms** The original diffusion model strictly limits the scope of diffusion process to be a family of linear diffusions that $\mathbf{f}$ is a linear function of $\mathbf{x}_t$ and $\mathbf{G}$ is an identity matrix multiplied by a $t$-function. For instance, VESDE [1, 7] satisfies $\mathbf{f} \equiv 0$ with $\mathbf{G} = \sqrt{\mathrm{d}\sigma^2(t)/\mathrm{d}t}\mathbf{I}$ and VPSDE [1, 8] satisfies $\mathbf{f} = -\frac{1}{2}\beta(t)\mathbf{x}_t \propto \mathbf{x}_t$ with $\mathbf{G} = \sqrt{\beta(t)}\mathbf{I}$. Few concurrent works have extended linear diffusions to nonlinear diffusions by 1) applyng a latent diffusion using VAE in LSGM [9], 2) applying a flow network to nonlinearize the drift term in DiffFlow [13], and 3) reformulating the diffusion model into a Schrodinger Bridge Problem (SBP) [14–16]. We further analyze these approaches in Section 5.

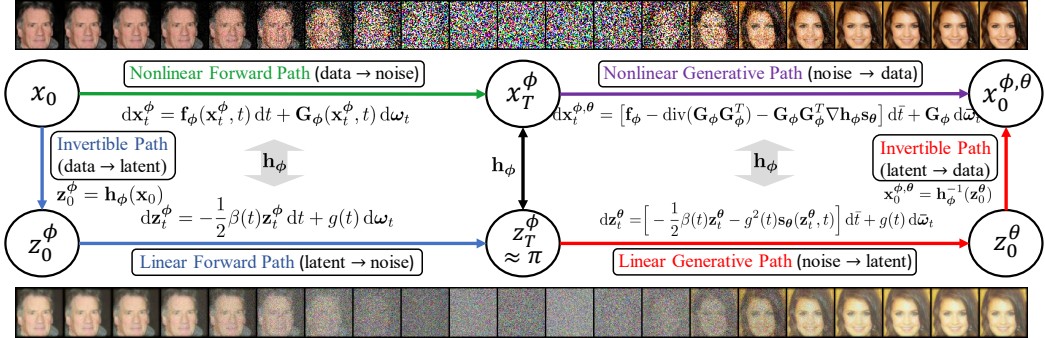

Figure 3: INDM attains a ladder structure between the data space and the latent space. The latent vector is visualized by normalizing the latent value, see Appendix F.5.2 for further visualization.

# 3   Motivation of Nonlinear Diffusion Process

Figure 1 illustrates various diffusion processes on a spiral toy dataset. In the top row, the diffusion path of VPSDE keeps its overall structure of the initial data manifold during the data deformation procedure to $\mathcal{N}(0, \mathbf{I})$. The drift vector field illustrated in Figure 2-(a) as black arrows presents that VPSDE *linearly* deforms its data distribution.

Unlike the linear diffusion, the middle row of Figure 1 with a nonlinear drift shows that the data is not linearly deformed to $\mathcal{N}(0, \mathbf{I})$. Figure 2-(b) illustrates the corresponding vector field, in which two distinctive components (orange/blue) are forced to separate each other. The nonlinearity of the drift term represented as rotating black arrows is the source of such nonlinear deformation at the intermediate steps,

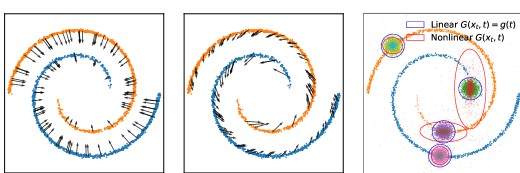

(a) Linear $\mathbf{f} \propto \mathbf{x}_t$  (b) Nonlinear $\mathbf{f}$  (c) Nonlinear $\mathbf{G}$

Figure 2: Vector fields on various SDEs at $t = 0$.

$\mathbf{x}_{0.2} \sim \mathbf{x}_{0.6}$. When it comes to the volatility term, the last row of Figure 1 presents the process with nonlinear $\mathbf{G}$. Figure 2-(c) illustrates the covariance matrices of the perturbation distribution at $t = 0$ with linear and nonlinear volatility terms, where the perturbation distribution induced by the volatility term is $\mathcal{N}(0, \mathbf{G}(\mathbf{x}_t, t)\mathbf{G}^T(\mathbf{x}_t, t))$[1]. It shows the non-diagonal and data-dependent covariances of $\mathbf{G}\mathbf{G}^T$ in red ellipses of a nonlinear volatility term, and the isotropic blue circles of linear diffusions.

# 4   Implicit Nonlinear Diffusion Model

There are two ways to nonlinearize the drift and volatility coefficients in SDE: explicit and implicit parametrizations. While explicit is a straightforward way to model the nonlinearity, it becomes impractical particularly in the training procedure. Concretely, in each of the training iteration, the denoising loss $\mathcal{L}(\{\mathbf{x}_t\}_{t=0}^T, \lambda; \boldsymbol{\theta})$ requires 1) the perturbed samples $\mathbf{x}_t$ from $p_{0t}(\mathbf{x}_t|\mathbf{x}_0)$ and 2) the calculation of $\nabla \log p_{0t}(\mathbf{x}_t|\mathbf{x}_0)$, and these two steps require long execution time because the transition probability $p_{0t}(\mathbf{x}_t|\mathbf{x}_0)$ is intractable for nonlinear diffusions in general. Therefore, we parametrize $\mathbf{f}_{\boldsymbol{\phi}}$ and $\mathbf{G}_{\boldsymbol{\phi}}$ *implicitly* for fast and tractable optimization. As visualized in Figure 3, we impose a linear diffusion model on the latent space, and connect this latent variable with the data variable through a normalizing flow. The nonlinear diffusion on the data space, then, is induced from the latent diffusion leveraged to the data space.

## 4.1   Data and Latent Diffusion Processes

**Latent Diffusion Processes** Let us define $\mathbf{z}_0^{\boldsymbol{\phi}}$ to be a transformed latent variable $\mathbf{z}_0^{\boldsymbol{\phi}} = \mathbf{h}_{\boldsymbol{\phi}}(\mathbf{x}_0)$, where $\mathbf{h}_{\boldsymbol{\phi}}$ is a transformation of the normalizing flow. Then, a forward linear diffusion

(Latent Forward SDE)
$$\mathrm{d}\mathbf{z}_t^{\boldsymbol{\phi}} = -\frac{1}{2}\beta(t)\mathbf{z}_t^{\boldsymbol{\phi}}\,\mathrm{d}t + g(t)\,\mathrm{d}\mathbf{w}_t,$$

---

[1]The covariance is $\frac{\mathrm{d}}{\mathrm{d}t}\mathbb{E}_{\mathbf{x}_{t+\mathrm{d}t}|\mathbf{x}_t}[(\mathbf{x}_{t+\mathrm{d}t} - \mathbf{x}_t - \mathbf{f}\,\mathrm{d}t)(\mathbf{x}_{t+\mathrm{d}t} - \mathbf{x}_t - \mathbf{f}\,\mathrm{d}t)^T] = \mathbf{G}(\mathbf{x}_t, t)\mathbf{G}^T(\mathbf{x}_t, t)$.

starting at $\mathbf{z}_0^{\boldsymbol{\phi}} = \mathbf{h}_{\boldsymbol{\phi}}(\mathbf{x}_0)$ with $\mathbf{x}_0 \sim p_r$, describes the forward diffusion process on the latent space (blue diffusion path in Figure 3). The corresponding reverse latent diffusion is given by $\mathrm{d}\mathbf{z}_t^{\boldsymbol{\phi}} = [-\frac{1}{2}\beta(t)\mathbf{z}_t^{\boldsymbol{\phi}} - g^2(t)\nabla_{\mathbf{z}_t^{\boldsymbol{\phi}}} \log p_t^{\boldsymbol{\phi}}(\mathbf{z}_t^{\boldsymbol{\phi}})]\,\mathrm{d}\bar{t} + g(t)\,\mathrm{d}\bar{\mathbf{w}}_t$, where $p_t^{\boldsymbol{\phi}}$ is the probability of $\mathbf{z}_t^{\boldsymbol{\phi}}$.

**Forward Data Diffusion** We have not defined the data diffusion process yet. We build the data diffusion from the latent diffusion and the normalizing flow. From the invertibility, we define random variables on the data space by transforming the latent linear diffusion back to the data space: $\mathbf{x}_t^{\boldsymbol{\phi}} := \mathbf{h}_{\boldsymbol{\phi}}^{-1}(\mathbf{z}_t^{\boldsymbol{\phi}})$ for any $t \in [0, T]$. Then, from the Ito's formula [17], the process $\{\mathbf{x}_t^{\boldsymbol{\phi}}\}_{t=0}^T$ follows

(Data Forward SDE) $$\mathrm{d}\mathbf{x}_t^{\boldsymbol{\phi}} = \mathbf{f}_{\boldsymbol{\phi}}(\mathbf{x}_t^{\boldsymbol{\phi}}, t)\,\mathrm{d}t + \mathbf{G}_{\boldsymbol{\phi}}(\mathbf{x}_t^{\boldsymbol{\phi}}, t)\,\mathrm{d}\mathbf{w}_t,$$

starting with $\mathbf{x}_0^{\boldsymbol{\phi}} = \mathbf{h}_{\boldsymbol{\phi}}^{-1}(\mathbf{z}_0^{\boldsymbol{\phi}})$. From $\mathbf{x}_0^{\boldsymbol{\phi}} = \mathbf{h}_{\boldsymbol{\phi}}^{-1}(\mathbf{h}_{\boldsymbol{\phi}}(\mathbf{x}_0)) = \mathbf{x}_0 \sim p_r$, we call this process by *induced diffusion* that permeates the data variable on the data space. We emphasize that this induced diffusion collapses to a linear diffusion if $\mathbf{h}_{\boldsymbol{\phi}_{id}} = id$. See Appendix A.2 for details on drift and volatility terms.

**Generative Data Diffusion** A diffusion model estimates the forward latent score $\mathbf{s}_{\boldsymbol{\phi}}(\mathbf{z}, t) = \nabla \log p_t^{\boldsymbol{\phi}}(\mathbf{z})$ with the score network, $\mathbf{s}_{\boldsymbol{\theta}}(\mathbf{z}, t)$, to mimic the forward linear diffusion on the latent space. Then, the generative SDE on the latent space becomes

(Latent Gen. SDE) $$\mathrm{d}\mathbf{z}_t^{\boldsymbol{\theta}} = \left[-\frac{1}{2}\beta(t)\mathbf{z}_t^{\boldsymbol{\theta}} - g^2(t)\mathbf{s}_{\boldsymbol{\theta}}(\mathbf{z}_t^{\boldsymbol{\theta}}, t)\right]\mathrm{d}\bar{t} + g(t)\,\mathrm{d}\bar{\mathbf{w}}_t$$

with a starting variable $\mathbf{z}_T^{\boldsymbol{\theta}} \sim \pi$. Thus, the process $\{\mathbf{x}_t^{\boldsymbol{\phi},\boldsymbol{\theta}}\}_{t=0}^T$ of $\mathbf{x}_t^{\boldsymbol{\phi},\boldsymbol{\theta}} := \mathbf{h}_{\boldsymbol{\phi}}^{-1}(\mathbf{z}_t^{\boldsymbol{\theta}})$ becomes a generative data diffusion (purple path in Figure 3) with SDE of

(Data Gen. SDE) $\mathrm{d}\mathbf{x}_t^{\boldsymbol{\phi},\boldsymbol{\theta}} = \left[\mathbf{f}_{\boldsymbol{\phi}} - \mathrm{div}(\mathbf{G}_{\boldsymbol{\phi}}\mathbf{G}_{\boldsymbol{\phi}}^T) - (\mathbf{G}_{\boldsymbol{\phi}}\mathbf{G}_{\boldsymbol{\phi}}^T\nabla\mathbf{h}_{\boldsymbol{\phi}})\mathbf{s}_{\boldsymbol{\theta}}\big(\mathbf{h}_{\boldsymbol{\phi}}(\mathbf{x}_t^{\boldsymbol{\phi},\boldsymbol{\theta}}), t\big)\right]\mathrm{d}\bar{t} + \mathbf{G}_{\boldsymbol{\phi}}\,\mathrm{d}\bar{\mathbf{w}}_t.$

## 4.2 Model Training and Sampling

**Likelihood Training** Theorem 1 estimates Negative Evidence Lower Bound (NELBO) of Negaitve Log-Likelihood (NLL). For the notational simplicity, we define the targetted score function by

(Target of Score Estimation) $$\mathbf{s}_{\boldsymbol{\phi}}(\mathbf{z}_t^{\boldsymbol{\phi}}, t) := \nabla \log p_t^{\boldsymbol{\phi}}(\mathbf{z}_t^{\boldsymbol{\phi}}).$$

Also, suppose $\mathcal{L}\big(\{\mathbf{z}_t^{\boldsymbol{\phi}}\}_{t=0}^T, g^2; \boldsymbol{\theta}\big) = \frac{1}{2}\int_0^T g^2(t)\mathbb{E}_{\mathbf{z}_0^{\boldsymbol{\phi}}, \mathbf{z}_t^{\boldsymbol{\phi}}}\big[\|\mathbf{s}_{\boldsymbol{\theta}}(\mathbf{z}_t^{\boldsymbol{\phi}}, t) - \nabla\log p_{0t}(\mathbf{z}_t^{\boldsymbol{\phi}}|\mathbf{z}_0^{\boldsymbol{\phi}})\|_2^2\big]\,\mathrm{d}t$, where $p_{0t}(\mathbf{z}_t^{\boldsymbol{\phi}}|\mathbf{z}_0^{\boldsymbol{\phi}})$ is the transition probability of the latent forward diffusion. In Theorem 1, we drop the constant terms that do not hurt the essence of the theorem to keep the simplicity. See full details and the proof in Appendix G.

**Theorem 1.** *Suppose that $p_{\boldsymbol{\phi},\boldsymbol{\theta}}$ is the likelihood of a generative random variable $\mathbf{x}_0^{\boldsymbol{\phi},\boldsymbol{\theta}}$. Then, the negative log-likelihood is upper bounded by*

$$\mathbb{E}_{\mathbf{x}_0}\big[-\log p_{\boldsymbol{\phi},\boldsymbol{\theta}}(\mathbf{x}_0)\big] \leq \mathcal{L}\big(\{\mathbf{x}_t\}_{t=0}^T, g^2; \{\boldsymbol{\phi}, \boldsymbol{\theta}\}\big),$$

*where*

$$\mathcal{L}\big(\{\mathbf{x}_t\}_{t=0}^T, g^2; \{\boldsymbol{\phi}, \boldsymbol{\theta}\}\big) = \frac{1}{2}\int_0^T g^2(t)\mathbb{E}_{\mathbf{z}_t^{\boldsymbol{\phi}}}\big[\|\mathbf{s}_{\boldsymbol{\theta}}(\mathbf{z}_t^{\boldsymbol{\phi}}, t) - \mathbf{s}_{\boldsymbol{\phi}}(\mathbf{z}_t^{\boldsymbol{\phi}}, t)\|_2^2\big]\,\mathrm{d}t + D_{KL}(p_T^{\boldsymbol{\phi}}\|\pi) \; (1)$$

$$= -\mathbb{E}_{\mathbf{x}_0}\big[\log\big|\det\big(\nabla\mathbf{h}_{\boldsymbol{\phi}}(\mathbf{x}_0)\big)\big|\big] + \mathcal{L}\big(\{\mathbf{z}_t^{\boldsymbol{\phi}}\}_{t=0}^T, g^2; \boldsymbol{\theta}\big) - \mathbb{E}_{\mathbf{z}_T^{\boldsymbol{\phi}}}\big[\log\pi(\mathbf{z}_T^{\boldsymbol{\phi}})\big]. \quad (2)$$

Eq. (1) is the KL divergence $D_{KL}(\boldsymbol{\mu}_{\boldsymbol{\phi}}\|\boldsymbol{\nu}_{\boldsymbol{\phi},\boldsymbol{\theta}})$, where $\boldsymbol{\mu}_{\boldsymbol{\phi}}$ and $\boldsymbol{\nu}_{\boldsymbol{\phi},\boldsymbol{\theta}}$ are the path measures of the forward and generative diffusions on the data space. Eq. (1) explains the reasoning of why $\mathbf{s}_{\boldsymbol{\phi}}$ is the target of the score estimation. However, the KL divergence is intractable, and Theorem 1 provides an equivalent tractable loss by Eq. (2), the summation of the flow loss with the denoising loss.

Algorithm 1 describes the line-by-line algorithm of INDM training. We obtain the flow loss by taking a flow evaluation. Afterward, we compute the denoising loss. We train the flow with Eq. (2). However, we train the score with $\mathcal{L}\big(\{\mathbf{x}_t\}_{t=0}^T, \lambda; \{\boldsymbol{\phi}, \boldsymbol{\theta}\}\big)$ with various $\lambda$ settings for a better Fréchet Inception Distance (FID) [18].

**Latent Sampling** While either of red or purple path in Figure 3 could synthesize the samples, we choose the red path for the fast sampling (because the red path feed-forwards the flow only once). Starting from a pure noise $\mathbf{z}_T^{\boldsymbol{\theta}} \sim \pi$, we denoise $\mathbf{z}_T^{\boldsymbol{\theta}}$ to $\mathbf{z}_0^{\boldsymbol{\theta}}$ by solving the generative process backward on the latent space. Then, we transform the fully denoised latent $\mathbf{z}_0^{\boldsymbol{\theta}}$ to the data space $\mathbf{x}_0^{\boldsymbol{\phi},\boldsymbol{\theta}} = \mathbf{h}_{\boldsymbol{\phi}}^{-1}(\mathbf{z}_0^{\boldsymbol{\theta}})$.

Table 1: Comparison of INDM with previous works. $N$ is the number of random variables.

| Model | Nonlinear Diffusion | Implemented Data Diffusion | Latent Diffusion | Nonlinear **f**-Modeling | Nonlinear **G**-Modeling | Explicit **f**&**G** Derived | Training Complexity | Sampling Cost |
|-------|---------------------|----------------------------|------------------|-------------------------|-------------------------|------------------------------|---------------------|---------------|
| DDPM++ | ✗ | Continuous | ✗ | ✗ | ✗ | ✓ | $O(1)$ | ↓ |
| LSGM | ✗ | ✗ | Continuous | ✗ | ✗ | ✗ | $O(1)$ | ↓ |
| SBP | △ | Discrete | ✗ | Explicit | ✗ | ✓ | $O(N)$ | ↓ |
| DiffFlow | ✓ | Discrete | ✗ | Explicit | ✗ | ✓ | $O(N)$ | ↑ |
| INDM | ✓ | Continuous | Continuous | Implicit | Implicit | ✓ | $O(1)$ | ↓ |

# 5 Related Work

In this section, we compare INDM with previous works, and summarize our arguments in Table 1.

**LSGM** Vahdat et al. [9] put a linear diffusion on the latent space like INDM but uses an auto-encoder structure. From this modeling choice, LSGM cannot be categorized as a nonlinear diffusion model in a strict sense. Concretely, recall that a diffusion process is (mathematically) defined as a sequence of random variables connected via a Markov chain. From this definition, one needs to satisfy two requirements to call it a

---

**Algorithm 1** Implicit Nonlinear Diffusion Model

1: **repeat**
2:      Get latent with flow by $\mathbf{z}_0^\phi = \mathbf{h}_\phi(\mathbf{x}_0)$ for $\mathbf{x}_0 \sim p_r$
3:      Compute $-\mathbb{E}_{\mathbf{x}_0}\big[\log \big| \det \big(\nabla \mathbf{h}_\phi(\mathbf{x}_0)\big)\big|\big]$
4:      Get diffused latents $\{\mathbf{z}_t^\phi\}_{t=0}^T$ with a linear SDE
5:      Compute $\mathcal{L}\big(\{\mathbf{z}_t^\phi\}_{t=0}^T, g^2; \boldsymbol{\theta}\big) - \mathbb{E}_{\mathbf{z}_T^\phi}\big[\log \pi(\mathbf{z}_T^\phi)\big]$
6:      Compute flow loss $\mathcal{L}_f = \mathcal{L}\big(\{\mathbf{x}_t\}_{t=0}^T, g^2; \{\phi, \boldsymbol{\theta}\}\big)$
7:      Update $\phi \leftarrow \phi - \eta \frac{\partial \mathcal{L}_f}{\partial \phi}$
8:      Compute $\mathcal{L}\big(\{\mathbf{z}_t^\phi\}_{t=0}^T, \lambda; \boldsymbol{\theta}\big) - \mathbb{E}_{\mathbf{z}_T^\phi}\big[\log \pi(\mathbf{z}_T^\phi)\big]$
9:      Compute score loss $\mathcal{L}_s = \mathcal{L}\big(\{\mathbf{x}_t\}_{t=0}^T, \lambda; \{\phi, \boldsymbol{\theta}\}\big)$
10:     Update $\boldsymbol{\theta} \leftarrow \boldsymbol{\theta} - \eta \frac{\partial \mathcal{L}_s}{\partial \boldsymbol{\theta}}$
11: **until** converged

---

diffusion process: 1) there must be multiple (possibly infinite) numbers of random variables; 2) the random variables should be connected via a Markov chain. Unlike INDM, LSGM cannot build forward data variables from the forward latent variables because there is no exact inverse function of the encoder map, as long as the data dimension differs to the latent dimension (Lemma 3 of Appendix D.1). This leads that LSGM has no forward data diffusion process. From this point, analyzing the data nonlinearity becomes infeasible in LSGM.

Moreover, LSGM has a pair of key differences in its training. First, the latent dimension of LSGM is 40,080, which is *15× higher* dimension than the data dimension (3,072) on CIFAR-10 [19]. In contrast, INDM always keeps its latent dimension by the data dimension. See Table 9 to compare the latent dimensions of INDM with LSGM on benchmark datasets. Furthermore, LSGM is repeatedly reported [9, 20] for its training instability on the best FID setting of $\lambda = \sigma^2$ (i.e., $L_{simple}$ [8]). Meanwhile, INDM is consistently stable for any of training configurations, see Table 8.

**DiffFlow** Zhang and Chen [13] explicitly model the drift term $\mathbf{f}_\phi$ as a flow network, so the forward diffusion becomes $\mathrm{d}\mathbf{x}_t = \mathbf{f}_\phi \, \mathrm{d}t + g \, \mathrm{d}\mathbf{w}_t$. However, there are differences between DiffFlow and INDM: 1) DiffFlow does not nonlinearize the volatility; 2) DiffFlow is too slow for its explicit parametrization (Table 18); 3) the flexibility of $\mathbf{f}_\phi$ is too restricted; 4) DiffFlow has a larger loss variance (Table 10). See Appendix D.2 for the full details of our arguments. Focusing on the slow training, observe that the denoising loss $\mathbb{E}_{\mathbf{x}_0, \mathbf{x}_t^\phi}[\|\mathbf{s}_{\boldsymbol{\theta}}(\mathbf{x}_t^\phi, t) - \nabla \log p_{0t}(\mathbf{x}_t^\phi | \mathbf{x}_0)\|_2^2]$ requires a pair of heavy computations: (A) sampling from $\mathbf{x}_t^\phi$, and (B) computation of $\nabla \log p_{0t}(\mathbf{x}_t^\phi | \mathbf{x}_0)$. Intractable transition probability $p_{0t}(\mathbf{x}_t^\phi | \mathbf{x}_0)$ is the major bottleneck of the slow training.

To overcome the bottleneck, DiffFlow discretizes the continuous diffusion with $N$ variables of a discrete diffusion and uses the DDPM-style loss [8], which does not need to calculate the transition probability. Under the discretization, however, the forward sampling of $\mathbf{x}_t^\phi$ takes $O(N)$ flow evaluations for every network update. This sampling issue is an inevitable fundamental problem when we parametrize the coefficients explicitly. Having that the flow evaluation is generally more expensive than score evaluation given the same number of parameters, a fast sampling is achievable only if we reduce $N$. However, it hurts the flexibility of a diffusion process, so DiffFlow suffers from the trade-off between training speed and model flexibility. On the other hand, the training of INDM is invariant of $N$, and INDM is free from such a trade-off. Analogously, DiffFlow generates a sample with the purple path in Figure 3, so it takes $O(N)$ flow evaluations, contrastive to INDM with a single flow evaluation in its sampling with the red path.

**SBP** De Bortoli et al. [15] learn the diffusion process with a problem of $\min_{\boldsymbol{\rho_\theta} \in \mathcal{P}(p_r, \pi)} D_{KL}(\boldsymbol{\rho_\theta} \| \boldsymbol{\mu})$, where $\mathcal{P}(p_r, \pi)$ is the collection of path measure with $p_r$ and $\pi$ as its marginal distributions at $t = 0$ and $t = T$, respectively. It is a bi-constrained optimization problem as any path measure on the search space that should satisfy boundary conditions at both $t = 0$ and $t = T$. $\boldsymbol{\mu}$ is the reference measure of a linear diffusion $\mathrm{d}\mathbf{x}_t = \mathbf{f}(\mathbf{x}_t, t) \, \mathrm{d}t + g(t)\mathbf{w}_t$; and the forward and reverse SDEs of $\boldsymbol{\rho_\theta}$ are $\mathrm{d}\mathbf{x}_t = [\mathbf{f}(\mathbf{x}_t, t) + g^2(t)\nabla \log \Psi(\mathbf{x}_t, t)] \, \mathrm{d}t + g(t) \, \mathrm{d}\mathbf{w}_t$ and $\mathrm{d}\mathbf{x}_t = [\mathbf{f}(\mathbf{x}_t, t) - g^2(t)\nabla \log \hat{\Psi}(\mathbf{x}_t, t)] \, \mathrm{d}\bar{t} + g(t) \, \mathrm{d}\bar{\mathbf{w}}_t$, respectively, where $(\Psi, \hat{\Psi})$ is the solution of a coupled PDE, called Hopf-Cole transform [21]. Solving this coupled PDE is intractable, so the estimation target of SBP is $\nabla \log \Psi$ and $\nabla \log \hat{\Psi}$. As $\mathbf{f}$ and $g$ are assumed to be linear functions, the nonlinearity of SBP is fully determined by $(\Psi, \hat{\Psi})$.

Analogous to DiffFlow, sampling from $\mathbf{x}_t$ in SBP needs a long time. Few works [15, 16] detour this training issue using the experience replay memory. Aside from the training time, the KL minimization puts the global optimal nonlinear diffusion $\boldsymbol{\rho_{\theta^*}}$ near a neighborhood of the linear diffusion $\boldsymbol{\mu}$. In other words, the optimal $\boldsymbol{\rho_{\theta^*}}$ is the closest path measure on $\mathcal{P}(p_r, \pi)$ to $\boldsymbol{\mu}$, so the inferred nonlinear diffusion would be the *most* linear diffusion on the space of $\mathcal{P}(p_r, \pi)$. For the demonstration, we illustrate

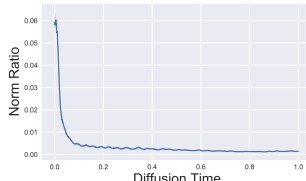

Figure 4: Norm Ratio of SBP.

$\|g^2(t)\nabla \log \Psi(\mathbf{x}_t, t)\Delta t\|_2 / \|g(t)\Delta \mathbf{w}_t\|_2$ in Figure 4. We used the released checkpoint of SB-FBSDE [16], an algorithm for solving SBP, trained with VESDE on CIFAR-10. As $\mathbf{f} \equiv 0$ in VESDE, this norm ratio measures how much nonlinearity is counted on a diffusion trajectory compared to the linear effect. Figure 4 shows that the ratio approaches zero except at the small range around $t \approx 0$, meaning that the nonlinear effect is virtually ignorable than the linear effect. Therefore, Figure 4 implies that the diffusion process is nearly linear in most of the diffusion time. We give a detailed discussion of SBP in Appendix D.3.

## 6 Discussion

This section investigates characteristics of INDM. We show that INDM training is faster and nearly MLE in Section 6.1, and INDM sampling is robust on discretization step sizes in Section 6.2.

### 6.1 Benefit of INDM in Training

Having that DDPM++ is a collapsed INDM with a fixed identity transformation $\mathbf{h}_{\phi_{id}} = id$, the difference lies in whether to train $\phi$ or not. This trainable nonlinearity provides the better optimization of INDM, as evidenced in Figure 5-(a), experimented on CIFAR-10 using VPSDE. It shows a pair of critical characteristics of INDM training: 1) it is faster than DDPM++ training, and 2) it is asymptotically an MLE training. For the training speed, recall that the regression target of the score estimation is $\mathbf{s}_\phi$, and this target is fixed in DDPM++ while keep moving in INDM. The target is constantly updated through the direction of $\mathbf{s}_\theta$ in Eq. (1) by optimizing $\|\mathbf{s}_\phi - \mathbf{s}_\theta\|_2^2$. This *bidirectional* attraction between $\mathbf{s}_\theta$ and $\mathbf{s}_\phi$ is what flow learning does in the optimization.

For the MLE training, as the flow training is intricately entangled with the score training, we analyze INDM training for a specific class of score networks. First, we define $\mathbf{S}_{sol}$ (Definition 1 in Appendix B) to be the class of forward score functions of a linear diffusion with some initial distribution. Then, it turns out that it is the whole class of zero variational gap (=NLL-NELBO).

**Theorem 2.** $\mathrm{Gap}(\boldsymbol{\mu_\phi}, \boldsymbol{\nu_{\phi,\theta}}) := D_{KL}(\boldsymbol{\mu_\phi} \| \boldsymbol{\nu_{\phi,\theta}}) - D_{KL}(p_r \| p_{\phi,\theta}) = 0$ *if and only if* $\mathbf{s}_\theta \in \mathbf{S}_{sol}$.

Song et al. [11] partially reveal the connection between the gap with $\mathbf{S}_{sol}$, by proving the *if* part of Theorem 2, in Theorem 2 of Song et al. [11] (see Lemma 2 in Appendix B). We completely characterize this connection by proving the *only-if* part in Theorem 2. Surprisingly, the variational gap is irrelevant to the flow parameters, and the MLE training of INDM implies that the score network is nearby $\mathbf{S}_{sol}$ throughout the training. Combining Theorem 2 with the global optimality analysis raises a qualitative discrepancy in the optimization of DDPM++ and INDM by Theorem 3.

**Theorem 3.** *For any fixed* $\mathbf{s}_\theta \in \mathbf{S}_{sol}$, *if* $\phi^* \in \arg\min_\phi D_{KL}(\boldsymbol{\mu_\phi} \| \boldsymbol{\nu_{\phi,\theta}})$, *then* $\mathbf{s}_{\phi^*}(\mathbf{z}, t) = \nabla \log p_t^{\phi^*}(\mathbf{z}) = \mathbf{s}_\theta(\mathbf{z}, t)$, *and* $D_{KL}(\boldsymbol{\mu_{\phi^*}} \| \boldsymbol{\nu_{\phi^*,\theta}}) = D_{KL}(p_r \| p_{\phi^*,\theta}) = Gap(\boldsymbol{\mu_{\phi^*}}, \boldsymbol{\nu_{\phi^*,\theta}}) = 0$.

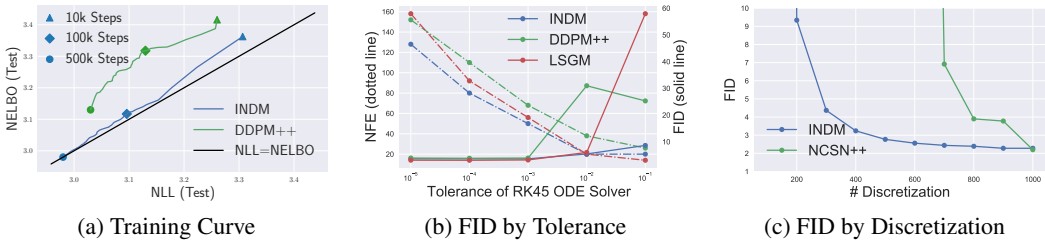

| (a) Training Curve | (b) FID by Tolerance | (c) FID by Discretization |

Figure 5: Comparison of INDM with baseline models, experimented on CIFAR-10.

Theorem 3 implies that there exists an optimal flow that the forward and generative SDEs on the latent space coincide, for any score network in $\mathbf{S}_{sol}$, if the flow is flexible enough. Therefore, INDM attains infinitely many ($= |\mathbf{S}_{sol}|$) global optimums in its optimization space. On the other hand, DDPM++ has only a unique optimal score network, i.e., $\mathbf{s}_{\boldsymbol{\theta}^*} = \mathbf{s}_{\boldsymbol{\phi}_{id}}$. Thus, Theorem 3 potentially explains the faster convergence of INDM. We give a detailed analysis in Appendix B.

## 6.2 Benefit of INDM in Sampling

Figure 5-(b,c) equally illustrate that INDM is more robust on the discretization step sizes in FID than DDPM++/NCSN++. To analyze the sample quality with respect to discretizations, recall that the Euler-Maruyama discretization of the generative SDE (or called the reverse diffusion sampler, or simply the predictor [1]) iteratively updates the sample $\tilde{\mathbf{z}}_k$ with

$$\tilde{\mathbf{z}}_{t_{k-1}} = \tilde{\mathbf{z}}_{t_k} + \gamma_k \left( \frac{1}{2}\beta(t_k)\tilde{\mathbf{z}}_{t_k} + g^2(t_k)\mathbf{s}_{\boldsymbol{\theta}}(\tilde{\mathbf{z}}_{t_k}, t_k) \right) + g(t_k)\sqrt{\gamma_k}\boldsymbol{\epsilon},$$

until time reaches to zero, where $\gamma_k = t_k - t_{k-1}$ is the step size of the discretized sampler and $\boldsymbol{\epsilon} \sim \mathcal{N}(0, \mathbf{I})$. The sampling error is the distributional discrepancy between the sample distribution of $\mathbf{h}_{\boldsymbol{\phi}}^{-1}(\tilde{\mathbf{z}}_0)$ and the data distribution. Theorem 4 decomposes the sampling error with three factors: 1) the prior error $E_{pri}$, 2) the discretization error $E_{dis}$, and 3) the score error $E_{est}$. Note that Theorem 4 is a straightforward application of the analysis done by De Bortoli et al. [15] and Guth et al. [22]. We omit regularity conditions to avoid unnecessary complications; see Appendix C.1.

**Theorem 4** (De Bortoli et al. [15] and Guth et al. [22]). *Assume that 1)* $\sup_{\mathbf{z},t} \|\mathbf{s}_{\boldsymbol{\theta}^*}(\mathbf{z}, t) - \nabla \log p_t^{\boldsymbol{\phi}}(\mathbf{z})\| \leq M$, *2)* $\sup_{\mathbf{z},t} \|\nabla^2 \log p_t^{\boldsymbol{\phi}}(\mathbf{z})\| \leq K$, *and 3)* $\sup_{\mathbf{z},t} \|\partial_t \nabla \log p_t^{\boldsymbol{\phi}}(\mathbf{z})\|/\|\mathbf{z}\| \leq Le^{-\alpha t}$, *for some* $K, L, M, \alpha > 0$. *Then*

$$\|p_r - (\mathbf{h}_{\boldsymbol{\phi}}^{-1})_\# \circ p_{0,N}^{\boldsymbol{\theta}}\|_{TV} \leq E_{pri}(\boldsymbol{\phi}) + E_{dis}(\boldsymbol{\phi}) + E_{est}(\boldsymbol{\phi}, \boldsymbol{\theta}) + o(\sqrt{\delta} + e^{-T}),$$

*where* $E_{pri}(\boldsymbol{\phi}) = \sqrt{2}e^{-T}D_{KL}(p_T^{\boldsymbol{\phi}}\|\pi)^{1/2}$, $E_{dis}(\boldsymbol{\phi}) = 6\sqrt{\delta}(1+\mathbb{E}[\|\mathbf{z}\|^4]^{1/4})(1+K+L(1+1/\sqrt{2\alpha}))$, *and* $E_{est}(\boldsymbol{\phi}, \boldsymbol{\theta}) = 2TM^2$ *with* $\delta = \max \gamma_k{}^2 / \min \gamma_k$.

There are a pair of implications from Theorem 4.

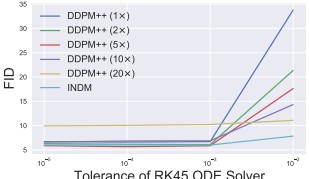

✓ $E_{pri}(\boldsymbol{\phi})$ and $E_{est}(\boldsymbol{\phi}, \boldsymbol{\theta})$ are independent of the discretization steps.

✓ $E_{dis}(\boldsymbol{\phi})/\sqrt{\delta}$ is the discretization sensitivity, entirely determined by the latent distribution's smoothness.

To the deep understanding of the second implication, let us assume $\mathbf{h}_{\boldsymbol{\phi}_a}(\mathbf{x}) = a\mathbf{x}$ for some scalar $a > 1$, then the sensitivity is antiproportional to $a$ with the identical discretizations, i.e., $E_{dis}(\boldsymbol{\phi}_a) \approx \frac{1}{a}E_{dis}(\boldsymbol{\phi}_{id})$. With a smaller sensitivity of $\boldsymbol{\phi}_a$, there is more room to

Figure 6: Sensitivity analysis on scaled-up scenario.

reduce the number of discretization steps for $\boldsymbol{\phi}_a$. Figure 6 empirically supports the theory, showing that the sampler (at the large tolerance with $10^{-2}$) becomes more robust as $a$ increases, on CIFAR-10.

Before we derive a concrete result from the implication, observe that the flow $\mathbf{h}_{\boldsymbol{\phi}}$ is maximizing $\det(\nabla \mathbf{h}_{\boldsymbol{\phi}})$ in Eq. (2). To understand the effect of flow training on the discretization sensitivity, let us restrict the hypothesis class of the transformation to be linear mappings of $\mathbf{h}_{\boldsymbol{\phi}_a}(\mathbf{x}) = a\mathbf{x}$. Then, as the determinant increases by $a$, the trained diffusion model would be insensitive to the discretizations. Now, for the general case, Figure 7-(a,b) illustrate $\|\nabla^2 \log p_t^{\boldsymbol{\phi}}(\mathbf{z})\|$ and $\|\partial_t \nabla \log p_t^{\boldsymbol{\phi}}(\mathbf{z})\|/\|\mathbf{z}\|$ on

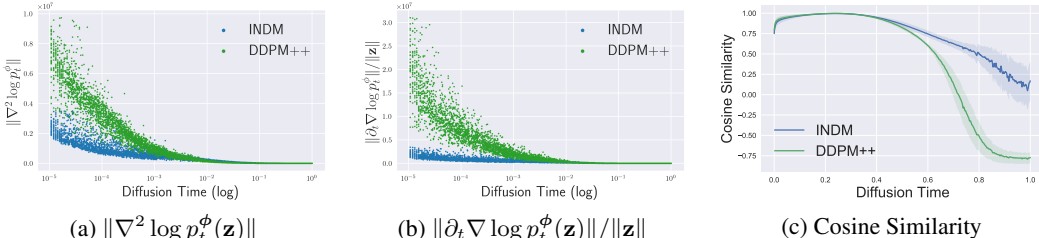

(a) $\|\nabla^2 \log p_t^\phi(\mathbf{z})\|$      (b) $\|\partial_t \nabla \log p_t^\phi(\mathbf{z})\|/\|\mathbf{z}\|$      (c) Cosine Similarity

Figure 7: (a,b) Comparison of INDM with DDPM++ for $K, L, \alpha$. (c) Cosine similarity of forward diffusion trajectories on CIFAR-10.

CIFAR-10, respectively. Also, $\mathbb{E}[\|\mathbf{z}\|^4]^{1/4}$ of INDM is slightly larger (1.3x) than DDPM++. Therefore, with these observations combined, we conclude that INDM is less sensitive to the discretization steps than DDPM++, from its loss design.

Second, the robustness could originate from the geometry of the diffusion trajectory. The forward solution of VPSDE is $\mathbf{x}_t = \mu(t)\mathbf{x}_0 + \sqrt{1-\mu^2(t)}\epsilon$ with $\mu(t) = e^{-\frac{1}{2}\int_0^t \beta(s)\,\mathrm{d}s}$, where the first term is a contraction mapping and the second term is the random perturbation. The contraction mapping points toward the origin, but the overall vector field of the diffusion path points outward because the prior manifold lies outside the data manifold, as shown in Table 2. This contrastive force leads the drift and volatility coefficients works in repulsive and raises a highly nonlinear

Table 2: Average $L_2^2$ Norm.

| Manifold | Norm |
|---|---|
| Data | 776 |
| Latent | 5,385 |
| Prior | 3,072 |

diffusion trajectory in DDPM++, see Figure 11 for a toy illustration. On the other hand, the flow mapping of INDM pushes the latent manifold outside the prior manifold, and the drift and volatility coefficients act coherently. Hence, INDM has the relatively linear diffusion path; see Figures 12, 13, and 14 for a quick intuition. Figure 7-(c) measures the cosine similarity of the ODE's diffusion trajectory with the straight line connecting the initial-final points of each trajectory. Figure 7-(c) implies that DDPM++ is under an inefficient nonlinear trajectory that reverts backward near the end of the trajectory, as in Figure 11. In contrast, INDM trajectory is relatively efficient and linear (Figure 15), which yields robust sampling by discretization steps; see Appendix C.2 for details.

## 7 Experiments

This section quantitatively analyzes suggested INDM on CIFAR-10 and CelebA $64 \times 64$. Throughout the experiments, we use NCSN++ with VESDE and DDPM++ with VPSDE [1] as the backbones of diffusion models, and a ResNet-based flow model [23, 24] as the backbone of the flow model. See Appendix F for experimental details. We experiment with a pair of weighting functions for the score training. One is the likelihood weighting [11] with $\lambda(t) = g^2(t)$, and we denote INDM (NLL) for this weighing choice. The other is the variance weighting [8] $\lambda(t) = \sigma^2(t)$ with an emphasis on FID, and we denote INDM (FID) for this weighting choice.

We use either the Predictor-Corrector (PC) sampler [1] or a numerical ODE solver (RK45 [25]) of the probability flow ODE [1]. For a better FID, we find the optimal signal-to-noise value (Table 14), sampling temperature (Table 15), and stopping time (Table 16). Moreover, sampling from $\mathbf{z}_T^\phi$ rather than $\pi$ improves FID because $E_{pri}(\phi)$ collapses to zero in Theorem 4, see Appendix F.2.2. We compute NLL/NELBO for performances of density estimation with Bits Per Dimension (BPD). We compute NLL with the uniform dequantization, instead of the variational dequantization [26] because it requires training an auxiliary network [11] only for the evaluation after the model training.

### 7.1 Correction on Likelihood Evaluation

A continuous diffusion model truncates the time horizon from $[0, T]$ to $[\epsilon, T]$ to avoid training instability [27]. In the model evaluation, this positive truncation could potentially be the primary source of poor evaluation (Figure 1-(c) of Kim et al. [27]), so we fix $\epsilon = 10^{-5}$ as default in our training and evaluation. In the model evaluation, as the score network is untrained on $[0, \epsilon]$, we calculate NLL by the Right-Hand-Side (RHS) of Eq. (3),

$$\text{NLL} = \mathbb{E}_{\mathbf{x}_0}[-\log p_0^m(\mathbf{x}_0)] \leq \mathbb{E}_{\mathbf{x}_0, \mathbf{x}_\epsilon}\left[-\log p_\epsilon^m(\mathbf{x}_\epsilon) + \log \frac{p_{\epsilon 0}^m(\mathbf{x}_0|\mathbf{x}_\epsilon)}{p_{0\epsilon}(\mathbf{x}_\epsilon|\mathbf{x}_0)}\right]. \tag{3}$$

Table 4: Performance comparison to linear/nonlinear diffusion models on CIFAR-10. We report the performance of linear diffusions by training our PyTorch implementation based on Song et al. [1, 11] with identical hyperparameters and score networks on both linear/nonlinear diffusions to quantify the effect of nonlinearity in a fair setting. Boldface numbers represent the best performance in a column.

| SDE | Model | Nonlinear Data Diffusion | # Params | NLL (↓) | | NELBO (↓) | | Gap (↓) (=NELBO-NLL) | | FID (↓) | |
|---|---|---|---|---|---|---|---|---|---|---|---|
| | | | | after correction | before correction | w/ residual (after) | w/o residual (before) | after | before | ODE | PC |
| VE | NCSN++ (FID) | ✗ | 63M | 4.86 | 3.66 | 4.89 | 4.45 | 0.03 | 0.79 | - | 2.38 |
| | INDM (FID) | ✓ | 76M | 3.22 | 3.13 | 3.28 | 3.24 | 0.06 | 0.11 | - | **2.29** |
| VP | DDPM++ (FID) | ✗ | 62M | 3.21 | 3.16 | 3.34 | 3.32 | 0.13 | 0.16 | 3.90 | 2.89 |
| | INDM (FID) | ✓ | 75M | 3.17 | 3.11 | 3.23 | 3.18 | 0.06 | 0.07 | **3.61** | 2.90 |
| | DDPM++ (NLL) | ✗ | 62M | 3.03 | 2.97 | 3.13 | 3.11 | 0.10 | 0.14 | 6.70 | 5.17 |
| | INDM (NLL) | ✓ | 75M | **2.98** | **2.95** | **2.98** | **2.97** | **0.00** | **0.02** | 6.01 | 5.30 |

Table 5: Performance comparison on CIFAR-10.

| SDE | Type | Model | # Params | NLL | FID |
|---|---|---|---|---|---|
| Linear | | NCSN++ (FID) [1] | 108M | 4.85 | 2.20 |
| | | DDPM++ (FID) [1] | 108M | 3.19 | 2.64 |
| | | DDPM++ (NLL) [1] | 108M | 3.01 | 4.88 |
| | | VDM [28] | - | **2.65** | 7.41 |
| | | CLD-SGM [20] | 108M | 3.31 | 2.25 |
| Nonlinear | SBP | SB-FBSDE [16] | 102M | 2.98 | 3.18 |
| | VAE -based | LSGM (FID) [9] | 476M | 3.45 | **2.10** |
| | | LSGM (NLL) [9] | 269M | 2.97 | 6.15 |
| | | LSGM (NLL) [9] | 506M | 2.87 | 6.89 |
| | | LSGM (balanced) [9] | 109M | 2.96 | 4.60 |
| | | LSGM (balanced) [9] | 476M | 2.98 | 2.17 |
| | Flow -based | DiffFlow (FID) [13] | ≈36M | 3.04 | 14.14 |
| | | INDM (FID) | 118M | 3.09 | 2.28 |
| | | INDM (NLL) | 121M | 2.97 | 4.79 |
| | | INDM (NLL) + ST | 75M | 3.01 | 3.25 |

Table 6: Performance comparison on CelebA $64 \times 64$.

| Model | NLL | NELBO | Gap | FID |
|---|---|---|---|---|
| UNCSN++ [27] | **1.93** | - | - | 1.92 |
| DDGM [29] | - | - | - | 2.92 |
| Efficient-VDVAE [30] | - | 1.83 | - | - |
| CR-NVAE [31] | - | 1.86 | - | - |
| DenseFlow-74-10 [4] | 1.99 | - | - | - |
| StyleFormer [32] | - | - | - | 3.66 |
| NCSN++ (VE, FID) | 3.41 | 3.42 | 0.01 | 3.95 |
| INDM (VE, FID) | 2.31 | 2.33 | 0.02 | 2.54 |
| DDPM++ (VP, FID) | 2.14 | 2.21 | 0.07 | 2.32 |
| INDM (VP, FID) | 2.27 | 2.31 | 0.04 | **1.75** |
| DDPM++ (VP, NLL) | 2.00 | 2.09 | 0.09 | 3.95 |
| INDM (VP, NLL) | 2.05 | 2.05 | **0.00** | 3.06 |

Here, $p_0^m$ and $p_\epsilon^m$ are the model probability distributions at $t = 0$ and $t = \epsilon$, respectively; and $p_{\epsilon 0}^m(\cdot|\mathbf{x}_\epsilon)$ is the model's reconstruction probability of $\mathbf{x}_0$ given $\mathbf{x}_\epsilon$. RHS of Eq. (3) is a generic formula to compute NLL in continuous diffusion models, including DDPM++, LSGM, and INDM. Previous continuous models [1, 11, 9] have approximated $\mathbb{E}_{\mathbf{x}_0}[-\log p_0^m(\mathbf{x}_0)]$ by $\mathbb{E}_{\mathbf{x}_0}[-\log p_\epsilon^m(\mathbf{x}_0)]$.

There are two significant differences between our and the previous calculation: 1) the input of $p_\epsilon^m$ is replaced with $\mathbf{x}_\epsilon$ from $\mathbf{x}_0$ (Table 11); 2) the residual term of $\log \frac{p_{\epsilon 0}^m(\mathbf{x}_0|\mathbf{x}_\epsilon)}{p_{0\epsilon}(\mathbf{x}_\epsilon|\mathbf{x}_0)}$ is added. With this modification, our NLL differs from the previous NLL of $\mathbb{E}_{\mathbf{x}_0}[-\log p_\epsilon^m(\mathbf{x}_0)]$ by about 0.03-0.06 in VPSDE, see Table 4. We report both previous/corrected ways in Table 4 and report corrected NLL/NELBO as default; see Appendix E for theoretical justification of our NLL/NELBO corrections.

## 7.2 Quantitative Results on Image Generation

**FID Boost with Pre-training** Training INDM from scratch improves NLL with the sacrifice of FID compared to DDPM++ in Table 3. Therefore, we pre-train the score network by DDPM++ as default. This pre-training is intended to search the data nonlinearity near well-trained linear diffusions. Table 3 shows that training INDM after 500k of pre-training steps performs better than DDPM++ on both NLL and FID. Appendix F.3 conducts the ablation study of pre-training steps.

Table 3: Effect of Pre-training.

| Model | NLL | FID |
|---|---|---|
| DDPM++ | 3.03 | 6.70 |
| INDM (w/ pre) | 2.98 | 6.01 |
| INDM (w/o pre) | 2.98 | 8.49 |

**Effect of Flow Training** Table 4 investigates how the flow training affects to performances, under the various linear diffusions and weighting functions. It compares the pre-trained NCSN++/DDPM++ with INDM, of which these pre-trained models initialize the score network of INDM. Experiments in Table 4 presents that INDM improves NELBO in any setting, and minimizes the variational gap to zero if we train the score network with the likelihood weighting.

**SOTA on CelebA** Tables 5 and 6 compare INDM to baseline models. With the emphasis on FID, LSGM is the State-Of-The-Art (SOTA) model on CIFAR-10, but INDM-118M (FID) is on par with LSGM-476M (FID). Moreover, we use Soft Truncation [27] to compare with LSGM (balanced). Soft Truncation softens the smallest diffusion time as a random variable in the training stage to boost sample performance by improving the score accuracy, particularly on large diffusion time. In the inference stage, Soft Truncation uses the fixed smallest diffusion time ($\epsilon$). INDM (NLL) + ST

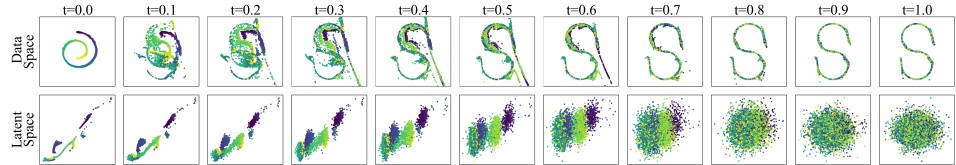

Figure 8: INDM enables to learn a diffusion bridge between two distinctive data distributions.

outperforms LSGM-109M (balanced) in terms of FID with comparable NLL. Also, INDM-121M (NLL) outperforms LSGM-269M (NLL) in FID with identical NLLs. We achieve SOTA FID of 1.75 on CelebA in Table 6. See Appendix F.8 for an extended comparison and Appendix F.9 for samples.

### 7.3   Application Task: Dataset Interpolation

The linear SDEs fixedly perturb data variables, so such SDEs should have the end distribution $p_T(\mathbf{x}_T)$ as an easy-to-compute distribution. With the nonlinear extension, a complex diffusion process exists to transport $p_r^{(1)}$ to another arbitrary $p_r^{(2)}$. However, a common practice of diffusion models constrains only the starting variable by $\mathbf{x}_0^\phi \sim p_r^{(1)}$, so the ending variable of $\mathbf{x}_T^\phi$ is free to deviate from $p_r^{(2)}$. Previous works have tackled this data interpolation task by using a conditional diffusion model [33] or a couple of jointly trained source-and-target diffusion models [34] on paired datasets. Among unconditional diffusion models using unpaired datasets, SBP [15] is a natural approach for the task by imposing bi-constraints with $\mathcal{P}(p_r, \pi)$ replaced by $\mathcal{P}(p_r^{(1)}, p_r^{(2)})$.

INDM can alternatively train the nonlinear diffusion from $p_r^{(1)}$ to $p_r^{(2)}$ with unpaired datasets. We train the score and flow networks with a loss

$$\underbrace{D_{KL}(\boldsymbol{\mu}_\phi \| \boldsymbol{\nu}_{\phi,\theta})}_{\text{INDM NELBO}} + \underbrace{D_{KL}(p_r^{(2)} \| p_\phi)}_{\text{Interpolation Loss}},$$

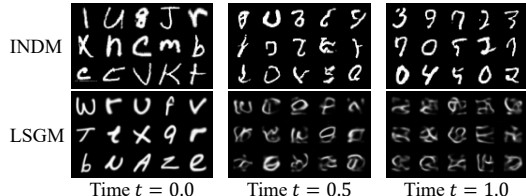

Figure 9: Image-to-image translation from EM-NIST letters dataset to MNIST digits dataset.

where $p_\phi$ is the probability distribution of $\mathbf{x}_T^\phi$, which is calculated by a single feed-forward computation of the flow network, see Algorithm 2 in Appendix F.2.3. The additional interpolation loss forces the diffusion bridge $\{\mathbf{x}_t^\phi\}_{t=0}^T$ to ahead towards $p_r^{(2)}$ by minimizing the KL divergence between $\mathbf{x}_T^\phi \sim p_\phi$ and $p_r^{(2)}$. As the destined variable of the bridge becomes $\mathbf{x}_T^\phi = \mathbf{h}_\phi^{-1}(\mathbf{z}_T) \approx \mathbf{h}_\phi^{-1}(\pi)$, the flow network is what constructs the interpolated bridge between a couple of data variables, see Figures 8 and 9. Particularly, Figure 9 shows that LSGM fails to interpolate a letter to a digit, and we attribute this failure to the non-existence of a diffusion bridge in LSGM. Also, INDM models $p_r^{(1)}$ with $p_{\phi,\theta}$ and $p_r^{(2)}$ with $p_\phi$, so we could compute NLL of each dataset *separately*: 1.10 BPD for MNIST and 1.56 BPD for EMNIST. In contrast, SBP cannot separately estimate densities on each dataset. We emphasize that no additional neural network is needed for the interpolation task with INDM.

## 8   Conclusion

This paper expands the linear diffusion to trainable nonlinear diffusion by combining an invertible transformation and a diffusion model, where the nonlinear diffusion learns the forward diffusion out of variational family of inference measures. A limitation of INDM lies in the training/evaluation time. Potential risk from this work is the negative use of deep generative models, such as deepfake images.

## Acknowledgements

This research was supported by AI Technology Development for Commonsense Extraction, Reasoning, and Inference from Heterogeneous Data(IITP) funded by the Ministry of Science and ICT(2022-0-00077). Also, this work was supported by the National Research Foundation of Korea (NRF) grant funded by the Korea government(MSIT) (NRF-2019R1A5A1028324).

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
