# Supplementary Material of Maximum Likelihood Training of Implicit Nonlinear Diffusion Models

**Dongjun Kim**[*]
KAIST
dongjoun57@kaist.ac.kr

**Byeonghu Na**[*]
KAIST
wp03052@kaist.ac.kr

**Se Jung Kwon**
NAVER CLOVA

**Dongsoo Lee**
NAVER CLOVA

**Wanmo Kang**
KAIST

**Il-Chul Moon**
KAIST / Summary.AI

## Contents

---

[*]Equal contribution

36th Conference on Neural Information Processing Systems (NeurIPS 2022).

## A  Derivations

### A.1  Derivation of Variational Bound for Nonlinear Diffusion

The variational bound derived in Song et al. [11] is only applicable when $\mathbf{G}(\mathbf{x}_t, t)$ is reduced to $g(t)\mathbf{I}$. This section, therefore, derives the variational bound of a general diffusion SDE of $d\mathbf{x}_t = \mathbf{f}(\mathbf{x}_t, t)\, dt + \mathbf{G}(\mathbf{x}_t, t)\, d\mathbf{w}_t$, and we analyze why learning $\mathbf{f}$ and $\mathbf{G}$ is infeasible if 1) the transition probability of $p_{0t}(\mathbf{x}_t|\mathbf{x}_0)$ is intractable and 2) $\mathbf{G}$ is anisotropic by $\mathbf{x}_t$.

From Anderson [10], the corresponding reverse SDE of $d\mathbf{x}_t = \mathbf{f}(\mathbf{x}_t, t)\, dt + \mathbf{G}(\mathbf{x}_t, t)\, d\mathbf{w}_t$ is

$$d\mathbf{x}_t = \left[ \mathbf{f}(\mathbf{x}_t, t) - \operatorname{div}(\mathbf{G}\mathbf{G}^T) - \mathbf{G}\mathbf{G}^T \nabla \log p_t(\mathbf{x}_t) \right] d\bar{t} + \mathbf{G}(\mathbf{x}_t, t)\, d\bar{\mathbf{w}}_t, \tag{4}$$

and the generative SDE becomes

$$d\mathbf{x}_t = \left[ \mathbf{f}(\mathbf{x}_t, t) - \operatorname{div}(\mathbf{G}\mathbf{G}^T) - \mathbf{G}\mathbf{G}^T \mathbf{s}(\mathbf{x}_t, t) \right] d\bar{t} + \mathbf{G}(\mathbf{x}_t, t)\, d\bar{\mathbf{w}}_t. \tag{5}$$

Then, from the Girsanov theorem [35] and the martingale property [17], using the disintegration property of the KL divergence, we have

$$\begin{aligned}
D_{KL}(\boldsymbol{\mu} \| \boldsymbol{\nu}) &= D_{KL}(p_T(\mathbf{x}_T) \| \pi(\mathbf{x}_T)) \\
&+ \frac{1}{2} \int_0^T \mathbb{E}_{\mathbf{x}_t} \left[ \left( \mathbf{s}(\mathbf{x}_t, t) - \nabla \log p_t(\mathbf{x}_t) \right)^T \mathbf{G}\mathbf{G}^T \left( \mathbf{s}(\mathbf{x}_t, t) - \nabla \log p_t(\mathbf{x}_t) \right) \right] dt,
\end{aligned} \tag{6}$$

where $\boldsymbol{\mu}$ is the path measure of Eq. (4) and $\boldsymbol{\nu}$ is the path measure of Eq. (5). Therefore, from the data processing inequality [36], we get

$$\begin{aligned}
D_{KL}(p_r \| p) &\leq D_{KL}(\boldsymbol{\mu} \| \boldsymbol{\nu}) = D_{KL}(p_T(\mathbf{x}_T) \| \pi(\mathbf{x}_T)) \\
&+ \frac{1}{2} \int_0^T \mathbb{E}_{\mathbf{x}_t} \left[ \left( \mathbf{s}(\mathbf{x}_t, t) - \nabla \log p_t(\mathbf{x}_t) \right)^T \mathbf{G}\mathbf{G}^T \left( \mathbf{s}(\mathbf{x}_t, t) - \nabla \log p_t(\mathbf{x}_t) \right) \right] dt,
\end{aligned}$$

where $p$ is the generative distribution at $t = 0$.

Now, from the Fokker-Planck equation, the density function satisfies

$$\frac{\partial p_t}{\partial t} = -\sum_j \frac{\partial}{\partial x_{t,j}} \left[ f_j(\mathbf{x}_t, t) p_t(\mathbf{x}_t) - \sum_i \frac{\partial}{\partial x_{t,j}} (H_{ij}(\mathbf{x}_t, t) p_t(\mathbf{x}_t)) \right],$$

where $\mathbf{H}(\mathbf{x}_t, t) = \frac{1}{2} \mathbf{G}(\mathbf{x}_t, t) \mathbf{G}(\mathbf{x}_t, t)^T$. Then, analogous to Theorem 4 of Song et al. [11], the entropy becomes

$$\begin{aligned}
\mathcal{H}(p_r) - \mathcal{H}(p_T) &= -\int_0^T \frac{\partial}{\partial t} \mathcal{H}(p_t)\, dt \\
&= \int_0^T \int \frac{\partial p_t}{\partial t} \log p_t(\mathbf{x}_t)\, d\mathbf{x}_t\, dt \\
&= -\int_0^T \int \sum_j \frac{\partial}{\partial x_{t,j}} \left[ f_j(\mathbf{x}_t, t) p_t(\mathbf{x}_t) - \sum_i \frac{\partial}{\partial x_{t,i}} (H_{ij}(\mathbf{x}_t, t) p_t(\mathbf{x}_t)) \right] \log p_t(\mathbf{x}_t)\, d\mathbf{x}_t\, dt \\
&= \int_0^T \int \sum_j \left[ f_j(\mathbf{x}_t, t) p_t(\mathbf{x}_t) - \sum_i \frac{\partial}{\partial x_{t,i}} (H_{ij}(\mathbf{x}_t, t) p_t(\mathbf{x}_t)) \right] \frac{\partial \log p_t(\mathbf{x}_t)}{\partial x_{t,j}}\, d\mathbf{x}_t\, dt \\
&= \int_0^T \int p_t(\mathbf{x}_t) \sum_j f_j(\mathbf{x}_t, t) \frac{\partial \log p_t(\mathbf{x}_t)}{\partial x_{t,j}}\, d\mathbf{x}_t\, dt \\
&\quad - \int_0^T \int \sum_j \sum_i \left( \frac{\partial H_{ij}}{\partial x_{t,i}} p_t + H_{ij} \frac{\partial p_t}{\partial x_{t,i}} \right) \frac{\partial \log p_t}{\partial x_{t,j}}\, d\mathbf{x}_t\, dt \\
&= -\int_0^T \mathbb{E}_{\mathbf{x}_t} [\operatorname{div}(\mathbf{f}(\mathbf{x}_t, t))]\, dt \\
&\quad - \int_0^T \mathbb{E}_{\mathbf{x}_t} [\operatorname{div}(\mathbf{H}(\mathbf{x}_t, t))^T \nabla \log p_t(\mathbf{x}_t)] + \mathbb{E}_{\mathbf{x}_t} [\nabla \log p_t(\mathbf{x}_t)^T \mathbf{H}(\mathbf{x}_t, t) \nabla \log p_t(\mathbf{x}_t)]\, dt
\end{aligned}$$

$$= -\frac{1}{2}\int_0^T \mathbb{E}_{\mathbf{x}_t}\Big[2\mathrm{div}(\mathbf{f}(\mathbf{x}_t,t)) + \mathrm{div}(\mathbf{G}(\mathbf{x}_t,t)\mathbf{G}(\mathbf{x}_t,t)^T)^T\nabla\log p_t(\mathbf{x}_t)$$
$$+\nabla\log p_t(\mathbf{x}_t)^T\mathbf{G}(\mathbf{x}_t,t)\mathbf{G}(\mathbf{x}_t,t)^T\nabla\log p_t(\mathbf{x}_t)\Big]\,dt.$$

Therefore, the variational bound of the model log-likelihood is derived by

$$\mathbb{E}_{p_r(\mathbf{x}_0)}\big[-\log p(\mathbf{x}_0)\big] = D_{KL}(p_r\|p) + \mathcal{H}(p_r) \le D_{KL}(\boldsymbol{\mu}\|\boldsymbol{\nu}) + \mathcal{H}(p_r)$$
$$= \frac{1}{2}\int_0^T \mathbb{E}_{\mathbf{x}_t}\big[(\nabla\log p_t(\mathbf{x}_t) - \mathbf{s}(\mathbf{x}_t,t))^T\mathbf{G}\mathbf{G}^T(\nabla\log p_t(\mathbf{x}_t) - \mathbf{s}(\mathbf{x}_t,t))\big]\,dt$$
$$+\mathbb{E}_{\mathbf{x}_T}\big[-\log\pi(\mathbf{x}_T)\big] + \mathcal{H}(p_r) - \mathcal{H}(p_T)$$
$$= \frac{1}{2}\int_0^T \mathbb{E}_{\mathbf{x}_t}\Big[\big(\mathbf{s}(\mathbf{x}_t,t) - \nabla\log p_t(\mathbf{x}_t)\big)^T\mathbf{G}\mathbf{G}^T\big(\mathbf{s}(\mathbf{x}_t,t) - \nabla\log p_t(\mathbf{x}_t)\big)$$
$$-\nabla\log p_t(\mathbf{x}_t)^T\mathbf{G}\mathbf{G}^T\nabla\log p_t(\mathbf{x}_t) - \mathrm{div}(\mathbf{G}\mathbf{G}^T)^T\nabla\log p_t(\mathbf{x}_t) - 2\mathrm{div}(\mathbf{f}(\mathbf{x}_t,t))\Big]\,dt$$
$$+\mathbb{E}_{\mathbf{x}_T}\big[-\log\pi(\mathbf{x}_T)\big].$$

Using the integration by parts, this variational bound transforms to

$$\mathbb{E}_{p_r(\mathbf{x}_0)}\big[-\log p_{\boldsymbol{\theta}}(\mathbf{x}_0)\big] \le \frac{1}{2}\int_0^T \mathbb{E}_{\mathbf{x}_t}\Big[\mathbf{s}^T\mathbf{G}\mathbf{G}^T\mathbf{s} + 2\mathrm{div}(\mathbf{G}\mathbf{G}^T\mathbf{s})$$
$$-\mathrm{div}(\mathbf{G}\mathbf{G}^T)\nabla\log p_t - 2\mathrm{div}(\mathbf{f})\Big]\,dt + \mathbb{E}_{\mathbf{x}_T}\big[-\log\pi(\mathbf{x}_T)\big]$$

Also, this variational bound is equivalently formulated as

$$\mathbb{E}_{p_r(\mathbf{x}_0)}\big[-\log p(\mathbf{x}_0)\big]$$
$$\le \frac{1}{2}\int_0^T \mathbb{E}_{\mathbf{x}_0,\mathbf{x}_t}\Big[\big(\mathbf{s}(\mathbf{x}_t,t) - \nabla\log p_{0t}(\mathbf{x}_t|\mathbf{x}_0)\big)^T\mathbf{G}\mathbf{G}^T\big(\mathbf{s}(\mathbf{x}_t,t) - \nabla\log p_{0t}(\mathbf{x}_t|\mathbf{x}_0)\big)$$
$$-\nabla\log p_{0t}(\mathbf{x}_t|\mathbf{x}_0)^T\mathbf{G}\mathbf{G}^T\nabla\log p_{0t}(\mathbf{x}_t|\mathbf{x}_0) - \mathrm{div}(\mathbf{G}\mathbf{G}^T)^T\nabla\log p_{0t}(\mathbf{x}_t|\mathbf{x}_0) - 2\mathrm{div}(\mathbf{f})\Big]\,dt$$
$$+\mathbb{E}_{\mathbf{x}_T}\big[-\log\pi(\mathbf{x}_T)\big].$$

Therefore, optimizing the nonlinear drift ($\mathbf{f}$) and diffusion ($\mathbf{G}$) terms are intractable in general for two reasons. First, the transition probability of $p_{0t}(\mathbf{x}_t|\mathbf{x}_0)$ is intractable for nonlinear SDEs. To compute $p_{0t}(\mathbf{x}_t|\mathbf{x}_0)$, one needs the Feynman-Kac formula [12] which requires expectation on every sample paths, see Appendix E.4.

Second, even if $p_{0t}(\mathbf{x}_t|\mathbf{x}_0)$ is tractable, computing the above variational bound would not be scalable due to the matrix multiplication of $\mathbf{G}\mathbf{G}^T$ that is of $O(d^2)$ complexity and the divergence computation [37]. These would become the main source of training bottleneck if dimension increases.

### A.2 Derivation of Nonlinear Drift and Volatility Terms for INDM

Throughout this section, we omit $\phi$ for notational simplicity. With the linear SDE on latent space

$$d\mathbf{z}_t = -\frac{1}{2}\beta(t)\mathbf{z}_t\,dt + g(t)\,d\mathbf{w}_t, \quad \mathbf{z}_0 = \mathbf{h}(\mathbf{x}_0) \text{ with } \mathbf{x}_0 \sim p_r, \tag{7}$$

from $\mathbf{x}_t = \mathbf{h}^{-1}(\mathbf{z}_t)$, the $k$-th component of the induced variable satisfies

$$d\mathbf{x}_{t,k} = \frac{\partial \mathrm{h}_k^{-1}}{\partial t}\,dt + \big[\nabla_{\mathbf{z}_t}\mathrm{h}_k^{-1}(\mathbf{z}_t)\big]^T\,d\mathbf{z}_t + \frac{1}{2}\mathrm{tr}\big(\nabla_{\mathbf{z}_t}^2\mathrm{h}_k^{-1}(\mathbf{z}_t)\,d\mathbf{z}_t\,d\mathbf{z}_t^T\big) \tag{8}$$

by the multivariate Ito's Lemma [17]. Plugging the linear SDE of Eq. (7), Eq. (8) is transformed to

$$d\mathbf{x}_{t,k} = \big[\nabla_{\mathbf{z}_t}\mathrm{h}_k^{-1}(\mathbf{z}_t)\big]^T\Big\{-\frac{1}{2}\beta(t)\mathbf{z}_t\,dt + g(t)\,d\mathbf{w}_t\Big\} + \frac{1}{2}g^2(t)\mathrm{tr}\big(\nabla_{\mathbf{z}_t}^2\mathrm{h}_k^{-1}(\mathbf{z}_t)\big)\,dt$$
$$= \Big\{-\frac{1}{2}\beta(t)\big[\nabla_{\mathbf{z}_t}\mathrm{h}_k^{-1}(\mathbf{z}_t)\big]^T\mathbf{z}_t + \frac{1}{2}g^2(t)\mathrm{tr}\big(\nabla_{\mathbf{z}_t}^2\mathrm{h}_k^{-1}(\mathbf{z}_t)\big)\Big\}\,dt + g(t)\big[\nabla_{\mathbf{z}_t}\mathrm{h}_k^{-1}(\mathbf{z}_t)\big]^T\,d\mathbf{w}_t, \tag{9}$$

because $\frac{\partial \mathrm{h}_k^{-1}}{\partial t} = 0$. Then, Eq. (9) in vector form becomes

$$\mathrm{d}\mathbf{x}_t = \mathbf{f}(\mathbf{x}_t, t)\,\mathrm{d}t + \mathbf{G}(\mathbf{x}_t, t)\,\mathrm{d}\mathbf{w}_t,$$

where the vector-valued drift and the matrix-valued volatility terms are given by

$$\begin{cases} \mathbf{f}(\mathbf{x}_t, t) = -\frac{1}{2}\beta(t)\nabla_{\mathbf{z}_t}\mathbf{h}^{-1}(\mathbf{z}_t)\mathbf{z}_t + \frac{1}{2}g^2(t)\mathrm{tr}\left(\nabla_{\mathbf{z}_t}^2\mathbf{h}^{-1}(\mathbf{z}_t)\right) \\ \mathbf{G}(\mathbf{x}_t, t) = g(t)\nabla_{\mathbf{z}_t}\mathbf{h}^{-1}(\mathbf{z}_t). \end{cases} \tag{10}$$

Here, $\nabla_{\mathbf{z}_t}^2\mathbf{h}^{-1}(\mathbf{z}_t)$ is a 3-dimensional tensor with $(i,j,k)$-th element to be $\left(\nabla_{\mathbf{z}_t}^2\mathrm{h}_k^{-1}(\mathbf{z}_t)\right)_{i,j}$, and the trace operator applied on this tensor is defined as a vector of $\left[\mathrm{tr}\left(\nabla_{\mathbf{z}_t}^2\mathrm{h}_1^{-1}(\mathbf{z}_t)\right), ..., \mathrm{tr}\left(\nabla_{\mathbf{z}_t}^2\mathrm{h}_d^{-1}(\mathbf{z}_t)\right)\right]^T$. From the inverse function theorem [38], the Jacobian of the inverse function $\nabla_{\mathbf{z}_t}\mathbf{h}^{-1}(\mathbf{z}_t)$ equals to the inverse Jacobian $\left[\nabla_{\mathbf{x}_t}\mathbf{h}(\mathbf{x}_t)\right]^{-1}$. Therefore, Eq. (10) is transformed to

$$\begin{cases} \mathbf{f}(\mathbf{x}_t, t) = -\frac{1}{2}\beta(t)\left[\nabla_{\mathbf{x}_t}\mathbf{h}(\mathbf{x}_t)\right]^{-1}\mathbf{h}(\mathbf{x}_t) + \frac{1}{2}g^2(t)\mathrm{tr}\left(\nabla_{\mathbf{z}_t}^2\mathbf{h}^{-1}(\mathbf{z}_t)\right) \\ \mathbf{G}(\mathbf{x}_t, t) = g(t)\left[\nabla_{\mathbf{x}_t}\mathbf{h}(\mathbf{x}_t)\right]^{-1}. \end{cases} \tag{11}$$

Now, we derive the second term of $\mathbf{f}$ in terms of $\mathbf{x}_t$ as follows: observe that $\sum_k \frac{\partial \mathrm{h}_i^{-1}}{\partial \mathrm{z}_{t,k}}\frac{\partial \mathrm{h}_k}{\partial \mathrm{x}_{t,j}} = \delta_{i,j}$, where $\delta_{i,j} = 1$ if $i = j$ and 0 otherwise. Differentiating both sides with respect to $\mathrm{x}_{t,l}$, we have

$$\sum_k \left\{ \frac{\partial}{\partial \mathrm{x}_{t,l}}\left(\frac{\partial \mathrm{h}_i^{-1}}{\partial \mathrm{z}_{t,k}}\right)\right\}\frac{\partial \mathrm{h}_k}{\partial \mathrm{x}_{t,j}} + \frac{\partial \mathrm{h}_i^{-1}}{\partial \mathrm{z}_{t,k}}\left\{\frac{\partial}{\partial \mathrm{x}_{t,l}}\left(\frac{\partial \mathrm{h}_k}{\partial \mathrm{x}_{t,j}}\right)\right\} = 0,$$

where the first term is

$$\sum_{k,m} \frac{\partial \mathrm{h}_m}{\partial \mathrm{x}_{t,l}}\left\{\frac{\partial}{\partial \mathrm{z}_{t,m}}\left(\frac{\partial \mathrm{h}_i^{-1}}{\partial \mathrm{z}_{t,k}}\right)\right\}\frac{\partial \mathrm{h}_k}{\partial \mathrm{x}_{t,j}} = \sum_{k,m}\left(\nabla_{\mathbf{x}_t}\mathbf{h}(\mathbf{x}_t)\right)_{l,m}^T\left(\nabla_{\mathbf{z}_t}^2\mathrm{h}_i^{-1}(\mathbf{z}_t)\right)_{m,k}\left(\nabla_{\mathbf{x}_t}\mathbf{h}(\mathbf{x}_t)\right)_{k,j},$$

using the chain rule, and the second term becomes

$$\sum_k \frac{\partial \mathrm{h}_i^{-1}}{\partial \mathrm{z}_{t,k}}\left\{\frac{\partial}{\partial \mathrm{x}_{t,l}}\left(\frac{\partial \mathrm{h}_k}{\partial \mathrm{x}_{t,j}}\right)\right\} = \sum_k \left(\nabla_{\mathbf{z}_t}\mathbf{h}^{-1}(\mathbf{z}_t)\right)_{i,k}\left(\nabla_{\mathbf{x}_t}^2\mathrm{h}_k(\mathbf{x}_t)\right)_{l,j}.$$

From the above, we derive the trace term of $\mathbf{f}$ in Eq. (11) as

$$\mathrm{tr}\left(\nabla_{\mathbf{z}_t}^2\mathbf{h}^{-1}(\mathbf{z}_t)\right) = -\mathrm{tr}\left(\left[\nabla_{\mathbf{x}_t}\mathbf{h}(\mathbf{x}_t)\right]^{-T}\left(\left[\nabla_{\mathbf{x}_t}\mathbf{h}(\mathbf{x}_t)\right]^{-1} * \nabla_{\mathbf{x}_t}^2\mathbf{h}(\mathbf{x}_t)\right)\left[\nabla_{\mathbf{x}_t}\mathbf{h}(\mathbf{x}_t)\right]^{-1}\right),$$

where $\nabla_{\mathbf{x}_t}^2\mathbf{h}(\mathbf{x}_t)$ is a 3-dimensional tensor with $(i,j,k)$-th element to be $\left(\nabla_{\mathbf{x}_t}^2\mathrm{h}_k(\mathbf{x}_t)\right)_{i,j}$. Also, we define $*$ operation as the element-wise matrix multiplication given by

$$\left(\left[\nabla_{\mathbf{x}_t}\mathbf{h}(\mathbf{x}_t)\right]^{-1} * \nabla_{\mathbf{x}_t}^2\mathbf{h}(\mathbf{x}_t)\right)_{i,j} := \nabla_{\mathbf{x}_t}\left[\mathbf{h}(\mathbf{x}_t)\right]^{-1}\left(\nabla_{\mathbf{x}_t}^2\mathbf{h}(\mathbf{x}_t)\right)_{i,j}.$$

Combining all together, thus, we derive the nonlinear drift term in Eq. (11) as a function of $\mathbf{x}_t$ given by

$$\mathbf{f}(\mathbf{x}_t, t) = -\frac{1}{2}\beta(t)\left[\nabla_{\mathbf{x}_t}\mathbf{h}(\mathbf{x}_t)\right]^{-1}\mathbf{h}(\mathbf{x}_t)$$
$$-\frac{1}{2}g^2(t)\mathrm{tr}\left(\left[\nabla_{\mathbf{x}_t}\mathbf{h}(\mathbf{x}_t)\right]^{-T}\left(\left[\nabla_{\mathbf{x}_t}\mathbf{h}(\mathbf{x}_t)\right]^{-1} * \nabla_{\mathbf{x}_t}^2\mathbf{h}(\mathbf{x}_t)\right)\left[\nabla_{\mathbf{x}_t}\mathbf{h}(\mathbf{x}_t)\right]^{-1}\right).$$

# B   Details on Section 6.1

It is one of central topics in the community of VAE to obtain a tighter ELBO [39, 40]. This section analyzes the variational gap and further theoretical analysis in diffusion models. Before we start, we remind the generalized Helmholtz decomposition in Lemma 1.

**Lemma 1** (Helmholtz Decomposition [41]). *Any twice continuously differentiable vector field* $\mathbf{s}$ *that decays faster than* $\|\mathbf{z}\|_2^{-c}$ *for* $\|\mathbf{z}\|_2 \to \infty$ *and* $c > 0$ *can be uniquely decomposed into two vector fields, one rotation-free and one divergence-free:* $\mathbf{s} = \nabla \log p + \mathbf{u}$.

A rotation-free vector field $\nabla \log p$, or the divergence part, is a score function of a probability density $p$, and a divergence-free vector field $\mathbf{u}$, or the rotation part, satisfies $\text{div}(\mathbf{u}) \equiv 0$. From this decomposition, any score network is decomposed by $\mathbf{s}_{\boldsymbol{\theta}}(\mathbf{z}_t, t) = \nabla \log p_t^{\boldsymbol{\theta}}(\mathbf{z}_t) + \mathbf{u}_t^{\boldsymbol{\theta}}(\mathbf{z}_t)$ for some probability $p_t^{\boldsymbol{\theta}}$ and vector field $\mathbf{u}_t^{\boldsymbol{\theta}}$, for any $t \in (0, T]$. Then, the generative SDE of the full score network

$$d\mathbf{z}_t = \left[ \mathbf{f}(\mathbf{z}_t, t) - g^2(t) \mathbf{s}_{\boldsymbol{\theta}}(\mathbf{z}_t, t) \right] d\bar{t} + g(t) \, d\bar{\mathbf{w}}_t \tag{12}$$

and the generative SDE of the divergence part

$$d\mathbf{z}_t = \left[ \mathbf{f}(\mathbf{z}_t, t) - g^2(t) \nabla \log p_t^{\boldsymbol{\theta}}(\mathbf{z}_t) \right] d\bar{t} + g(t) \, d\bar{\mathbf{w}}_t \tag{13}$$

has distinctive path measures. Throughout this section, $\mathbf{f}(\mathbf{z}_t, t)$ does not have to be a linear vector field, such as $-\frac{1}{2}\beta(t)\mathbf{z}_t$. If $\boldsymbol{\nu}_{\boldsymbol{\theta}}$ and $\boldsymbol{\rho}_{\boldsymbol{\theta}}$ are the path measures of SDEs of Eqs. (12) and (13), respectively, then using the Girsanov theorem [11, 35] and the martingale property [17], we have

$$\begin{aligned} D_{KL}(\boldsymbol{\nu}_{\boldsymbol{\theta}} \| \boldsymbol{\rho}_{\boldsymbol{\theta}}) =& \frac{1}{2} \int_0^T g^2(t) \mathbb{E}_{\mathbf{z}_t \sim \boldsymbol{\nu}_{\boldsymbol{\theta}}|_t} \left[ \| \nabla \log p_t^{\boldsymbol{\theta}}(\mathbf{z}_t) - \mathbf{s}_{\boldsymbol{\theta}}(\mathbf{z}_t, t) \|_2^2 \right] dt \\ =& \frac{1}{2} \int_0^T g^2(t) \mathbb{E}_{\mathbf{z}_t \sim \boldsymbol{\nu}_{\boldsymbol{\theta}}|_t} \left[ \| \mathbf{u}_{\boldsymbol{\theta}}(\mathbf{z}_t, t) \|_2^2 \right] dt. \end{aligned} \tag{14}$$

This KL divergence of two path measures quantifies how much the score network contains the rotation part $\mathbf{u}_t^{\boldsymbol{\theta}}$. Recall that the forward SDE satisfies

$$d\mathbf{z}_t = \mathbf{f}(\mathbf{z}_t, t) \, dt + g(t) \, d\mathbf{w}_t,$$

which starts at $p_0^{\boldsymbol{\phi}}$, and the marginal distribution of its path measure $\boldsymbol{\mu}_{\boldsymbol{\phi}}$ at $t$ is $p_t^{\boldsymbol{\phi}}$. As NELBO is equivalent to

$$D_{KL}(\boldsymbol{\mu}_{\boldsymbol{\phi}} \| \boldsymbol{\nu}_{\boldsymbol{\phi},\boldsymbol{\theta}}) = \frac{1}{2} \int_0^T g^2(t) \mathbb{E}_{p_t^{\boldsymbol{\phi}}(\mathbf{z}_t)} \left[ \| \mathbf{s}_{\boldsymbol{\theta}}(\mathbf{z}_t, t) - \nabla \log p_t^{\boldsymbol{\phi}}(\mathbf{z}_t) \|_2^2 \right] dt + D_{KL}\left( p_T^{\boldsymbol{\phi}}(\mathbf{z}_T) \| \pi(\mathbf{z}_T) \right), \tag{15}$$

for $\boldsymbol{\theta}$-optimization, the optimal $\boldsymbol{\theta}^*$ satisfies $\mathbf{s}_{\boldsymbol{\theta}^*}(\mathbf{z}_t, t) = \nabla \log p_t^{\boldsymbol{\phi}}(\mathbf{z}_t)$. At this optimality, $\boldsymbol{\theta}$ should satisfy a pair of constraints: 1) the zero-rotation part $\mathbf{u}_t^{\boldsymbol{\theta}^*} \equiv 0$, which is equivalent to $D_{KL}(\boldsymbol{\nu}_{\boldsymbol{\theta}^*} \| \boldsymbol{\rho}_{\boldsymbol{\theta}^*}) = 0$; 2) the coincidence of $\nabla \log p_t^{\boldsymbol{\phi}}(\mathbf{z}_t) \equiv \nabla \log p_t^{\boldsymbol{\theta}}(\mathbf{z}_t)$. The starting point to analyze the variational gap with respect to the Helmholtz decomposition is the next theorem. We defer the proofs in Section G.

**Proposition 1.** *Suppose $q_t^{\boldsymbol{\theta}}$ is the marginal distribution of $\boldsymbol{\nu}_{\boldsymbol{\theta}}$ at t. The variational gap is*

$$\begin{aligned} \text{Gap}\left( \boldsymbol{\mu}_{\boldsymbol{\phi}}(\{\mathbf{x}_t\}), \boldsymbol{\nu}_{\boldsymbol{\phi},\boldsymbol{\theta}}(\{\mathbf{x}_t\}) \right) :=& D_{KL}\left( \boldsymbol{\mu}_{\boldsymbol{\phi}}(\{\mathbf{x}_t\}) \| \boldsymbol{\nu}_{\boldsymbol{\phi},\boldsymbol{\theta}}(\{\mathbf{x}_t\}) \right) - D_{KL}\left( p_0^{\boldsymbol{\phi}}(\mathbf{x}_0) \| q_0^{\boldsymbol{\theta}}(\mathbf{x}_0) \right) \\ =& \frac{1}{2} \int_0^T g^2(t) \mathbb{E}_{p_t^{\boldsymbol{\phi}}(\mathbf{z}_t)} \big[ \underbrace{\| \nabla \log q_t^{\boldsymbol{\theta}}(\mathbf{z}_t) - \mathbf{s}_{\boldsymbol{\theta}}(\mathbf{z}_t, t) \|_2^2}_{\text{Score-only error}} \big] dt. \end{aligned}$$

*Remark* 1. The generative SDE of $d\mathbf{z}_t^{\boldsymbol{\theta}} = \left[ -\frac{1}{2}\beta(t)\mathbf{z}_t^{\boldsymbol{\theta}} - g^2(t)\mathbf{s}_{\boldsymbol{\theta}}(\mathbf{z}_t^{\boldsymbol{\theta}}, t) \right] d\bar{t} + g(t) \, d\bar{\mathbf{w}}_t$ does not necessarily start from the prior $\pi$. Proposition 1 holds for an arbitrary distribution $p_T^{\boldsymbol{\theta}}$. At the same spirit, Proposition 1 holds for any distribution $p_0^{\boldsymbol{\phi}}$.

*Remark* 2. Throughout the section, we follow the assumptions made in Appendix A of Song et al. [11]. On top of that, we assume that both $\mathbf{s}_{\boldsymbol{\theta}}$ and $q_t^{\boldsymbol{\theta}}$ are continuously differentiable.

The variational gap derived in Proposition 1 does not include the forward score, $\nabla \log p_t^{\boldsymbol{\phi}}(\mathbf{z}_t)$, except for taking the expectation, $\mathbb{E}_{p_t^{\boldsymbol{\phi}}(\mathbf{z}_t)}$. Therefore, the gap is intuitively connected to the score training, rather than the flow training. To elucidate the logic, we decompose the variational gap in Proposition 1 into

$$\text{Gap}(\boldsymbol{\mu}_{\boldsymbol{\phi}}, \boldsymbol{\nu}_{\boldsymbol{\phi},\boldsymbol{\theta}}) = \frac{1}{2} \int_0^T g^2(t) \mathbb{E}_{p_t^{\boldsymbol{\phi}}(\mathbf{z}_t)} \big[ \| \nabla \log q_t^{\boldsymbol{\theta}}(\mathbf{z}_t) - \mathbf{s}_{\boldsymbol{\theta}}(\mathbf{z}_t, t) \|_2^2 \big] dt$$

$$(16) \quad \leq \frac{1}{2} \int_0^T g^2(t) \mathbb{E}_{p_t^{\boldsymbol{\phi}}(\mathbf{z}_t)} \left[ \left\| \nabla \log q_t^{\boldsymbol{\theta}}(\mathbf{z}_t) - \nabla \log p_t^{\boldsymbol{\theta}}(\mathbf{z}_t) \right\|_2^2 + \left\| \nabla \log p_t^{\boldsymbol{\theta}}(\mathbf{z}_t) - \mathbf{s}_{\boldsymbol{\theta}}(\mathbf{z}_t, t) \right\|_2^2 \right] dt.$$

$$= \frac{1}{2} \int_0^T g^2(t) \mathbb{E}_{p_t^\phi(\mathbf{z}_t)} \Big[ \underbrace{\|\nabla \log q_t^\theta(\mathbf{z}_t) - \nabla \log p_t^\theta(\mathbf{z}_t)\|_2^2}_{\text{How close is } \boldsymbol{\nu}_\theta \text{ to forward measure}} + \underbrace{\|\mathbf{u}_t^\theta(\mathbf{z}_t)\|_2^2}_{\text{How close is } \mathbf{u}_t^\theta \text{ to zero}} \Big] \, dt.$$

The second term, $\|\nabla \log p_t^\theta(\mathbf{z}_t) - \mathbf{s}_\theta(\mathbf{z}_t, t)\|_2^2$ (see Eq. (14)), equals to the $L^2$-norm of the rotation term, $\|\mathbf{u}_t^\theta(\mathbf{z}_t, t)\|_2^2$, so it measures how close is the score network to the space of

$$\mathbf{S}_{div} := \{\mathbf{s} : \mathbb{R}^d \to \mathbb{R}^d | \text{ the rotation term of } \mathbf{s} \text{ is zero}\}.$$

On the other hand, having assumed the rotation part to be zero, the first term, $\|\nabla \log q_t^\theta(\mathbf{z}_t) - \nabla \log p_t^\theta(\mathbf{z}_t)\|_2^2$, becomes zero only if the generative score $\nabla \log q_t^\theta$ equals to a forward score $\nabla \log q_t$ with certain initial distribution $q_0$, meaning that if there exists a $q_0$ and $q_t$ is a marginal density of the forward SDE starting from $q_0$, then

$$\nabla \log q_t(\mathbf{z}_t) = \nabla \log q_t^\theta(\mathbf{z}_t) \text{ is equivalent to } \nabla \log q_t(\mathbf{z}_t) = \nabla \log p_t^\theta(\mathbf{z}_t),$$

and only in that case, $\nabla \log q_t^\theta(\mathbf{z}_t) = \nabla \log p_t^\theta(\mathbf{z}_t)$. Therefore, the gap becomes tight if 1) $\mathbf{u}_t^\theta \equiv 0$ and 2) $\nabla \log q_t^\theta = \nabla \log q_t$ for some $q_t$ following the forward SDE, which is concretely proved in Lemma 2. It turns out that this is the only case of the gap being zero proved in Theorem 2. For that, we provide a rigorous definition of the class of score functions of interest as below.

**Definition 1.** Let $\mathbf{S}_{sol} \subseteq \mathbf{S}_{div}$ be a sub-family of rotation-free score functions $\mathbf{s} : \mathbb{R}^d \to \mathbb{R}^d$ such that $\mathbf{s}(\mathbf{z}_t, t) = \nabla \log p_t(\mathbf{z}_t)$ almost everywhere for $p_t$ that is the marginal distribution of the path measure of $d\mathbf{z}_t = \mathbf{f}(\mathbf{z}_t, t) \, dt + g(t) \, d\mathbf{w}_t$ at $t$.

*Remark* 3. Analogous to Theorem 1, no condition for the starting and ending variables is imposed in Definition 1.

*Remark* 4. $\mathbf{S}_{sol}$ is the space of score functions of the forward SDE $d\mathbf{z}_t = \mathbf{f}(\mathbf{z}_t, t) \, dt + g(t) \, d\mathbf{w}_t$ with arbitrary initial variable.

Although Song et al. [11] focused on the data diffusion, their theory is applicable for a diffusion process that starts with an arbitrary initial distribution. Lemma 2 describes the theoretic analysis done by Song et al. [11].

**Lemma 2** (Theorem 2 of Song et al. [11])**.** $\text{Gap}(\boldsymbol{\mu}_\phi, \boldsymbol{\nu}_{\phi,\theta}) = 0$ if $\mathbf{s}_\theta \in \mathbf{S}_{sol}$.

With Lemma 2, however, we cannot certainly be sure that the score network $\mathbf{s}_\theta$ of INDM falls to $\mathbf{S}_{sol}$ when the variational gap is zero. Thus, we take a step further to identify the connection of zero variational gap and the class of rotation-free score functions $\mathbf{S}_{sol}$ in Theorem 2. This Theorem 2 completely characterizes all admissible score networks that achieve the zero variational gaps, and we are certain that the zero variational gap implies $\mathbf{s}_\theta \in \mathbf{S}_{sol}$, which turns out to be a solution space in Theorem 3.

**Theorem 2.** $\text{Gap}(\boldsymbol{\mu}_\phi, \boldsymbol{\nu}_{\phi,\theta}) = 0$ *if and only if* $\mathbf{s}_\theta \in \mathbf{S}_{sol}$.

From Theorem 2, the variational gap is strictly positive as long as the rotation part of the score network remains to be nonzero. NELBO of Eq. (15) optimizes its score network towards $\mathbf{s}_\theta(\mathbf{z}_t, t) \to \nabla \log p_t^\phi(\mathbf{z}_t) := \mathbf{s}_\phi(\mathbf{z}_t, t)$, which is equivalent to $\log p_t^\theta(\mathbf{z}_t) \to \nabla \log p_t^\phi(\mathbf{z}_t)$ (or equivalently, $\log q_t^\theta(\mathbf{z}_t) \to \nabla \log p_t^\phi(\mathbf{z}_t)$) and $\mathbf{u}_t^\theta(\mathbf{z}_t, t) \to 0$. In contrast to DDPM++ with fixed $\phi$, optimizing

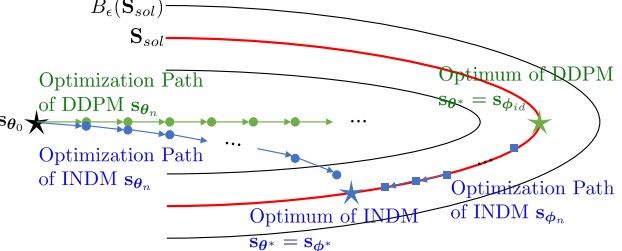

Figure 10: Descriptive Illustration On Nearly MLE Training.

$D_{KL}(\boldsymbol{\mu}_\phi \| \boldsymbol{\nu}_{\phi,\theta})$ w.r.t $\phi$ finds the closest $\mathbf{s}_\phi$ among $\mathbf{S}_{sol}$ to $\mathbf{s}_\theta$. Thus, if $\mathbf{s}_\theta \in \mathbf{S}_{sol}$, then $\mathbf{s}_{\phi^*} = \mathbf{s}_\theta$, which is proved in Theorem 3. If $\mathbf{s}_\theta \notin \mathbf{S}_{sol}$, then $\mathbf{s}_{\phi^*}$ is not equal to $\mathbf{s}_\theta$, anymore, but $\mathbf{s}_{\phi^*}$ will be the closest among $\mathbf{S}_{sol}$ to $\mathbf{s}_\theta$ because $D_{KL}(\boldsymbol{\mu}_\phi \| \boldsymbol{\nu}_{\phi,\theta})$ is the weighted $L^2$-norm of $\mathbf{s}_\phi - \mathbf{s}_\theta$.

**Theorem 3.** *For any fixed* $\mathbf{s}_{\bar{\theta}} \in \mathbf{S}_{sol}$, *if* $\phi^* \in \arg\min_\phi D_{KL}(\boldsymbol{\mu}_\phi \| \boldsymbol{\nu}_{\phi,\bar{\theta}})$, *then* $\mathbf{s}_{\phi^*}(\mathbf{z}_t, t) = \nabla \log p_t^{\phi^*}(\mathbf{z}_t) = \mathbf{s}_{\bar{\theta}}(\mathbf{z}_t, t)$, *and* $D_{KL}(\boldsymbol{\mu}_{\phi^*} \| \boldsymbol{\nu}_{\phi^*, \bar{\theta}}) = D_{KL}(p_r \| p_{\phi^*, \bar{\theta}}) = \text{Gap}(\boldsymbol{\mu}_{\phi^*}, \boldsymbol{\nu}_{\phi^*, \bar{\theta}}) = 0$.

Indeed, Theorem 3 implies that the whole class of $\mathbf{S}_{sol}$ is the solution space, which means that any $\mathbf{s}_{\boldsymbol{\theta}}$ in $\mathbf{S}_{sol}$ is a candidate for an optimal score function as there always exists $\boldsymbol{\phi}^*$ corresponding to a given $\boldsymbol{\theta}$ that achieves the perfect match of the model distribution to the data distribution. This is contrastive to DDPM++ that only has a unique optimal point of $\mathbf{s}_{\boldsymbol{\theta}^*}(\mathbf{z}_t, t) = \nabla \log p_t^{\boldsymbol{\phi}_{id}}(\mathbf{z}_t) \in \mathbf{S}_{sol}$. Figure 10 illustrates that the optimal point of DDPM is a single point in $\mathbf{S}_{sol}$, whereas any $\mathbf{s}_{\boldsymbol{\theta}} \in \mathbf{S}_{sol}$ is a candidate for the optimal point of INDM by Theorem 3. In other words, the number of DDPM optimality is one, while INDM has infinite number of optimalities.

## B.1 Restricting Search Space of $\mathbf{s}_{\boldsymbol{\theta}}$ into $\mathbf{S}_{div}$

Due to the space limit, the argument in this section has not been included in the main paper. Below, we provide the rationale that it is the number of optimal points that affect the NLL performance. For that, we optimize DDPM++ with a regularization, suggested in Proposition 5. This regularization restricts the score network from not deviating $\mathbf{S}_{div}$ too far by keeping the rotation term, $\mathbf{u}_t^{\boldsymbol{\theta}}$, being consistently small. Consequently, a fastly converging rotation term is advantageous in reducing the variational gap (see Inequality (16)), and this regularization helps the MLE training of DDPM++.

Proposition 2 proves that $\mathbf{S}_{div}$ is identical to a class of score functions that have symmetric derivatives. From this, Proposition 3 provides a motivation of the regularization by proving that a symmetric matrix satisfies a certain equality. Then, Proposition 4 implies that the formula suggested in Proposition 3 indeed measures how close is the matrix symmetric. Lastly, Proposition 5 provides the minimum variance estimator of the formula. With these propositions, we conclude that the constraint of

$$\mathbb{E}_{\boldsymbol{\epsilon}_1, \boldsymbol{\epsilon}_2}\left[\left(\boldsymbol{\epsilon}_2^T\left(\nabla \mathbf{s}_{\boldsymbol{\theta}}(\mathbf{z}_t, t) - (\nabla \mathbf{s}_{\boldsymbol{\theta}})^T(\mathbf{z}_t, t)\right)\boldsymbol{\epsilon}_1\right)^2\right] = 0 \tag{17}$$

with $\boldsymbol{\epsilon}_1$ and $\boldsymbol{\epsilon}_2$ sampled from the random variable suggested in Proposition 5 would optimize $\mathbf{s}_{\boldsymbol{\theta}}$ in the space of $\mathbf{S}_{div}$. Using the Lagrangian form, we could add the left-hand-side of Eq. (17) as a regularization term in NELBO to force the score network not deviate from $\mathbf{S}_{div}$ too much.

With the clear mathematical properties, however, obtaining the full matrix of $\nabla \mathbf{s}_{\boldsymbol{\theta}}$ is a bottleneck in the computation of the regularization term. Specifically, each row of $\nabla \mathbf{s}_{\boldsymbol{\theta}}$ needs to be computed separately [42], so it takes $O(d)$ complexity to compute $\nabla \mathbf{s}_{\boldsymbol{\theta}}$, which is prohibitively expensive. Therefore, we use a trick to reduce $O(d)$ to $O(1)$ motivated from the Hutchinson's estimator [43, 23]: first, we compute the gradient of $\boldsymbol{\epsilon}_2^T \mathbf{s}_{\boldsymbol{\theta}}$ and $\boldsymbol{\epsilon}_1^T \mathbf{s}_{\boldsymbol{\theta}}$, separately. Afterwards, we apply vector multiplication between $\boldsymbol{\epsilon}_1$ and $\nabla(\boldsymbol{\epsilon}_2^T \mathbf{s}_{\boldsymbol{\theta}})$, which gives us $\boldsymbol{\epsilon}_2^T \nabla \mathbf{s}_{\boldsymbol{\theta}} \boldsymbol{\epsilon}_1$; and analogously, the multiplication of $\boldsymbol{\epsilon}_2$ with $\nabla(\boldsymbol{\epsilon}_1^T \mathbf{s}_{\boldsymbol{\theta}})$ yields $\boldsymbol{\epsilon}_2^T(\nabla \mathbf{s}_{\boldsymbol{\theta}})^T \boldsymbol{\epsilon}_1$. This trick requires only second time of gradient computations to estimate the regularization. Hence, the computational complexity of $\boldsymbol{\epsilon}_2^T(\nabla \mathbf{s}_{\boldsymbol{\theta}} - \nabla \mathbf{s}_{\boldsymbol{\theta}}^T)\boldsymbol{\epsilon}_1$ is $O(1)$.

**Proposition 2.** $\mathbf{s}_{\boldsymbol{\theta}} \in \mathbf{S}_{div}$ if and only if $\nabla_{\mathbf{z}_t} \mathbf{s}_{\boldsymbol{\theta}}(\mathbf{z}_t, t)$ is symmetric.

**Proposition 3.** A matrix $A \in \mathbb{R}^{d \times d}$ is symmetric if and only if $\mathbb{E}_{\boldsymbol{\epsilon}_1, \boldsymbol{\epsilon}_2 \sim \mathcal{N}(0, \mathbf{I})}\left[(\boldsymbol{\epsilon}_2^T(A - A^T)\boldsymbol{\epsilon}_1)^2\right] = 0$.

In fact, we can prove a bit stronger results in the next propositions.

**Proposition 4.** Let $\boldsymbol{\epsilon}_1$ and $\boldsymbol{\epsilon}_2$ be vectors of $d$ independent samples from a random variable $U$ with mean zero. Then

$$\mathbb{E}_{\boldsymbol{\epsilon}_1, \boldsymbol{\epsilon}_2}[(\boldsymbol{\epsilon}_2^T(A - A^T)\boldsymbol{\epsilon}_1)^2] = \mathbb{E}_U[U^2]^2 \|A - A^T\|_F^2$$

and

$$\begin{aligned}
Var\left((\boldsymbol{\epsilon}_2^T(A - A^T)\boldsymbol{\epsilon}_1)^2\right) &= Var(U^2)\Big(Var(U^2) + 2\big(Var(U) + \mathbb{E}_U[U]^2\big)^2\Big)\sum_{a,b}(\Delta A)_{ab}^4 \\
&+ 2\big(Var(U) + \mathbb{E}_U[U]^2\big)^2\Big(3Var(U^2) + 2\big(Var(U) + \mathbb{E}_U[U]^2\big)^2\Big)\sum_a \sum_{b \neq d}(\Delta A)_{ab}^2(\Delta A)_{ad}^2 \\
&+ 2\big(Var(U) + \mathbb{E}_U[U]^2\big)^4\Big(\sum_{a \neq c}\sum_{b \neq d}(\Delta A)_{ab}^2(\Delta A)_{cd}^2 \\
&\qquad\qquad + 3\sum_{a \neq c}\sum_{b \neq d}(\Delta A)_{ab}(\Delta A)_{ad}(\Delta A)_{cb}(\Delta A)_{cd}\Big),
\end{aligned}$$

where $(\Delta A)_{ab} := A_{ab} - A_{ba}$.

**Proposition 5.** *Let $U$ be the discrete random variable which takes the values $1, -1$ each with probability $1/2$. Then $(\boldsymbol{\epsilon}_2^T(A - A^T)\boldsymbol{\epsilon}_1)^2$ is the unbiased estimator of $\|A - A^T\|_F^2$. Moreover, $U$ is the unique random variable amongst zero-mean random variables for which the estimator is an unbiased estimator, and attains a minimum variance.*

Summing altogether, if it is the main focus to eliminate the rotation term in the score estimation, we could optimize $D_{KL}(\boldsymbol{\mu}_{\boldsymbol{\phi}}\|\boldsymbol{\nu}_{\boldsymbol{\phi},\boldsymbol{\theta}}) + \lambda \mathbb{E}_{\boldsymbol{\epsilon}_1,\boldsymbol{\epsilon}_2}\left[(\boldsymbol{\epsilon}_2^T(\nabla \mathbf{s}_{\boldsymbol{\theta}} - \nabla \mathbf{s}_{\boldsymbol{\theta}}^T)\boldsymbol{\epsilon}_1^T)^2\right]$, where $\boldsymbol{\epsilon}_1$ and $\boldsymbol{\epsilon}_2$ are the random variables of minimum variance, as proposed in Proposition 5. In practice, we find that the above regularized training loss is unnecessary for INDM because we already achieves the nearly MLE training, but it helps DDPM++ to reduce the variational gap at the expense of $4\times$ slower training speed than the training with unregularized loss in DDPM++. Even with reduced variational gap, we find that NLL of DDPM++ is improved only marginally only on certain training scenarios, and has no effect in most trials, so we leave the detailed effect of MLE training in diffusion models as a future work. Notably, therefore, we conclude that the NLL gain in INDM, compared to DDPM++, essentially originates from $\phi$-training and its consequential expanded solution space to $\mathbf{S}_{sol}$.

## C    Details on Section 6.2

### C.1    Full Statement of Theorem 4

We provide a full statement of Theorem 4. Theorem 4 is heavily influenced by the theoretic analysis of De Bortoli et al. [15], Guth et al. [22], and it could be considered as merely an application of their results. It is possible that the inequality in Theorem 4 could not be tight, but empirically the robustness is significantly connected to the initial distribution's smoothness.

**Theorem 4.** *Assume that there exists $M \geq 0$ such that for any $t \in [0, T]$ and $\mathbf{z} \in \mathbb{R}^d$, the score estimation is close enough to the forward score by $M$, $\|\mathbf{s}_{\boldsymbol{\theta}}(\mathbf{x}, t) - \nabla \log p_t^{\boldsymbol{\phi}}(\mathbf{x})\| \leq M$, with $\mathbf{s}_{\boldsymbol{\theta}} \in C([0, T] \times \mathbb{R}^d, \mathbb{R}^d)$. Assume that $\nabla \log p_t^{\boldsymbol{\phi}}(\mathbf{z})$ is $C^2$ in both $t$ and $\mathbf{z}$, and that $\sup_{\mathbf{z},t} \|\nabla^2 \log p_t^{\boldsymbol{\phi}}(\mathbf{z})\| \leq K$ and $\|\frac{\partial}{\partial t}\nabla \log p_t^{\boldsymbol{\phi}}(\mathbf{z})\| \leq Me^{-\alpha t}\|\mathbf{z}\|$ for some $K, M, \alpha > 0$. Suppose $(\mathbf{h}_{\boldsymbol{\phi}}^{-1})_\#$ s a push-forward map. Then $\|p_r - (\mathbf{h}_{\boldsymbol{\phi}}^{-1})_\# p_{0,N}^{\boldsymbol{\theta}}\|_{TV} \leq E_{pri}(\boldsymbol{\phi}) + E_{dis}(\boldsymbol{\phi}) + E_{est}(\boldsymbol{\phi}, \boldsymbol{\theta})$, where $E_{pri}(\boldsymbol{\phi}) = \sqrt{2}e^{-T}D_{KL}(p_T^{\boldsymbol{\phi}}\|\pi)^{1/2}$ is the error originating from the prior mismatch; $E_{dis}(\boldsymbol{\phi}) = 6\sqrt{\delta}(1 + \mathbb{E}_{p_0^{\boldsymbol{\phi}}(\mathbf{z})}[\|\mathbf{z}\|^4]^{1/4})(1 + K + M(1 + \frac{1}{\sqrt{2\alpha}}))$ is the discretization error with $\delta = \frac{\max \gamma_k^2}{\min \gamma_k}$; $E_{est}(\boldsymbol{\phi}, \boldsymbol{\theta}) = 2TM^2$ is the score estimation error.*

### C.2    Geometric Interpretation of Latent Diffusion

Figure 11 illustrates the diffusion trajectories of the probability flow ODE of VPSDE. It shows that the trajectories are highly nonlinear, and this section is devoted to analyze why such nonlinear trajectory occurs. Figure 12 shows two diffusion paths differing only on their scales on (a) the two moons dataset and (b) the ring dataset. The standard Gaussian distribution at $T$ has a larger variance than the initial data at the top row and has a smaller one at the bottom row on each dataset. For the visualization purpose, we zoom in the top row, and we zoom out the bottom row for each dataset, but we fix the xlim and ylim arguments in the matplotlib package [44] row-wisely. With this discrepancy of the initial data

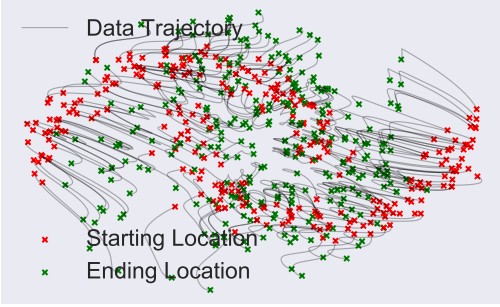

Figure 11: Particle trajectories of the probability flow ODE for VPSDE on the synthetic two moons 2d dataset.

scale, the particle trajectory at the bottom row is much more straightforward than in the top row, and it implies that the scale of initial data matters to the straightness of the bridge even if the diffusion SDE is identically linear.

A behind rationale for this observation comes from the closed-form solution of VPSDE. Suppose the forward diffusion follows VPSDE of $\mathrm{d}\mathbf{x}_t = -\frac{1}{2}\beta(t)\mathbf{x}_t\,\mathrm{d}t + \sqrt{\beta(t)}\,\mathrm{d}\mathbf{w}_t$. Then, the solution of this

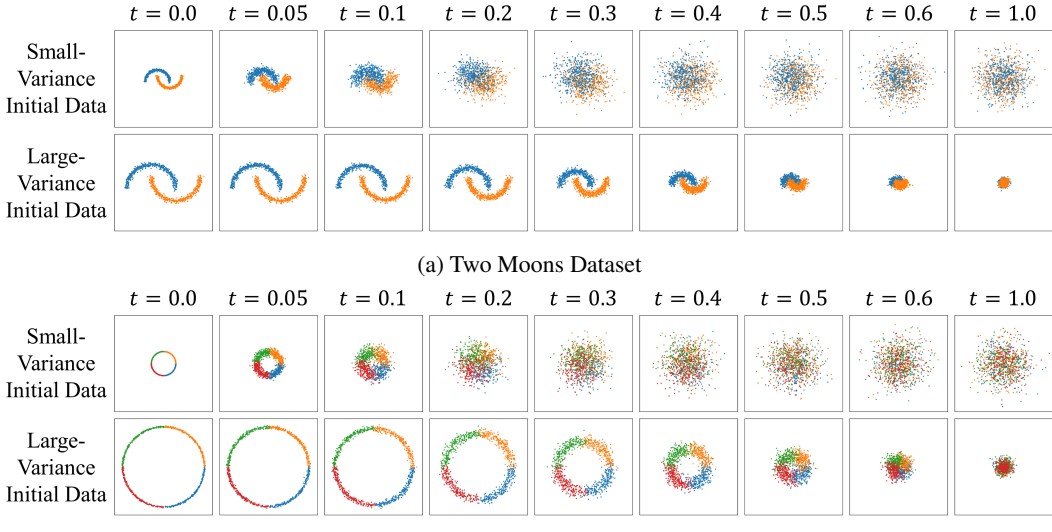

|  | t = 0.0 | t = 0.05 | t = 0.1 | t = 0.2 | t = 0.3 | t = 0.4 | t = 0.5 | t = 0.6 | t = 1.0 |
|---|---|---|---|---|---|---|---|---|---|

(a) Two Moons Dataset

(b) Ring Dataset

Figure 12: Comparison of linear diffusion bridges on data and latent spaces in diverse datasets.

Table 7: Statistics of data variable and latent variable on CIFAR-10. All statistics are averaged by dimension.

|  | Mean | Variance | Min | Max |
|---|---|---|---|---|
| DDPM++ ($\mathbf{x}_0 = \mathbf{z}_0^{\phi_{id}}$) | -0.05 | 0.25 | -1 | 1 |
| INDM ($\mathbf{z}_0^{\phi}$) | 0.70 | 9.74 | -8.66 | 12.17 |

SDE becomes

$$\mathbf{x}_t = \underbrace{e^{-\frac{1}{2} \int_0^t \beta(s)\,ds} \mathbf{x}_0}_{\text{linearly contraction mapping}} + \underbrace{\sqrt{1 - e^{-\int_0^t \beta(s)\,ds}} \boldsymbol{\epsilon}}_{\text{random perturbation}}, \tag{18}$$

where $\boldsymbol{\epsilon} \sim \mathcal{N}(0, \mathbf{I})$. As the drift term $-\frac{1}{2}\beta(t)\mathbf{x}_t$ ahead towards the origin of $\mathbb{R}^d$, the solution in Eq. (18) is a summation of the contraction mapping to the origin, $0 \in \mathbb{R}^d$, with a random noise function, where the magnitude of the random perturbation depends solely on the diffusion coefficient, $g(t) = \sqrt{\beta(t)}$. If $\mathbf{x}_0$ is inflated by $c\mathbf{x}_0$, then it becomes $\mathbf{x}_t = c \times e^{-\frac{1}{2} \int_0^t \beta(s)\,ds} \mathbf{x}_0 + \sqrt{1 - e^{-\int_0^t \beta(s)\,ds}} \boldsymbol{\epsilon}$ with contraction mapping multiplied by $c$. Therefore, as $c$ increases, the contraction force outweighs the random perturbing effect, and the particle trajectory is becoming straight.

On a high-dimensional dataset, most of the mass of the standard Gaussian $\pi = \mathcal{N}(0, \mathbf{I})$, which is the prior, is concentrated on a thin spherical shell with squared radius of $d$, according to the Gaussian annulus theorem [45], as

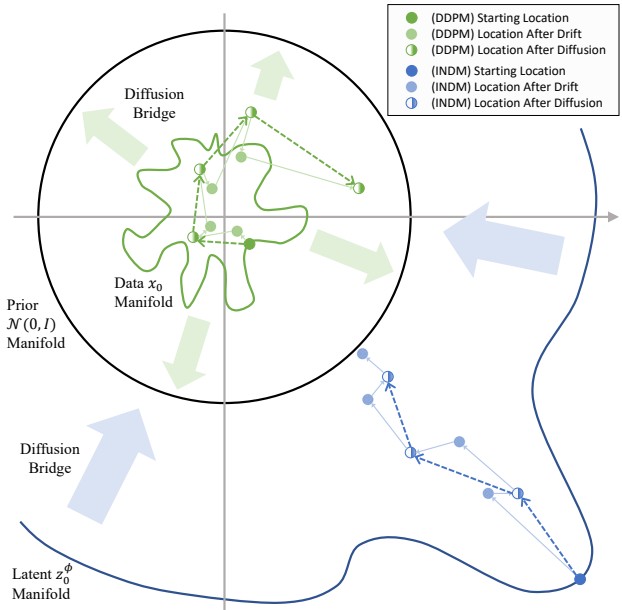

Figure 13: Descriptive Illustration On Diffusion Bridge.

described in the black circle of Figure 13. On CIFAR-10, the data distribution has the smaller average square radius of $\mathbb{E}_{p_r(\mathbf{x}_0)}\big[\|\mathbf{x}_0\|_2^2\big] = 776 < 3072 = d$, whereas the latent distribution has a larger

average square radius of $\mathbb{E}_{p_r(\mathbf{x}_0)}\big[\|\mathbf{h}_\phi(\mathbf{x}_0)\|_2^2\big] = \mathbb{E}_{p_0^\phi(\mathbf{z}_0^\phi)}\big[\|\mathbf{z}_0^\phi\|_2^2\big] > d$ than a standard Gaussian distribution. The latent radius varies from $5,385$ to $31,399$ by experimental settings. Thus, the latent manifold is located outside of the prior on CIFAR-10 as depicted in Figure 13.

When the latent manifold envelops the prior manifold, i.e., $\|\mathbf{z}_0^\phi\|_2 > \|\mathbf{z}_T^\phi\|_2$, the drift term, $-\frac{1}{2}\beta(t)\mathbf{z}_t^\phi$, and the vector of $\mathbf{z}_T^\phi - \mathbf{z}_0^\phi$ aligns towards the origin. On the other hand, if the initial manifold is located inside the prior manifold, i.e., $\|\mathbf{x}_0\|_2 < \|\mathbf{x}_T\|_2$, then the drift term points towards the opposite direction of $\mathbf{x}_T - \mathbf{x}_0$. This leads that the contraction mapping disturbs the particle to move towards $\mathbf{x}_T$, and it is the random perturbation that leads the particle to converge to $\mathbf{x}_T$. In latent trajectory, the contraction mapping driven by the drift term helps the particle moving towards $\mathbf{z}_T^\phi$. Therefore, the particle trajectory is more straightforward in the latent trajectory, which moves *outside* of the prior manifold, compared to the data trajectory that lives *inside* of the prior manifold. This clarifies why the sampling-friendly bridge is constructed in INDM.

Figure 14 presents the 2d toy case of the two moons dataset. It illustrates a simple visualization of the flow training. Figure 14 shows that even though the latent manifold is located near the data manifold at the initial phase of training in Figure 14-(a), after the training, the latent manifold is inflated to the outside of the real data in Figure 14-(b). Therefore, the probability flow ODE (deterministic trajectory), after the training, transports the initial mass to the final mass with a nearly linear line in Figure 14-(d), in contrast to the curvy VPSDE trajectory at the initial phase of training in Figure 14-(c). In this example, the flow training puts the latent manifold out of the data manifold, and this helps the robust sampling.

In addition, Figure 14 illustrates the Monge trajectories between the latent initial distribution and the prior distribution. As theoretically demonstrated in Gaussian and empirically shown in general distribution in Khrulkov and Oseledets [47], the encoder map of VPSDE is nearly optimal transport under the squared Euclidean cost function, where the encoder map is the mapping from the initial point to the final point passed through the probability flow ODE. Figure 14 supports this, and the diffusion trajectory becomes more straight alike to the optimal Monge map after the training.

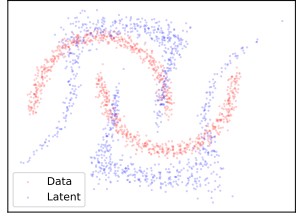

(a) Data and Latent Manifolds At Initial Stage of Training

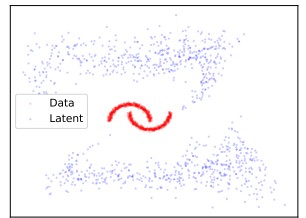

(b) Data and Latent Manifolds Afer Training of 10k Steps

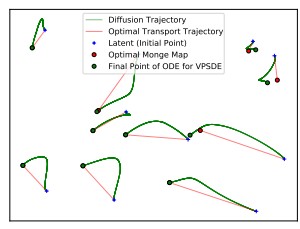

(c) Diffusion and (optimal) Monge Trajectories At Initial Stage of Training

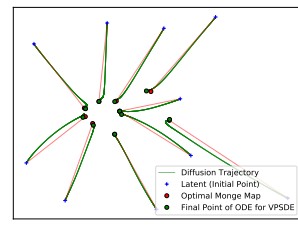

(d) Diffusion and (optimal) Monge Trajectories After Training of 10k Steps

Figure 14: (a,b) Latent manifold by training iterations (c,d) Diffusion trajectories by training iterations. We use Python Optimal Transport (POT) library [46] to obtain the optimally transported Monge map between 1,000 samples from the latent starting variable and the latent ending variable. We only visualize 10 samples out of 1,000 transport maps for a clear implication. In (c), we train the score network further until converged (with the fixed flow) to visualize accurate diffusion paths.

Figure 15 illustrates the concept of linearized diffusion path. As the flow inflates the latent manifold, the diffusion trajectory becomes more linear, and Figure 7 supports the conceptual illustration of Figure 15 on CIFAR-10.

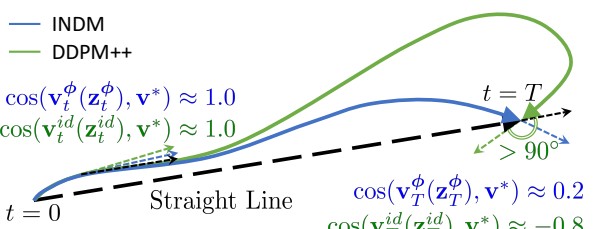

Figure 15: Illustrative Particle Trajectory.

# D   Related Work

## D.1   Latent Score-based Generative Model (LSGM)

The diffusion process on latent space is firstly introduced in LSGM. LSGM transforms the data variable to a latent variable, and estimates the prior distribution with a diffusion model. Suppose $\boldsymbol{\theta}$, $\boldsymbol{\phi}$, and $\boldsymbol{\psi}$ represent for the parameters for the score network, the encoder network, and the decoder network, respectively. Then, LSGM optimizes the loss of

$$
\begin{aligned}
D_{KL}(p_r \| p_{\boldsymbol{\theta},\boldsymbol{\psi}}) &\leq D_{KL}\big(p_r(\mathbf{x}_0)q_{\boldsymbol{\phi}}(\mathbf{z}_0|\mathbf{x}_0) \| p_{\boldsymbol{\theta}}(\mathbf{z}_0)p_{\boldsymbol{\psi}}(\mathbf{x}_0|\mathbf{z}_0)\big) \\
&= D_{KL}\big(p_r(\mathbf{x}_0)q_{\boldsymbol{\phi}}(\mathbf{z}_0|\mathbf{x}_0) \| q_{\boldsymbol{\phi}}(\mathbf{z}_0)p_{\boldsymbol{\psi}}(\mathbf{x}_0|\mathbf{z}_0)\big) + D_{KL}\big(q_{\boldsymbol{\phi}}(\mathbf{z}_0) \| p_{\boldsymbol{\theta}}(\mathbf{z}_0)\big) \\
&\leq D_{KL}\big(p_r(\mathbf{x}_0)q_{\boldsymbol{\phi}}(\mathbf{z}_0|\mathbf{x}_0) \| q_{\boldsymbol{\phi}}(\mathbf{z}_0)p_{\boldsymbol{\psi}}(\mathbf{x}_0|\mathbf{z}_0)\big) + D_{KL}\big(\boldsymbol{\mu}_{\boldsymbol{\phi}}(\{\mathbf{z}_t\}_{t=0}^T) \| \boldsymbol{\nu}_{\boldsymbol{\theta}}(\{\mathbf{z}_t\}_{t=0}^T)\big) \\
&= \mathcal{L}_{LSGM}(\boldsymbol{\theta},\boldsymbol{\phi},\boldsymbol{\psi})
\end{aligned}
$$

where $q_{\boldsymbol{\phi}}(\mathbf{z}_0)$ is the marginal distribution of the encoder posterior, $q_{\boldsymbol{\phi}}(\mathbf{z}_0) = \int p_r(\mathbf{x}_0)q_{\boldsymbol{\phi}}(\mathbf{z}_0|\mathbf{x}_0)\,\mathrm{d}\mathbf{x}_0$.

As well as INDM, LSGM also optimizes the log-likelihood of the model distribution by using a diffusion model in the latent space. Though both INDM and LSGM losses include a denoising score loss on the latent space (which is the KL divergence between path measures on the latent space), $\mathcal{L}_{LSGM}(\boldsymbol{\theta},\boldsymbol{\phi},\boldsymbol{\psi})$ is not equivalent to the KL divergence between the forward and generative path measures on the data space, in contrast to INDM with $D_{KL}(\boldsymbol{\mu}_{\boldsymbol{\phi}}(\{\mathbf{x}_t\}_{t=0}^T \| \boldsymbol{\nu}_{\boldsymbol{\phi},\boldsymbol{\theta}}(\{\mathbf{x}_t\}_{t=0}^T)$ as its loss function. In fact, there is no forward SDE (green path in Figure 3) on the data space in LSGM according to Lemma 3, which is a direct application of the Borsuk-Ulam theorem [48].

**Lemma 3** ($\mathbb{R}^n$ **is not homeomorphic to** $\mathbb{R}^m$ [48])**.** *If $n \neq m$, there is no continuous map $E : \mathbb{R}^n \to \mathbb{R}^m$ that has the continuous inverse map $E^{-1} : \mathbb{R}^m \to \mathbb{R}^n$.*

Lemma 3 implies that there is no inverse function of the encoder as long as the latent dimension is different from the data dimension (and the activation function is continuous, such as ReLU). From this, LSGM cannot define a random variable on the data space by $\mathbf{x}_t^{\boldsymbol{\phi}} = E_{\boldsymbol{\phi}}^{-1}(\mathbf{z}_t)$, in contrast to INDM that defines $\mathbf{x}_t^{\boldsymbol{\phi}} = \mathbf{h}_{\boldsymbol{\phi}}^{-1}(\mathbf{z}_t)$. This non-existence of random variables on the data space implies that *the forward diffusion process does not exists as long as the latent dimension differs to the data dimension*.

With the above theoretic dilemma of LSGM, one could build a generative diffusion process on the data space. If $\mathbf{x}_t^{\boldsymbol{\psi},\boldsymbol{\theta}} := D_{\boldsymbol{\psi}}(\mathbf{z}_t^{\boldsymbol{\theta}})$, where $\mathbf{z}_t^{\boldsymbol{\theta}}$ is a generative random variable on the latent space, and $D_{\boldsymbol{\psi}}$ is a decoder map, then we could build a generative diffusion process on the data space through the Ito's formula in the same way as we did in INDM. Inspired by this, one could argue that the forward diffusion could be constructed by $\mathbf{x}_t^{\boldsymbol{\psi}} := D_{\boldsymbol{\psi}}(\mathbf{z}_t)$, where $\mathbf{z}_t$ is a forward random variable on the latent space. This construction enables to construct a forward diffusion process on the latent space, but there are a couple of caveats to this construction.

Theoretically, this forward diffusion process starts from the reconstructed variable, $\mathbf{x}_0^{\boldsymbol{\psi}} = D_{\boldsymbol{\psi}}(\mathbf{z}_0) = D_{\boldsymbol{\psi}}(E_{\boldsymbol{\phi}}(\mathbf{x}_0)) = \mathbf{x}_{rec}$, where $\mathbf{x}_0$ and $\mathbf{x}_{rec}$ differs throughout the training procedure. In addition, even if we admit $\{\mathbf{x}_t^{\boldsymbol{\psi}}\}$ as a forward diffusion, $\mathcal{L}_{LSGM}(\boldsymbol{\theta},\boldsymbol{\phi},\boldsymbol{\psi})$ cannot be derived as the KL divergence of path measures for the forward diffusion (admittably $\{\mathbf{x}_t^{\boldsymbol{\psi}}\}$, but not true to be

Table 8: LSGM training fails when using the variance weighting function.

|  | NLL | NELBO | FID |
|---|---|---|---|
| LSGM (VP, FID) | NaN | NaN | NaN |
| INDM (VP, FID) | 3.23 | 3.17 | 2.90 |

precise) and the generative diffusion ($\mathbf{x}_t^{\boldsymbol{\psi},\boldsymbol{\theta}}$) on the data space. Instead, the loss contains the encoder parameters to optimize, and the loss diverges from the KL divergence on the data space. Also, hypothetically, even if the loss is the KL divergence of the forward and generative path measures on the data space, the optimization could be drifted away from the optimal point because the forward diffusion starts from untrained reconstructed variable, $\mathbf{x}_{rec}$, which is not close to the data variable, $\mathbf{x}_0$. This analysis provides a clue to explain the training instability of LSGM as reported in Vahdat et al. [9] and Dockhorn et al. [20], in contrast to INDM that is stable to train in any training configuration. Table 8 shows a fast comparison of LSGM and INDM with variance weighting function, sampled from $t \in \mathcal{U}[0,1]$. *NaN* indicates experiments that fail due to training instability, see Section 5.2 and Table 6 of Vahdat et al. [9] and Section E.2.7 of Dockhorn et al. [20].

Table 9: Comparison of latent dimension of INDM and LSGM.

| Datset | Data Dimension | Latent Dimension of INDM | Latent Dimension of INDM |
|---|---|---|---|
| MNIST | 784 | 784 | 2,560 |
| CIFAR-10 | 3,072 | 3,072 | 46,080 |
| CelebA-HQ 256 | 196,608 | 196,608 | 819,200 |

Moreover, Table 9 compares INDM with LSGM in terms of the latent dimension. We compute the latent dimension of LSGM, according to their paper and released checkpoint. Contrary to the dimensional reduction property which is the crux of the auto-encoding structure, LSGM maps data into a latent space of a much higher dimension than the data dimension. LSGM is known to perform well, but having observed 15x higher latent dimension than the data dimension on CIFAR-10, the good performance was not gained for free. On the other hand, INDM always retains the same dimension to the data, while keeping the invertibility.

## D.2 Diffusion Normalizing Flow (DiffFlow)

The Girsanov theorem [35] proves that the variational bound is derived by

$$D_{KL}(p_r\|p_{\boldsymbol{\theta}}) \leq \frac{1}{2}\int_0^T g^2(t)\mathbb{E}_{p_r(\mathbf{x}_0)}\mathbb{E}_{p_{0t}(\mathbf{x}_t|\mathbf{x}_0)}\big[\|\mathbf{s}_{\boldsymbol{\theta}}(\mathbf{x}_t,t) - \nabla_{\mathbf{x}_t}\log p_{0t}(\mathbf{x}_t|\mathbf{x}_0)\|_2^2\big]\,\mathrm{d}t + D_{KL}(p_T\|\pi). \tag{19}$$

When the forward diffusion is given as $\mathrm{d}\mathbf{x}_t = \mathbf{f}_{\boldsymbol{\phi}}(\mathbf{x}_t,t)\,\mathrm{d}t + g(t)\,\mathrm{d}\mathbf{w}_t$, where $\mathbf{f}_{\boldsymbol{\phi}}$ is an explicit parametrization of the drift term by a normalizing flow with parameters $\boldsymbol{\phi}$, then the transition probability, $p_{0t}(\mathbf{x}_t|\mathbf{x}_0)$, becomes intractable. Therefore, optimizing the continuous variational bound is not feasible. One might detour this issue by alternatively optimizing the continuous DDPM++ loss of

$$\int_0^T \tilde{\lambda}(t)\mathbb{E}_{p_r(\mathbf{x}_0)}\mathbb{E}_{\boldsymbol{\epsilon}\sim\mathcal{N}(0,\mathbf{I})}\big[\|\boldsymbol{\epsilon} - \hat{\boldsymbol{\epsilon}}_{\boldsymbol{\theta}}(\mathbf{x}_t,t)\|_2^2\big]\,\mathrm{d}t, \tag{20}$$

but the denoising score loss of Eq. (19) is not equivalent to the continuous DDPM++ loss of Eq. (20) when the transition probability is no longer a Gaussian distribution.

DiffFlow detours the intractability issue of the continuous loss of Eq. (19) by discretizing the nonlinear SDE in the Euler-Maruyama (EM) fashion [49]. We construct the discrete random variables that approximate the nonlinear SDE by the induction. If $\mathbf{x}_{t_0}^{\phi,\mathrm{EM}} := \mathbf{x}_0^{\phi}$ and $\Delta t_i := t_i - t_{i-1}$, where $\{t_i\}_{t=0}^N$ are discretization timesteps with $t_0 = 0$ and $t_N = T$, then the solution of the nonlinear SDE that starts from $\mathbf{x}_{t_{i-1}}^{\phi,\mathrm{EM}}$ is

$$\mathbf{x}_{t_i}^{\phi} - \mathbf{x}_{t_{i-1}}^{\phi,\mathrm{EM}} = \int_{t_{i-1}}^{t_i} \mathbf{f}_{\boldsymbol{\phi}}(\mathbf{x}_t^{\phi},t)\,\mathrm{d}t + \int_{t_{i-1}}^{t_i} g(t)\,\mathrm{d}\mathbf{w}_t. \tag{21}$$

Here, the integral of the drift term is

$$\int_{t_{i-1}}^{t_i} \mathbf{f}_{\boldsymbol{\phi}}(\mathbf{x}_t^{\phi},t)\,\mathrm{d}t = \int_{t_{i-1}}^{t_i} \mathbf{f}_{\boldsymbol{\phi}}\big(\mathbf{x}_{t_{i-1}}^{\phi,\mathrm{EM}} + (\mathbf{x}_t^{\phi} - \mathbf{x}_{t_{i-1}}^{\phi,\mathrm{EM}}), t_{i-1} + (t - t_{i-1})\big)\,\mathrm{d}t$$

$$= \int_{t_{i-1}}^{t_i} \mathbf{f}_{\boldsymbol{\phi}}(\mathbf{x}_{t_{i-1}}^{\phi,\mathrm{EM}}, t_{i-1})\,\mathrm{d}t + O(\Delta t_i^{3/2})$$

$$= \mathbf{f}_{\boldsymbol{\phi}}(\mathbf{x}_{t_{i-1}}^{\phi,\mathrm{EM}}, t_{i-1})\Delta t_i + O(\Delta t_i^{3/2}),$$

and the integral of the volatility term is

$$\int_{t_{i-1}}^{t_i} g(t)\,\mathrm{d}\mathbf{w}_t = g(t_{i-1})(\mathbf{w}_{t_i} - \mathbf{w}_{t_{i-1}}) + O(\Delta t_i^{3/2}) = g(t_{i-1})\boldsymbol{\epsilon}\sqrt{\Delta t_i} + O(\Delta t_i^{3/2}),$$

where $\boldsymbol{\epsilon} \sim \mathcal{N}(0,\mathbf{I})$. Therefore, DiffFlow defines the next discretized random variable, $\mathbf{x}_{t_i}^{\phi,\mathrm{EM}}$, to be

$$\mathbf{x}_{t_i}^{\phi} = \mathbf{x}_{t_{i-1}}^{\phi,\mathrm{EM}} + \mathbf{f}_{\boldsymbol{\phi}}(\mathbf{x}_{t_{i-1}}^{\phi,\mathrm{EM}}, t_{i-1})\Delta t_i + g(t_{i-1})\boldsymbol{\epsilon}\sqrt{\Delta t_i} + O(\Delta t_i^{2/3})$$

$$\approx \mathbf{x}_{t_{i-1}}^{\phi,\text{EM}} + \mathbf{f}_\phi(\mathbf{x}_{t_{i-1}}^{\phi,\text{EM}}, t_{i-1})\Delta t_i + g(t_{i-1})\boldsymbol{\epsilon}\sqrt{\Delta t_i}$$
$$= \mathbf{x}_{t_i}^{\phi,\text{EM}},$$

and this Euler-Maruyama random variable $\mathbf{x}_{t_i}^{\phi,\text{EM}}$ follows a Gaussian distribution of mean $\mathbf{x}_{t_{i-1}}^{\phi,\text{EM}} + \mathbf{f}_\phi(\mathbf{x}_{t_{i-1}}^{\phi,\text{EM}}, t_{i-1})\Delta t_i$ and variance $g^2(t_{i-1})\Delta t_i$. Note that this discretization approximates the nonlinear SDE with a finite Markov chain of $\{\mathbf{x}_{t_i}^{\text{EM}}\}_{i=0}^N$.

DiffFlow constructs the generative process as

$$\mathbf{x}_{t_{i-1}}^{\boldsymbol{\theta}} = \mathbf{x}_{t_i}^{\boldsymbol{\theta}} - \left[ \mathbf{f}_\phi(\mathbf{x}_{t_i}^\phi, t_i) - g^2(t_i)\mathbf{s}_{\boldsymbol{\theta}}(\mathbf{x}_{t_i}^\phi, t_i) \right]\Delta t_i + g(t_i)\boldsymbol{\epsilon}\sqrt{\Delta t_i}.$$

Then, from the Jensen's inequality, the discrete DDPM loss satisfies

$$D_{KL}(p_r \| p_{\phi,\boldsymbol{\theta}}) \le \sum_{i=1}^{N-1} \mathbb{E}_{p_r(\mathbf{x}_{t_0}^{\text{EM}})} \mathbb{E}_{p_\phi(\mathbf{x}_{t_i}^{\text{EM}}, \mathbf{x}_{t_{i-1}}^{\text{EM}} | \mathbf{x}_{t_0}^{\text{EM}})} \left[ D_{KL}(p_\phi(\mathbf{x}_{t_{i-1}}^{\text{EM}} | \mathbf{x}_{t_i}^{\text{EM}}, \mathbf{x}_{t_0}^{\text{EM}}) \| p_{\boldsymbol{\theta}}(\mathbf{x}_{t_{i-1}}^{\text{EM}} | \mathbf{x}_{t_i}^{\text{EM}})) \right]. \tag{22}$$

While the true inference distribution on the continuous variables, $p_\phi(\mathbf{x}_{t_{i-1}} | \mathbf{x}_{t_i}, \mathbf{x}_{t_0})$, is not a Gaussian distribution due to terms related to $O(\Delta t_i^{3/2})$, the inference distribution on the *discretized* variables, $p_\phi(\mathbf{x}_{t_{i-1}}^{\text{EM}} | \mathbf{x}_{t_i}^{\text{EM}}, \mathbf{x}_{t_0}^{\text{EM}})$, becomes a Gaussian distribution by the Euler-Maruyama-style discretization. Therefore, Eq. (22) reduces to a tractable loss that does not need to compute the transition probability:

$$D_{KL}(p_r \| p_{\phi,\boldsymbol{\theta}}) \le \sum_{i=1}^{N-1} \mathbb{E}_{p_r(\mathbf{x}_{t_0}^{\text{EM}})} \mathbb{E}_{p_\phi(\mathbf{x}_{t_i}^{\text{EM}}, \mathbf{x}_{t_{i-1}}^{\text{EM}} | \mathbf{x}_{t_0}^{\text{EM}})} \left[ D_{KL}(p_\phi(\mathbf{x}_{t_{i-1}}^{\text{EM}} | \mathbf{x}_{t_i}^{\text{EM}}, \mathbf{x}_{t_0}^{\text{EM}}) \| p_{\boldsymbol{\theta}}(\mathbf{x}_{t_{i-1}}^{\text{EM}} | \mathbf{x}_{t_i}^{\text{EM}})) \right]$$
$$= \frac{1}{2} \sum_{i=1}^{N-1} \mathbb{E}_{p_r(\mathbf{x}_{t_0}^{\text{EM}})} \mathbb{E}_{p_\phi(\mathbf{x}_{t_i}^{\text{EM}}, \mathbf{x}_{t_{i-1}}^{\text{EM}} | \mathbf{x}_{t_0}^{\text{EM}})} \left[ \frac{1}{g^2(t_i)\Delta t_i} \right\| \mathbf{x}_{t_{i-1}}^{\text{EM}} - \mathbf{x}_{t_i}^{\text{EM}}$$
$$+ \left[ \mathbf{f}_\phi(\mathbf{x}_{t_i}^{\text{EM}}, t_i) - g^2(t_i)\mathbf{s}_{\boldsymbol{\theta}}(\mathbf{x}_{t_i}^{\text{EM}}, t_i) \right]\Delta t_i \Big\|_2^2 \right]$$
$$= \mathcal{L}_{\text{DiffFlow}}(\phi, \boldsymbol{\theta}) \tag{23}$$

While Eq. (23) does not need to compute the transition probability, another issue of optimizing the variational bound originates from the expectation of $\mathbb{E}_{p_\phi(\mathbf{x}_{t_i}^{\text{EM}}, \mathbf{x}_{t_{i-1}}^{\text{EM}} | \mathbf{x}_{t_0}^{EM})}$. The empirical Monte-Carlo estimation is too expensive because a realization of $\mathbf{x}_{t_i}^{\text{EM}}$ needs $i$ number of flow evaluations. In total, summing $i$ over $i = 1$ to $N$ requires $O(N^2)$ flow evaluations to estimate the discrete variational bound of Eq. (23). Therefore, DiffFlow exchanges the summation and the expectation to reduce the number of flow evaluations by

$$\mathcal{L}_{\text{DiffFlow}}(\phi, \boldsymbol{\theta}) = \frac{1}{2} \mathbb{E}_{\{\mathbf{x}_{t_i}\}_{i=0}^{N-1} \sim p_\phi(\mathbf{x}_{t_0}, \dots, \mathbf{x}_{t_{N-1}})} \left[ \sum_{i=1}^{N-1} \frac{1}{g^2(t_i)\Delta t_i} \right\| \mathbf{x}_{t_{i-1}} - \mathbf{x}_{t_i}$$
$$+ \left[ \mathbf{f}_\phi(\mathbf{x}_{t_i}, t_i) - g^2(t_i)\mathbf{s}_{\boldsymbol{\theta}}(\mathbf{x}_{t_i}, t_i) \right]\Delta t_i \Big\|_2^2 \right]. \tag{24}$$

This reformulated Eq. (24) estimates $\mathcal{L}_{\text{DiffFlow}}$ with a single sample path from the Markov chain of $\{\mathbf{x}_{t_i}^{\text{EM}}\}_{t=1}^N$, so it requires $O(N)$ flow evaluations to estimate $\mathcal{L}_{\text{DiffFlow}}(\phi, \boldsymbol{\theta})$. Therefore, DiffFlow takes $O(N)$ computational complexity in total for every optimization step.

There are five differences between DiffFlow and INDM. Basically, these differences arise from the different usage of the flow transformation between DiffFlow and INDM. First, INDM enables to train the continuous diffusion model without the sacrifice on training time, while DiffFlow is limited on the discrete diffusion model at the expense of slower training time. DiffFlow approximates the forward nonlinear SDE with a finite Markov chain. Suppose $\mathbf{x}_t^{\text{EM}}$ to be the continuous-time random variable defined by $\mathbf{x}_t^{\text{EM}} = \mathbf{x}_{t_{i-1}}^{\text{EM}} + \mathbf{f}_\phi(\mathbf{x}_{t_{i-1}}^{\text{EM}}, t_{i-1})(t - t_{i-1}) + g(t_{i-1})\boldsymbol{\epsilon}\sqrt{t - t_{i-1}}$ on time range of $t \in [t_{i-1}, t_i)$, then we have

$$\mathbb{E}\left[ \|\mathbf{x}_t - \mathbf{x}_t^{\text{EM}}\|_2 \right] \le C\sqrt{\Delta t_i}, \tag{25}$$

where $C = C(T, K, \mathbb{E}[\|\mathbf{x}_0\|_2^2]) \geq O(K^2)$ is a constant with $K$ being a Lipschits constant of

$$\|\mathbf{f}_\phi(\mathbf{x}, t) - \mathbf{f}_\phi(\mathbf{y}, t)\|_2 \leq K\|\mathbf{x} - \mathbf{y}\|_2$$

and

$$\|\mathbf{f}_\phi(\mathbf{x}, t)\|_2 + |g(t)| \leq K(1 + \|\mathbf{x}\|_2)$$

for all $\mathbf{x}, \mathbf{y} \in \mathbb{R}^d$ and $t \in [t_{i-1}, t_i)$. Having that $\Delta t_i$ is fixed a-priori, the upper bound in Inequality (25) could be arbitrarily large becuase it depends on $K$ that represents the magnitude of nonlinearity of $\mathbf{f}_\phi$. For instance, if $\mathbf{f}_\phi(\mathbf{x}_t, t) = \mathbf{x}_t^2$, then there does not exist any $K > 0$ that satisfies above Lipschitz bounds. In such case, it is unable to guarantee the tightness of the discretized Markov chain to the continuous nonlinear SDE in the classical sense. Therefore, the Euler-Maruyama approximation of the nonlinear SDE should take $N$ as many as possible if we want to regard the finite Markov chain as a discretized nonlinear SDE, which would eventually increase the training, evaluation, and sampling time.

Second, the computational complexity of INDM is $O(1)$ because the flow is evaluated only once at every optimization step. This is because the INDM loss is simply an addition of the flow loss and the linear diffusion loss. The training time of DiffFlow will be prohibitive as $N$ increases.

Third, our INDM jointly models both drift and volatility terms nonlinearly, whereas DiffFlow nonlinearly models only the drift term. As illustrated in Figure 1 and 2-(c) in the main paper, nonlinearizing the volatility term brings a different diffusion to the overall process, compared to a diffusion that arises from a nonlinear drift. In particular, Figure 2-(c) depicts that the data-dependent volatility term yields an ellipsoidal covariance in the noise distribution, which was assumed to have a fixed diagonal covariance in previous research, as illustrated in Figure 6. In INDM, this covariance becomes the subject of matter to optimize.

DiffFlow, as its current form, cannot impose nonlinearity to the volatility term because the discretized Markov chain is not a Gaussian distribution, anymore. To clarify, suppose a SDE of $d\mathbf{x}_t = \mathbf{f}_\phi(\mathbf{x}_t, t)\,dt + \mathbf{G}_\phi(\mathbf{x}_t, t)\,d\mathbf{w}_t$ (think of the green path of Figure 3 in the main paper) starts from a random variable $\mathbf{x}_{t_{i-1}}^{\text{EM}}$. The next discrete random variable of the Euler-Maruyama discretization is the approximate solution of this SDE at $t = t_i$, so let us approximate the right-hand-side of Eq. (26):

$$\mathbf{x}_{t_i} - \mathbf{x}_{t_{i-1}}^{\text{EM}} = \int_{t_{i-1}}^{t_i} \mathbf{f}_\phi(\mathbf{x}_t, t)\,dt + \int_{t_{i-1}}^{t_i} \mathbf{G}_\phi(\mathbf{x}_t, t)\,d\mathbf{w}_t. \tag{26}$$

The integral of the volatility term is

$$\int_{t_{i-1}}^{t_i} \mathbf{G}_\phi(\mathbf{x}_t, t)\,d\mathbf{w}_t = \int_{t_{i-1}}^{t_i} \mathbf{G}_\phi\big(\mathbf{x}_{t_{i-1}}^{\text{EM}} + (\mathbf{x}_t - \mathbf{x}_{t_{i-1}}^{\text{EM}}), t_{i-1} + (t - t_{i-1})\big)\,d\mathbf{w}_t$$

and since $\mathbf{x}_t - \mathbf{x}_{t_{i-1}}^{\text{EM}} = \mathbf{G}_\phi(\mathbf{x}_{t_{i-1}}^{\text{EM}}, t_{i-1})(\mathbf{w}_t - \mathbf{w}_{t_{i-1}}) + O(\Delta t_i)$, we get

$$\int_{t_{i-1}}^{t_i} \mathbf{G}_\phi(\mathbf{x}_t, t)\,d\mathbf{w}_t$$

$$= \mathbf{G}_\phi(\mathbf{x}_{t_{i-1}}^{\text{EM}}, t_{i-1})(\mathbf{w}_{t_i} - \mathbf{w}_{t_{i-1}})$$

$$\quad + \mathbf{G}_\phi(\mathbf{x}_{t_{i-1}}^{\text{EM}}, t_{i-1})\frac{\partial \mathbf{G}_\phi(\mathbf{x}_t, t)}{\partial \mathbf{x}_t}\big|_{\mathbf{x}_{t_{i-1}}^{\text{EM}}} \int_{t_{i-1}}^{t_i} \mathbf{w}_t - \mathbf{w}_{t_{i-1}}\,d\mathbf{w}_t + O(\Delta t_i^2)$$

$$= \mathbf{G}_\phi(\mathbf{x}_{t_{i-1}}^{\text{EM}}, t_{i-1})(\mathbf{w}_{t_i} - \mathbf{w}_{t_{i-1}})$$

$$\quad + \mathbf{G}_\phi(\mathbf{x}_{t_{i-1}}^{\text{EM}}, t_{i-1})\nabla_{\mathbf{x}_{t_{i-1}}^{\text{EM}}} \mathbf{G}_\phi(\mathbf{x}_{t_{i-1}}^{\text{EM}}, t_{i-1})\frac{1}{2}\big((\mathbf{w}_{t_i} - \mathbf{w}_{t_{i-1}})^2 - \Delta t_i\big) + O(\Delta t_i^2)$$

$$= \mathbf{G}_\phi(\mathbf{x}_{t_{i-1}}^{\text{EM}}, t_{i-1})\epsilon\sqrt{\Delta t_i}$$

$$\quad + \frac{1}{2}\mathbf{G}_\phi(\mathbf{x}_{t_{i-1}}^{\text{EM}}, t_{i-1})\nabla_{\mathbf{x}_{t_{i-1}}^{\text{EM}}} \mathbf{G}_\phi(\mathbf{x}_{t_{i-1}}^{\text{EM}}, t_{i-1})\big(\epsilon^2 - 1\big)\Delta t_i + O(\Delta t_i^2),$$

where $\epsilon \sim \mathcal{N}(0, \mathbf{I})$ and $\int_{t_{i-1}}^{t_i} \mathbf{w}_t - \mathbf{w}_{t_{i-1}}\,d\mathbf{w}_t = \int_{t_{i-1}}^{t_i} \mathbf{w}_t\,d\mathbf{w}_t - \mathbf{w}_{t_{i-1}}(\mathbf{w}_{t_i} - \mathbf{w}_{t_{i-1}}) = \int_{t_{i-1}}^{t_i} \frac{1}{2}d(\mathbf{w}_t^2) - \int_{t_{i-1}}^{t_i} \frac{1}{2}dt - \mathbf{w}_{t_{i-1}}(\mathbf{w}_{t_i} - \mathbf{w}_{t_{i-1}}) = \frac{1}{2}(\mathbf{w}_{t_i}^2 - \mathbf{w}_{t_{i-1}}^2 - \Delta t_i) - \mathbf{w}_{t_{i-1}}(\mathbf{w}_{t_i} - \mathbf{w}_{t_{i-1}}) =$

$\frac{1}{2}\big((\mathbf{w}_{t_i} - \mathbf{w}_{t_{i-1}})^2 - \Delta t_i\big)$ is according to the Ito's formula [17]. As $\mathbf{G}_\phi(\mathbf{x}_t, t)$ now depends on $\mathbf{x}_t$, the term including $(\epsilon^2 - 1)$ does not vanish. Therefore, $\mathbf{x}_{t_i}^{\text{EM}}$ is approximated by

$$
\begin{aligned}
\mathbf{x}_{t_i} =& \mathbf{x}_{t_{i-1}}^{\text{EM}} + \mathbf{f}_\phi(\mathbf{x}_{t_{i-1}}^{\text{EM}}, t_{i-1})\Delta t_i + \mathbf{G}_\phi(\mathbf{x}_{t_{i-1}}^{\text{EM}}, t_{i-1})\epsilon\sqrt{\Delta t_i} \\
& + \frac{1}{2}\mathbf{G}_\phi(\mathbf{x}_{t_{i-1}}^{\text{EM}}, t_{i-1})\nabla_{\mathbf{x}_{t_{i-1}}^{\text{EM}}}\mathbf{G}_\phi(\mathbf{x}_{t_{i-1}}^{\text{EM}}, t_{i-1})(\epsilon^2 - 1)\Delta t_i + O(\Delta t_i^{3/2}) \\
\approx & \mathbf{x}_{t_{i-1}}^{\text{EM}} + \mathbf{f}_\phi(\mathbf{x}_{t_{i-1}}^{\text{EM}}, t_{i-1})\Delta t_i + \mathbf{G}(\mathbf{x}_{t_{i-1}}^{\text{EM}}, t_{i-1})\epsilon\sqrt{\Delta t_i} \\
& + \frac{1}{2}\mathbf{G}_\phi(\mathbf{x}_{t_{i-1}}^{\text{EM}}, t_{i-1})\nabla_{\mathbf{x}_{t_{i-1}}^{\text{EM}}}\mathbf{G}_\phi(\mathbf{x}_{t_{i-1}}^{\text{EM}}, t_{i-1})(\epsilon^2 - 1)\Delta t_i \\
:= & \mathbf{x}_{t_i}^{\text{EM}}.
\end{aligned}
\tag{27}
$$

The order of the term $\frac{1}{2}\mathbf{G}_\phi(\mathbf{x}_{t_{i-1}}^{\text{EM}}, t_{i-1})\nabla_{\mathbf{x}_{t_{i-1}}^{\text{EM}}}\mathbf{G}_\phi(\mathbf{x}_{t_{i-1}}^{\text{EM}}, t_{i-1})(\epsilon^2 - 1)\Delta t_i$ is $O(\Delta t_i)$, which is the same order of the term $\mathbf{f}_\phi(\mathbf{x}_{t_{i-1}}^{\text{EM}}, t_{i-1})\Delta t_i$. Thus, this last term including $\epsilon^2$ cannot be ignored in the approximation.

With this approximation, the discretized random variable, $\mathbf{x}_{t_i}^{\text{EM}}$, includes a term of $\epsilon^2$, which is the square of the Brownian motion that does not follow a Gaussian distribution. Therefore, the variational bound of Eq. (22) is no longer reduced to a tractable loss, such as Eq. (23), and as a consequence, Eq. (22) is not optimizable even though the nonlinear SDE is discretized. Therefore, we have to ignore the last term, $\frac{1}{2}\mathbf{G}_\phi\nabla\mathbf{G}_\phi(\epsilon^2 - 1)\Delta t_i$, to tractably optimize the variational bound, but such ingorance equals to the approximation of DiffFlow, which would incur a large approximation error if $\mathbf{G}_\phi$ nonlinearly depends on $\mathbf{x}_t$. This leads DiffFlow limited on $\mathbf{G}_\phi(\mathbf{x}_t, t) = g_\phi(t)$, at its maximal capacity. This is contrastive to the result of INDM illustrated in Figure 6.

Fourth, as the generative process of DiffFlow starts from an easy-to-sample prior distribution, the flexibility of $\mathbf{f}_\phi$ is severely restricted to constrain $p_T^\phi(\mathbf{x}_T^\phi) \approx \pi(\mathbf{x}_T^\phi)$. The feasible space of nonlinear $\mathbf{f}_\phi$ that satisfies this constraint does not seem to be derived explicitly. Contrastive to DiffFlow, the data diffusion does not have to end at $\pi$ in INDM. Instead, INDM assumes the linear diffusion on the latent variable, so the ending variable on the latent space, $\mathbf{z}_T^\phi$, is already close to the prior distribution. Therefore, the space of admissible nonlinear drift in INDM, which is *explicitly* desribed in Eq. (10), should be larger than the space of DiffFlow. A lesson from this is that the explicit parametrization seems to be intuitive, but underneath the surface, not many properties could be uncovered explicitly, whereas the implicit parametrization using the invertible transformation enjoys its explicit derivations that enable to analyze concrete properties.

Fifth, DiffFlow estimates its loss of Eq. (24) using a single (or multiple) path to update the parameters with the reparametrization trick [50]. On the other hand, the discretized diffusion model estimates its loss with Eq. (23), where the sampling from $p_{0t}(\mathbf{x}_t|\mathbf{x}_0)$ is inexpensive because the transition probability is a Gaussian distribution. Therefore, the losses of Eqs. (24) and (23) coincide in the expectation sense, but they are estimated differently between DiffFlow and diffusion models with analytic transition probabilities. Taking $\frac{1}{g^2(t_i)\Delta t_i}\big\|\mathbf{x}_{t_{i-1}}^{\text{EM}} - \mathbf{x}_{t_i}^{\text{EM}} + \big[\mathbf{f}_\phi(\mathbf{x}_{t_i}^{\text{EM}}, t_i) - g^2(t_i)\mathbf{s}_\theta(\mathbf{x}_{t_i}^{\text{EM}}, t_i)\big]\Delta t_i\big\|_2^2$ as a random variable $X_i$, Eq. (23) is reduced to $\frac{1}{2}\sum\mathbb{E}[X_i]$, and Eq. (24) is reduced to $\frac{1}{2}\mathbb{E}_{\text{sample-path}}[\sum X_i]$. Therefore, the variance of the Monte-Carlo estimation of Eq. (23) becomes $\frac{1}{2}\sum\text{Var}(X_i)$, whereas the variance of the Monte-Carlo estimation of Eq. (24) becomes

$$
\frac{1}{2}\text{Var}\Big(\sum X_i\Big) = \frac{1}{2}\Big[\sum\text{Var}(X_i) + 2\sum\text{Cov}(X_i, X_j)\Big],
$$

where $\text{Cov}(X_i, X_j)$ represents the covariance of two random variables $X_i$ and $X_j$. Table 10 represents the ratio of these two variances,

$$
\text{Ratio} := \frac{\text{Var}(\sum X_i)}{\sum\text{Var}(X_i)} = \frac{\sum\text{Var}(X_i) + 2\sum\text{Cov}(X_i, X_j)}{\sum\text{Var}(X_i)} = 1 + 2\frac{\sum\text{Cov}(X_i, X_j)}{\sum\text{Var}(X_i)},
$$

and it shows that the DiffFlow loss has prohibitively large variance as $N$ increases, compared to the INDM loss, which computes its Monte-Carlo estimation in spirit of Eq. (23) with $N = \infty$.

Note that throughout our argument, we have omitted the prior and reconstruction terms on the variational bounds in this section.

Table 10: The variance ratio between the variances of the analytic transition probability-based estimation of Eq. (23) and the sample-based estimation of Eq. (24).

| | Number of Random Variables ($N$) | | | | |
|---|---|---|---|---|---|
| | 1 | 10 | 100 | 1000 | 10000 |
| Estimation Variance Ratio | 1.00 | 1.02 | 2.08 | 16.68 | 76.08 |

### D.3 Schrödinger Bridge Problem (SBP)

Schrödinger Bridge Problem (SBP) [14–16] has recently been highlighted in machine learning for its connection to the score-based diffusion model. Schrödinger Bridge Problem is a bi-constrained problem of

$$\min_{\boldsymbol{\rho}\in\mathcal{P}(p_r,\pi)} D_{KL}(\boldsymbol{\rho}\|\boldsymbol{\mu}),$$

where $\mathcal{P}(p_r,\pi)$ is a family of path measure with bi-constraints of $p_r$ and $\pi$ as its marginal distributions at $t = 0$ and $t = T$, respectively, and $\boldsymbol{\mu}$ is a reference path measure that is governed by

$$d\mathbf{x}_t = \mathbf{f}(\mathbf{x}_t, t)\, dt + g(t)\, d\mathbf{w}_t, \quad \mathbf{x}_0 \sim p_r. \tag{28}$$

As the KL divergence becomes infinite if the diffusion coefficient of $\boldsymbol{\rho}$ is not equal to $g(t)$ (because quadratic variations of $\boldsymbol{\mu}$ and $\boldsymbol{\rho}$ becomes different), SBP is equivalently formulated as

$$\min_{\boldsymbol{\rho}\in\mathcal{P}(p_r,\pi)} D_{KL}(\boldsymbol{\rho}\|\boldsymbol{\mu}) = \min_{\boldsymbol{\rho}_{\mathbf{v}}\in\mathcal{P}(p_r,\pi)} D_{KL}(\boldsymbol{\rho}_{\mathbf{v}}\|\boldsymbol{\mu}),$$

where the path measure $\boldsymbol{\rho}_{\mathbf{v}} \in \mathcal{P}(p_r,\pi)$ follows the SDE of

$$d\mathbf{x}_t = \left[\mathbf{f}(\mathbf{x}_t, t) + g^2(t)\mathbf{v}(\mathbf{x}_t, t)\right] dt + g(t)\mathbf{w}_t. \tag{29}$$

From the Girsanov theorem and the Martingale property [51], we have

$$D_{KL}(\boldsymbol{\rho}_{\mathbf{v}}\|\boldsymbol{\mu}) = \frac{1}{2}\int_0^T g^2(t)\mathbb{E}_{\boldsymbol{\rho}_{\mathbf{v}}}[\|\mathbf{v}(\mathbf{x}_t,t)\|_2^2]\, dt + D_{KL}(\pi\|p_T),$$

where $p_T$ is the marginal distribution of $\boldsymbol{\mu}$ at $t = T$. If $\mathcal{V}(p_r, \pi)$ is the space of all vector fields $\mathbf{v}$ of which forward SDE with Eq. (29) satisfies the boundary conditions, then SBP is equivalent to

$$\min_{\boldsymbol{\nu}\in\mathcal{P}(p_r,\pi)} D_{KL}(\boldsymbol{\nu}\|\boldsymbol{\mu}) = \min_{\mathbf{v}\in\mathcal{V}(p_r,\pi)} \frac{1}{2}\int_0^T g^2(t)\mathbb{E}_{\boldsymbol{\rho}_{\mathbf{v}}}[\|\mathbf{v}(\mathbf{x}_t,t)\|_2^2]\, dt, \tag{30}$$

where $\boldsymbol{\rho}_{\mathbf{v}}$ is the associated path measure of Eq. (29). Eq. (30) interprets the solution of SBP as the least energy (weighted by $g^2$) of the auxiliary vector field ($\mathbf{v}$) among admissible space of vector fields ($\mathcal{V}(p_r,\pi)$). Hence, if $\boldsymbol{\mu} \in \mathcal{P}(p_r,\pi)$, then the trivial vector field, $\mathbf{v} \equiv 0$, is the solution of SBP. When the reference SDE of Eq. (28) is one of the family of linear SDEs, such as VESDE or VPSDE, then $\boldsymbol{\mu} \notin \mathcal{P}(p_r,\pi)$, so the trivial vector field is not the solution of SBP, anymore. Instead, $\boldsymbol{\mu}$'s ending variable is close enough to $\pi$ (e.g., $D_{KL}(p_T\|\pi) \approx 10^{-5}$ in bpd scale [8]), so the closest path measure in $\mathcal{V}(p_r,\pi)$ to $\boldsymbol{\mu}$ is nearly identical to a trivial vector field, $\mathbf{v}^* \approx 0$, and the nonlinearity of SBP is limited.

Chen et al. [16] connects the optimal solution of SBP with PDEs. At the optimal point, if we denote by $\boldsymbol{\rho}^* = \arg\min_{\boldsymbol{\rho}\in\mathcal{P}(p_r,\pi)} D_{KL}(\boldsymbol{\rho}\|\boldsymbol{\mu})$, then this *optimal* diffusion process follows a forward diffusion SDE [16] of

$$d\mathbf{x}_t = \left[\mathbf{f}(\mathbf{x}_t, t) + g^2(t)\nabla_{\mathbf{x}_t} \log \Psi(\mathbf{x}_t, t)\right] dt + g(t)\, d\mathbf{w}_t, \quad \mathbf{x}_0 \sim p_r,$$

with the corresponding reverse diffusion as

$$d\mathbf{x}_t = \left[\mathbf{f}(\mathbf{x}_t, t) - g^2(t)\nabla_{\mathbf{x}_t} \log \hat{\Psi}(\mathbf{x}_t, t)\right] d\bar{t} + g(t)\, d\bar{\mathbf{w}}_t, \quad \mathbf{x}_T \sim \pi,$$

where $\Psi(\mathbf{x}_t, t)$ and $\hat{\Psi}(\mathbf{x}_t, t)$ are the solutions of a system of PDEs [21]:

$$\begin{aligned} \frac{\partial \Psi}{\partial t} &= -\nabla_{\mathbf{x}_t}\Psi^T\mathbf{f} - \frac{1}{2}\text{tr}(g^2\nabla_{\mathbf{x}_t}^2 \Psi) \\ \frac{\partial \hat{\Psi}}{\partial t} &= -\text{div}(\hat{\Psi}\mathbf{f}) + \frac{1}{2}\text{tr}(g^2\nabla_{\mathbf{x}_t}^2 \hat{\Psi}), \end{aligned} \tag{31}$$

such that $\Psi(\mathbf{x}_0, 0)\hat{\Psi}(\mathbf{x}_0, 0) = p_r(\mathbf{x}_0)$ and $\Psi(\mathbf{x}_T, T)\hat{\Psi}(\mathbf{x}_T, T) = \pi(\mathbf{x}_T)$. With $\Psi$ and $\hat{\Psi}$, the forward diffusion SDE ends exactly at $\pi$, and the corresponding reverse SDE ends at $p_r$. Therefore, SBP is equivalent to solve the system of PDEs given by Eq. (31).

Chen et al. [16] solves the system of coupled PDEs with Eq. (31) using a theory of forward-backward SDEs, which requires a deep understanding of PDE theory. SB-FBSDE [16] uses the fact that the solution $(\Psi, \hat{\Psi})$ of Hopf-Cole transform in Eq. (31) is derived from the solution of the forward-backward SDEs of

$$
\begin{cases}
\mathrm{d}\mathbf{x}_t = \left[\mathbf{f}(\mathbf{x}_t, t) + g(t)\mathbf{z}_t(\mathbf{x}_t, t)\right]\mathrm{d}t + g(t)\,\mathrm{d}\mathbf{w}_t \\
\mathrm{d}\mathbf{y}_t = \frac{1}{2}(\mathbf{z}_t^T\mathbf{z}_t)(\mathbf{x}_t, t)\,\mathrm{d}t + \mathbf{z}_t^T(\mathbf{x}_t, t)\,\mathrm{d}\mathbf{w}_t \\
\mathrm{d}\hat{\mathbf{y}}_t = \left[\frac{1}{2}(\hat{\mathbf{z}}_t^T\hat{\mathbf{z}}_t)(\mathbf{x}_t, t) + \mathrm{div}\left(g(t)\hat{\mathbf{z}}_t(\mathbf{x}_t, t) - \mathbf{f}(\mathbf{x}_t, t)\right) + (\hat{\mathbf{z}}_t^T\mathbf{z}_t)(\mathbf{x}_t, t)\right]\mathrm{d}t + \hat{\mathbf{z}}_t^T(\mathbf{x}_t, t)\,\mathrm{d}\mathbf{w}_t,
\end{cases}
\tag{32}
$$

where the boundary conditions are given by $\mathbf{x}(0) = \mathbf{x}_0$ and $\mathbf{y}_T + \hat{\mathbf{y}}_T = \log\pi(\mathbf{x}_T)$. The solution of the above system of forward-backward SDEs satisfies $\mathbf{z}_t(\mathbf{x}_t, t) = g(t)\nabla\log\Psi(\mathbf{x}_t, t)$ and $\hat{\mathbf{z}}_t(\mathbf{x}_t, t) = g(t)\nabla\log\hat{\Psi}(\mathbf{x}_t, t)$, where $(\Psi, \hat{\Psi})$ is the solution of Eq. (31). SB-FBSDE parametrizes $(\mathbf{z}_t, \hat{\mathbf{z}}_t)$ as $\boldsymbol{\theta}$ and $\boldsymbol{\phi}$, and it estimates the solution $(\mathbf{z}_t, \hat{\mathbf{z}}_t)$ of Eq. (32) from MLE training of the log-likelihood $\log p_{\boldsymbol{\phi}, \boldsymbol{\theta}}(\mathbf{x}_0)$.

Other than the PDE-driven approach [16], SBP has been traditionally solved via Iterative Proportional Fitting (IPF) [52]. Concretely, suppose

$$
\mathrm{d}\mathbf{x}_t^{\boldsymbol{\phi}} = \left[\mathbf{f}(\mathbf{x}_t^{\boldsymbol{\phi}}, t) + g^2(t)\mathbf{s}_{\boldsymbol{\phi}}(\mathbf{x}_t^{\boldsymbol{\phi}}, t)\right]\mathrm{d}t + g(t)\,\mathrm{d}\mathbf{w}_t, \quad \mathbf{x}_0^{\boldsymbol{\phi}} \sim p_r,
\tag{33}
$$

is a forward diffusion with a parametrized vector field of $\mathbf{s}_{\boldsymbol{\phi}}$, and

$$
\mathrm{d}\mathbf{x}_t^{\boldsymbol{\theta}} = \left[\mathbf{f}(\mathbf{x}_t^{\boldsymbol{\theta}}, t) - g^2(t)\mathbf{s}_{\boldsymbol{\theta}}(\mathbf{x}_t^{\boldsymbol{\theta}}, t)\right]\mathrm{d}\bar{t} + g^2(t)\bar{\mathbf{w}}_t, \quad \mathbf{x}_T^{\boldsymbol{\theta}} \sim \pi,
$$

is a generative diffusion with a parametrized vector field of $\mathbf{s}_{\boldsymbol{\theta}}$. Then, IPF get its optimal vector fields by alternatively solving below half-bridge problems

$$
\boldsymbol{\nu}_{\boldsymbol{\phi}_n} = \arg\min_{\boldsymbol{\nu}_{\boldsymbol{\phi}} \in \mathcal{P}(p_r, \cdot)} D_{KL}(\boldsymbol{\nu}_{\boldsymbol{\phi}} \| \boldsymbol{\nu}_{\boldsymbol{\theta}_{n-1}}),
\tag{34}
$$

$$
\boldsymbol{\nu}_{\boldsymbol{\theta}_n} = \arg\min_{\boldsymbol{\nu}_{\boldsymbol{\theta}} \in \mathcal{P}(\cdot, \pi)} D_{KL}(\boldsymbol{\nu}_{\boldsymbol{\theta}} \| \boldsymbol{\nu}_{\boldsymbol{\phi}_n}),
\tag{35}
$$

where the convergence of $\boldsymbol{\nu}_{\boldsymbol{\phi}^n} \to \boldsymbol{\nu}_{\boldsymbol{\phi}^*}$ and $\boldsymbol{\mu}_{\boldsymbol{\theta}^n} \to \boldsymbol{\nu}_{\boldsymbol{\theta}^*}$ is guaranteed in De Bortoli et al. [15]. Here, analogously, $\mathcal{P}(\cdot, \pi)$ is a family of path measure with $\pi$ as its marginal distribution at $t = T$. Notably, each of the half-bridge problem is a diffusion problem with the KL divergence replaced with the reverse KL divergence. Since SBP learns the forward SDE, sampling particle paths is expensive as it requires to solve an SDE numerically, so the training of IPF is slow.

# E    Correction of Density Estimation Metrics of Diffusion Models with Time Truncation

## E.1    Equivalent Reverse SDEs

Throughout Section E, the diffusion process is assumed to follow a SDE of $\mathrm{d}\mathbf{x}_t = \mathbf{f}(\mathbf{x}_t, t)\,\mathrm{d}t + g(t)\,\mathrm{d}\mathbf{w}_t$ because the below argument is generally applicable for any continuous diffusion models. For INDM, we apply the below argument on the latent space, which has a linear drift term. Let $\mathrm{d}\mathbf{x}_t = \left[\mathbf{f}(\mathbf{x}_t, t) - \frac{1+\lambda^2}{2}g^2(t)\nabla_{\mathbf{x}_t}\log p_t^\lambda(\mathbf{x}_t)\right]\mathrm{d}\bar{t} + \lambda g(t)\bar{\mathbf{w}}_t$ be the reverse SDEs starting from $p_T$, where $p_t^\lambda$ is the probability law of the solution at $t$. Then, the reverse Kolmogorov equation (or Fokker-Planck equation) becomes

$$
\frac{\partial p_t^\lambda(\mathbf{x}_t, t)}{\partial t} = -\sum_{i=1}^d \frac{\partial}{\partial x_i}\left(\left[f_i(\mathbf{x}_t, t) - \frac{1+\lambda^2}{2}g^2(t)\left(\nabla_{\mathbf{x}_t}\log p_t^\lambda(\mathbf{x}_t, t)\right)_i\right]p_t^\lambda(\mathbf{x}_t, t)\right)
$$

$$
- \frac{\lambda^2 g^2(t)}{2}\sum_{i=1}^d \frac{\partial^2}{\partial x_i^2}\left[p_t^\lambda(\mathbf{x}_t, t)\right]
$$

$$= -\sum_{i=1}^{d} \frac{\partial}{\partial x_i} \left( f_i(\mathbf{x}_t, t) p_t^\lambda(\mathbf{x}_t, t) - \frac{1+\lambda^2}{2} g^2(t) \frac{\partial p_t^\lambda(\mathbf{x}_t, t)}{\partial x_i} \right)$$

$$- \frac{\lambda^2 g^2(t)}{2} \sum_{i=1}^{d} \frac{\partial^2}{\partial x_i^2} \left[ p_t^\lambda(\mathbf{x}_t, t) \right]$$

$$= -\sum_{i=1}^{d} \frac{\partial}{\partial x_i} \left[ f_i(\mathbf{x}_t, t) p_t^\lambda(\mathbf{x}_t, t) \right] + \frac{1}{2} g^2(t) \sum_{i=1}^{d} \frac{\partial^2}{\partial x_i^2} \left[ p_t^\lambda(\mathbf{x}_t, t) \right],$$

which is independent of $\lambda$. Therefore, it satisfies $p_t^\lambda = p_t^{\lambda'}$ for any $\lambda \neq \lambda'$.

For any $\lambda \in [0, 1]$, the generative SDE is constructed by plugging $\mathbf{s}_{\boldsymbol{\theta}}(\mathbf{x}_t, t)$ in place of $\nabla_{\mathbf{x}_t} \log p_t^\lambda(\mathbf{x}_t)$ in the reverse SDE as $\mathrm{d}\mathbf{x}_t = \left[ \mathbf{f}(\mathbf{x}_t, t) - \frac{1+\lambda^2}{2} g^2(t) \mathbf{s}_{\boldsymbol{\theta}}(\mathbf{x}_t, t) \right] \mathrm{d}\bar{t} + \lambda g(t) \bar{\mathbf{w}}_t$. Suppose we denote $p_t^{\lambda, \boldsymbol{\theta}}$ as the marginal distribution of the model at $t$. Then, the generative SDEs with different $\lambda$ have distinctive marginal distributions: $p_t^{\lambda, \boldsymbol{\theta}} \neq p_t^{\lambda', \boldsymbol{\theta}}$ for $\lambda \neq \lambda'$.

### E.2 Log-Likelihood for Diffusion Models with Time Truncation

Due to the unbounded score loss illustrated in [27], a diffusion model truncates the diffusion time to be $[\epsilon, 1]$ for small enough $\epsilon > 0$. However, since the small range of diffusion time contributes significant portion of the log-likelihood [27], the effect of truncation should be counted both on training and evaluation. To describe, as we have no knowledge on the score estimation at $t \in [0, \epsilon)$, we have estimate the data log-likelihood by using the variational inferecence:

$$\log p_0^{\lambda, \boldsymbol{\theta}}(\mathbf{x}_0) = \log \int p_0^{\lambda, \boldsymbol{\theta}}(\mathbf{x}_0, \mathbf{x}_\epsilon) \, \mathrm{d}\mathbf{x}_\epsilon$$

$$\geq \int p_{0\epsilon}(\mathbf{x}_\epsilon | \mathbf{x}_0) \log \frac{p_\epsilon^{\lambda, \boldsymbol{\theta}}(\mathbf{x}_\epsilon) p_{\epsilon 0}^{\boldsymbol{\theta}}(\mathbf{x}_0 | \mathbf{x}_\epsilon)}{p_{0\epsilon}(\mathbf{x}_\epsilon | \mathbf{x}_0)} \, \mathrm{d}\mathbf{x}_\epsilon$$

$$= \mathbb{E}_{p_{0\epsilon}(\mathbf{x}_\epsilon | \mathbf{x}_0)} \left[ \log p_\epsilon^{\lambda, \boldsymbol{\theta}}(\mathbf{x}_\epsilon) \right] + \mathbb{E}_{p_{0\epsilon}(\mathbf{x}_\epsilon | \mathbf{x}_0)} \left[ \log \frac{p_{\epsilon 0}^{\boldsymbol{\theta}}(\mathbf{x}_0 | \mathbf{x}_\epsilon)}{p_{0\epsilon}(\mathbf{x}_\epsilon | \mathbf{x}_0)} \right],$$

where $p_\epsilon^{\lambda, \boldsymbol{\theta}}$ is the generative distribution perturbed by $\epsilon$, and $p_{\epsilon 0}^{\boldsymbol{\theta}}(\mathbf{x}_0 | \mathbf{x}_\epsilon)$ is the reconstruction transition probability given $\mathbf{x}_\epsilon$. Then, we have

$$\mathbb{E}_{p_r(\mathbf{x}_0)} [-\log p_0^{\lambda, \boldsymbol{\theta}}(\mathbf{x}_0)] \leq \mathbb{E}_{\mathbf{x}_\epsilon} \left[ -\log p_\epsilon^{\lambda, \boldsymbol{\theta}}(\mathbf{x}_\epsilon) \right] - \mathbb{E}_{\mathbf{x}_0, \mathbf{x}_\epsilon} \left[ \log \frac{p_{\epsilon 0}^{\boldsymbol{\theta}}(\mathbf{x}_0 | \mathbf{x}_\epsilon)}{p_{0\epsilon}(\mathbf{x}_\epsilon | \mathbf{x}_0)} \right]$$

$$= D_{KL}(p_\epsilon \| p_\epsilon^{\lambda, \boldsymbol{\theta}}) - \mathbb{E}_{\mathbf{x}_0, \mathbf{x}_\epsilon} \left[ \log \frac{p_{\epsilon 0}^{\boldsymbol{\theta}}(\mathbf{x}_0 | \mathbf{x}_\epsilon)}{p_{0\epsilon}(\mathbf{x}_\epsilon | \mathbf{x}_0)} \right] + \mathcal{H}(p_\epsilon),$$

which is equivalent to

$$D_{KL}(p_r \| p_0^{\lambda, \boldsymbol{\theta}}) \leq D_{KL}(p_\epsilon \| p_\epsilon^{\lambda, \boldsymbol{\theta}}) - \mathbb{E}_{\mathbf{x}_0, \mathbf{x}_\epsilon} \left[ \log \frac{p_{\epsilon 0}^{\boldsymbol{\theta}}(\mathbf{x}_0 | \mathbf{x}_\epsilon)}{p_{0\epsilon}(\mathbf{x}_\epsilon | \mathbf{x}_0)} \right] + \mathcal{H}(p_\epsilon) - \mathcal{H}(p_r)$$

$$= D_{KL}(p_\epsilon \| p_\epsilon^{\lambda, \boldsymbol{\theta}}) + D_{KL} \left( p_r(\mathbf{x}_0) p_{0\epsilon}(\mathbf{x}_\epsilon | \mathbf{x}_0) \| p_\epsilon(\mathbf{x}_\epsilon) p_{\epsilon 0}^{\boldsymbol{\theta}}(\mathbf{x}_0 | \mathbf{x}_\epsilon) \right).$$

### E.3 NELBO Correction

Suppose $\boldsymbol{\mu}_\epsilon$ is the path measure of $\mathrm{d}\mathbf{x}_t = \mathbf{f}(\mathbf{x}_t, t) \, \mathrm{d}t + g(t) \, \mathrm{d}\mathbf{w}_t$ on $[\epsilon, T]$, and $\boldsymbol{\nu}_{\boldsymbol{\theta}, \epsilon}^\lambda$ is the path measure of $\mathrm{d}\mathbf{x}_t = \left[ \mathbf{f}(\mathbf{x}_t, t) - \frac{1+\lambda^2}{2} g^2(t) \nabla_{\mathbf{x}_t} \log p_t^\lambda(\mathbf{x}_t) \right] \mathrm{d}\bar{t} + \lambda g(t) \bar{\mathbf{w}}_t$ on $[\epsilon, T]$. Then, the continuous variational bound on the truncated diffusion model becomes

$$\mathbb{E}_{p_r(\mathbf{x}_0)} \left[ -\log p_0^{\lambda, \boldsymbol{\theta}}(\mathbf{x}_0) \right] \leq D_{KL}(p_\epsilon \| p_\epsilon^{\lambda, \boldsymbol{\theta}}) - \mathbb{E}_{\mathbf{x}_0, \mathbf{x}_\epsilon} \left[ \log \frac{p_{\epsilon 0}^{\boldsymbol{\theta}}(\mathbf{x}_0 | \mathbf{x}_\epsilon)}{p_{0\epsilon}(\mathbf{x}_\epsilon | \mathbf{x}_0)} \right] + \mathcal{H}(p_\epsilon)$$

$$\leq D_{KL}(\boldsymbol{\mu}_\epsilon \| \boldsymbol{\nu}_{\boldsymbol{\theta}, \epsilon}^\lambda) - \mathbb{E}_{\mathbf{x}_0, \mathbf{x}_\epsilon} \left[ \log \frac{p_{\epsilon 0}^{\boldsymbol{\theta}}(\mathbf{x}_0 | \mathbf{x}_\epsilon)}{p_{0\epsilon}(\mathbf{x}_\epsilon | \mathbf{x}_0)} \right] + \mathcal{H}(p_\epsilon)$$

$$= \frac{1}{2} \frac{(1+\lambda^2)^2}{4\lambda^2} \int_\epsilon^T g^2(t) \mathbb{E}_{\mathbf{x}_t} \left[ \| \mathbf{s}_{\boldsymbol{\theta}}(\mathbf{x}_t, t) - \log p_t(\mathbf{x}_t) \|_2^2 \right] \mathrm{d}t - \mathbb{E}_{\mathbf{x}_T} \left[ \log \pi(\mathbf{x}_T) \right]$$

$$-\mathbb{E}_{\mathbf{x}_0,\mathbf{x}_\epsilon}\left[\log\frac{p_{\epsilon 0}^{\boldsymbol{\theta}}(\mathbf{x}_0|\mathbf{x}_\epsilon)}{p_{0\epsilon}(\mathbf{x}_\epsilon|\mathbf{x}_0)}\right]+\mathcal{H}(p_\epsilon)-\mathcal{H}(p_T)$$

$$=\frac{1}{2}\int_\epsilon^T\mathbb{E}_{\mathbf{x}_0,\mathbf{x}_t}\left[\frac{(1+\lambda^2)^2}{4\lambda^2}g^2(t)\|\log p_{0t}(\mathbf{x}_t|\mathbf{x}_0)-\mathbf{s}_{\boldsymbol{\theta}}(\mathbf{x}_t,t)\|_2^2-g^2(t)\|\nabla_{\mathbf{x}_t}\log p_{0t}(\mathbf{x}_t|\mathbf{x}_0)\|_2^2\right.$$

$$\left.-2\nabla_{\mathbf{x}_t}\cdot\mathbf{f}(\mathbf{x}_t,t)\right]\mathrm{d}t-\mathbb{E}_{\mathbf{x}_T}\left[\log\pi(\mathbf{x}_T)\right]-\mathbb{E}_{\mathbf{x}_0,\mathbf{x}_\epsilon}\left[\log\frac{p_{\epsilon 0}^{\boldsymbol{\theta}}(\mathbf{x}_0|\mathbf{x}_\epsilon)}{p_{0\epsilon}(\mathbf{x}_\epsilon|\mathbf{x}_0)}\right],$$

where $\mathcal{H}(p_\epsilon)-\mathcal{H}(p_T)$ is derived to be $-\frac{1}{2}\int_\epsilon^T\mathbb{E}\left[g^2\|\nabla\log p_{0t}\|_2^2+2\nabla\cdot\mathbf{f}\right]\mathrm{d}t$ by Theorem 4 of Song et al. [11]. The residual term, $\mathbb{E}_{p_r(\mathbf{x}_0)p_{0\epsilon}(\mathbf{x}_\epsilon|\mathbf{x}_0)}\left[\log\frac{p_{\epsilon 0}^{\boldsymbol{\theta}}(\mathbf{x}_0|\mathbf{x}_\epsilon)}{p_{0\epsilon}(\mathbf{x}_\epsilon|\mathbf{x}_0)}\right]$, has been ignored both on training and evaluation in previous research. Therefore, we report the *correct* NELBO (denoted by *w/ residual* in the main paper) by counting the residual term $\mathbb{E}_{p_r(\mathbf{x}_0)p_{0\epsilon}(\mathbf{x}_\epsilon|\mathbf{x}_0)}\left[\log\frac{p_{\epsilon 0}^{\boldsymbol{\theta}}(\mathbf{x}_0|\mathbf{x}_\epsilon)}{p_{0\epsilon}(\mathbf{x}_\epsilon|\mathbf{x}_0)}\right]$ into account. Note that $\frac{(1+\lambda^2)^2}{4\lambda^2}$ is minimized when $\lambda=1$, so our reported NELBO is based on the generative SDE at $\lambda=1$.

### E.4 NLL Correction

NLL of the generative SDE can be computed through the Feynman-Kac formula [12] by

$$p_{\boldsymbol{\theta},\epsilon}^\lambda(\mathbf{x}_\epsilon)=\mathbb{E}_{\{\mathbf{x}_t\}_{t>\epsilon}|\mathbf{x}_\epsilon}\left[\pi(\mathbf{x}_T)\exp\left(-\int_\epsilon^T\mathrm{tr}\left(\nabla_{\mathbf{x}_t}\left[\mathbf{f}(\mathbf{x}_t,t)-\frac{1+\lambda^2}{2}g^2(t)\mathbf{s}_{\boldsymbol{\theta}}(\mathbf{x}_t,t)\right]\right)\mathrm{d}t\right)\right].\tag{36}$$

However, the expectation is intractable because there are infinitely-many sample paths. Fortunately, the sample variance diminishes as $\lambda\to 0$, and the generative SDE collapses to a generative ODE when $\lambda=0$ [1], i.e., the generative SDE of $\lambda=0$ becomes

$$\mathrm{d}\mathbf{x}_t=\left[\mathbf{f}(\mathbf{x}_t,t)-\frac{1}{2}g^2(t)\mathbf{s}_{\boldsymbol{\theta}}(\mathbf{x}_t,t)\right]\mathrm{d}\bar{t},$$

which corresponds to the generative ODE of forward time as

$$\mathrm{d}\mathbf{x}_t=\left[\mathbf{f}(\mathbf{x}_t,t)-\frac{1}{2}g^2(t)\mathbf{s}_{\boldsymbol{\theta}}(\mathbf{x}_t,t)\right]\mathrm{d}t.\tag{37}$$

Then, the sample path becomes deterministic, and the expectation in Eq. (36) is degenerated as the single sample path of ODE with Eq. (37) starting $\mathbf{x}_\epsilon$. The instantaneous change-of-variable formula [1], which is a collapsed Feynman-Kac formula in Eq. (36), guarantees that there is a corresponding ODE of Eq. (37) as

$$\frac{\mathrm{d}\log p_t^{0,\boldsymbol{\theta}}(\mathbf{y}_t)}{\mathrm{d}t}=-\mathrm{tr}\left(\nabla_{\mathbf{y}_t}\left[\mathbf{f}(\mathbf{y}_t,t)-\frac{1}{2}g^2(t)\mathbf{s}_{\boldsymbol{\theta}}(\mathbf{y}_t,t)\right]\right).\tag{38}$$

From the fact that the reverse SDEs have the identical marginal distributions described in Section E.1, we approximate the model log-likelihood at $\lambda=1$ by the log-likelihood at $\lambda=0$ at the expense of slight difference between the model distributions of different $\lambda$s. When computing the model log-likelihood at $\lambda=0$, we integrate the ODE of Eq. (38) over $[\epsilon,1]$ using an ODE solver, such as the Runge-Kutta 45 method [25].

There are minor subtleties in computing the log-likelihood at $\lambda=0$ that significantly affects to bpd evaluation. To the best of our knowledge, all the current practice on continuous diffusion models computes bpd by integrating

$$\frac{\mathrm{d}\log p_t^{0,\boldsymbol{\theta}}(\mathbf{y}_t)}{\mathrm{d}t}=-\mathrm{tr}\left(\nabla_{\mathbf{y}_t}\left[\mathbf{f}(\mathbf{y}_t,t)-\frac{1}{2}g^2(t)\mathbf{s}_{\boldsymbol{\theta}}(\mathbf{y}_t,t)\right]\right),$$

on $t\in[\epsilon,T]$, where $\{\mathbf{y}_t\}_{t=\epsilon}^T$ is a sample path starting from $\mathbf{y}_\epsilon:=\mathbf{x}_0$. This is equivalent of computing $\log p_\epsilon^{0,\boldsymbol{\theta}}(\mathbf{x}_0)$. However, strarting from $\mathbf{x}_0$ incurs large discrepancy on the NLL output, compared to starting from instead of $\mathbf{x}_\epsilon$. Since the integration is on $[\epsilon,1]$, the starting variable should follow $\mathbf{x}_\epsilon$, which is a slightly perturbed variable.

Table 11: The difference of the integration with different initial points of $\mathbf{x}_\epsilon$ and $\mathbf{x}_0$ on DDPM++ (VP, NLL). The difference increases by $\epsilon$.

| | $\epsilon$ | | | |
| --- | --- | --- | --- | --- |
| | $10^{-2}$ | $10^{-3}$ | $10^{-4}$ | $10^{-5}$ |
| $\mathbb{E}_{\mathbf{x}_\epsilon}\big[-\log p_\epsilon^{\boldsymbol{\theta}}(\mathbf{x}_\epsilon)\big] - \mathbb{E}_{\mathbf{x}_0}\big[-\log p_\epsilon^{\boldsymbol{\theta}}(\mathbf{x}_0)\big]$ | 1.13 | 0.73 | 0.24 | 0.05 |

To fix this subtlety, we solve the below alternative differential equation of

$$\frac{\mathrm{d}\log p_t^{0,\boldsymbol{\theta}}}{\mathrm{d}t} = -\mathrm{tr}\bigg(\nabla_{\mathbf{y}_t}\Big[\mathbf{f}(\mathbf{y}_t,t) - \frac{1}{2}g^2(t)\mathbf{s}_{\boldsymbol{\theta}}(\mathbf{y}_t,t)\Big]\bigg), \tag{39}$$

on $t \in [\epsilon, T]$, where $\{\mathbf{y}_t^0\}_{t=\epsilon}^T$ is a sample path starting from $\mathbf{y}_\epsilon := \mathbf{x}_\epsilon$. By replacing the initial value to $\mathbf{x}_\epsilon$ from $\mathbf{x}_0$, we could correctly compute $\log p_\epsilon^{\boldsymbol{\theta}}(\mathbf{x}_\epsilon)$. Table 11 presents the difference of $\mathbb{E}_{\mathbf{x}_\epsilon}\big[-\log p_\epsilon^{0,\boldsymbol{\theta}}(\mathbf{x}_\epsilon)\big] - \mathbb{E}_{\mathbf{x}_0}\big[-\log p_\epsilon^{0,\boldsymbol{\theta}}(\mathbf{x}_0)\big]$ with various $\epsilon$. We report the *correct* NLL as

$$\mathbb{E}_{\mathbf{x}_\epsilon}\big[-\log p_\epsilon^{0,\boldsymbol{\theta}}(\mathbf{x}_\epsilon)\big] - \mathbb{E}_{\mathbf{x}_0,\mathbf{x}_\epsilon}\bigg[\log \frac{p_{\epsilon 0}^{\boldsymbol{\theta}}(\mathbf{x}_0|\mathbf{x}_\epsilon)}{p_{0\epsilon}(\mathbf{x}_\epsilon|\mathbf{x}_0)}\bigg],$$

where $\log p_\epsilon^{0,\boldsymbol{\theta}}(\mathbf{x}_\epsilon)$ is computed based on the initial point of $\mathbf{x}_\epsilon$ when $\lambda = 0$.

### E.5 Calculating the Residual Term

This section calculates the residual term, $\mathbb{E}_{p_r(\mathbf{x}_0)p_{0\epsilon}(\mathbf{x}_\epsilon|\mathbf{x}_0)}\big[\log \frac{p_{\epsilon 0}^{\boldsymbol{\theta}}(\mathbf{x}_0|\mathbf{x}_\epsilon)}{p_{0\epsilon}(\mathbf{x}_\epsilon|\mathbf{x}_0)}\big]$. The transition probability of $p_{0\epsilon}(\mathbf{x}_\epsilon|\mathbf{x}_0)$ is the Gaussian distribution of $\mathcal{N}(\mathbf{x}_\epsilon; \mu(\epsilon)\mathbf{x}_0, \sigma^2(\epsilon)\mathbf{I})$ if $\mathbf{f}(\mathbf{x}_t,t) = -\frac{1}{2}\beta(t)\mathbf{x}_t$ with

$$\mu(\epsilon) = e^{-\frac{1}{2}\int_0^\epsilon \beta(s)\,\mathrm{d}s} \text{ and } \sigma^2(\epsilon) = \bigg(\frac{g^2(t)}{\beta(t)} - \frac{g^2(0)}{\beta(0)} + 1 - e^{-\int_0^t \beta(s)\,\mathrm{d}s}\bigg),$$

see Appendix A.1 of Kim et al. [27] for detailed computation. On the other hand, the generative distribution of $p_{\boldsymbol{\theta},\epsilon 0}(\mathbf{x}_0|\mathbf{x}_\epsilon)$ is assumed to be a Gaussian distribution of $\mathcal{N}\big(\mathbf{x}_0; \boldsymbol{\mu}_{\boldsymbol{\theta},\epsilon 0}(\mathbf{x}_\epsilon), \sigma^2_{\boldsymbol{\theta},\epsilon 0}\mathbf{I}\big)$, where $\boldsymbol{\mu}_{\boldsymbol{\theta},\epsilon 0}(\mathbf{x}_\epsilon) = \frac{1}{\mu(\epsilon)}\big(\mathbf{x}_\epsilon + \sigma^2(\epsilon)\mathbf{s}_{\boldsymbol{\theta}}(\mathbf{x}_\epsilon,\epsilon)\big)$ [28]. Then, we have

$$\mathbb{E}_{p_{0\epsilon}(\mathbf{x}_\epsilon|\mathbf{x}_0)}\bigg[\log \frac{p_{\epsilon 0}^{\boldsymbol{\theta}}(\mathbf{x}_0|\mathbf{x}_\epsilon)}{p_{0\epsilon}(\mathbf{x}_\epsilon|\mathbf{x}_0)}\bigg]$$
$$= \log \mu(\epsilon) - \frac{1}{2\sigma^2_{\boldsymbol{\theta},\epsilon 0}(\epsilon)}\mathbb{E}_{p_{0\epsilon}(\mathbf{x}_\epsilon|\mathbf{x}_0)}\bigg[\Big\|\mathbf{x}_0 - \frac{1}{\mu(\epsilon)}\big(\mathbf{x}_t + \sigma^2(\epsilon)\mathbf{s}_{\boldsymbol{\theta}}(\mathbf{x}_\epsilon,\epsilon)\big)\Big\|_2^2\bigg] + \frac{1}{2}.$$

We could approximate the variance of $p_{\epsilon 0}^{\boldsymbol{\theta}}(\mathbf{x}_0|\mathbf{x}_\epsilon)$ to be the variance of $p_{\epsilon 0}(\mathbf{x}_0|\mathbf{x}_\epsilon)$, if $p_{\epsilon 0}(\mathbf{x}_0|\mathbf{x}_\epsilon)$ is derived as a closed-form. For that, let us assume $\mathbf{x}_0 \sim \mathcal{N}(0,\sigma^2)$. Then,

$$p(\mathbf{x}_0,\mathbf{x}_\epsilon) = p(\mathbf{x}_0)p_{0\epsilon}(\mathbf{x}_\epsilon|\mathbf{x}_0)$$
$$\propto \exp\bigg(-\frac{\|\mathbf{x}_0\|_2^2}{2\sigma^2} - \frac{\|\mathbf{x}_\epsilon - \mu(\epsilon)\mathbf{x}_0\|_2^2}{2\sigma^2(\epsilon)}\bigg)$$
$$= \exp\bigg(-\frac{1}{2}\Big(\frac{1}{\sigma^2} + \frac{\mu^2(\epsilon)}{\sigma^2(\epsilon)}\Big)\Big\|\mathbf{x}_0 - \frac{\mu(\epsilon)\sigma^2}{\sigma^2(\epsilon) + \mu^2(\epsilon)\sigma^2}\mathbf{x}_\epsilon\Big\|_2^2 + O(\|\mathbf{x}_\epsilon\|_2^2)\bigg).$$

Therefore, $p_{\epsilon 0}(\mathbf{x}_0|\mathbf{x}_\epsilon) = \mathcal{N}\big(\mathbf{x}_0\big| \frac{\mu(\epsilon)\sigma^2}{\sigma^2(\epsilon)+\mu^2(\epsilon)\sigma^2}\mathbf{x}_\epsilon, 1/(\frac{1}{\sigma^2} + \frac{\mu^2(\epsilon)}{\sigma^2(\epsilon)})\big)$. When $\sigma$ is sufficiently large compared to $\frac{\sigma^2(\epsilon)}{\mu^2(\epsilon)}$, the variance of $p_{\epsilon 0}(\mathbf{x}_0|\mathbf{x}_\epsilon)$ is approximately $\frac{\sigma^2(\epsilon)}{\mu^2(\epsilon)}$. Now, if $\mathbf{x}_0 \sim p_r$, then the variance of $\mathbf{x}_0$ is large enough compared to $\frac{\sigma^2(\epsilon)}{\mu^2(\epsilon)}$, so we could approximate $\sigma^2_{\boldsymbol{\theta},\epsilon 0}(\epsilon)$ to be $\frac{\sigma^2(\epsilon)}{\mu^2(\epsilon)}$. Note that DDPM [8] assumes the variance to be $\sigma^2(\epsilon)$. We compute the residual term with $\frac{\sigma^2(\epsilon)}{\mu^2(\epsilon)}$ variance for both VESDE and VPSDE. Note that this residual term is inspired from the released code of Song et al. [11].

Table 12: Despite of our implementation is built deeply based on Song et al. [1], our pytorch implementation and the jax implementation of Song et al. [11] differs in their final performances.

| SDE | Model | NLL | | NELBO | | Gap | | FID |
|-----|-------|-----|---|-------|---|-----|---|-----|
| | | after | before | w/ residual | w/o residual | after | before | ODE |
| VP | DDPM++ (NLL, reported) [11] | - | 2.95 | - | 3.08 | - | 0.13 | 6.03 |
| | DDPM++ (NLL, ours) | 3.03 | 2.97 | 3.13 | 3.11 | 0.10 | 0.14 | 6.70 |
| | INDM (NLL) | 2.98 | 2.95 | 2.98 | 2.97 | 0.00 | 0.02 | 6.01 |

# F Experimental Details and Additional Results

## F.1 Model Architecture

**Diffusion Model** We implement two diffusion models as backbone: NCSN++ (VE) [1] and DDPM++ (VP) [1], where two backbones are one of the best performers in CIFAR-10 dataset. In our setting, NCSN++ assumes the score network with parametrization of $s_\theta(z_t, \log \sigma^2(t))$, where $\sigma^2(t) = \sigma_{min}^2 (\frac{\sigma_{max}}{\sigma_{min}})^{2t}$ is the variance of the transition probability $p_{0t}(z_t|z_0)$ with VESDE. As introduced in Song et al. [1], we use the Gaussian Fourier embeddings [53] to model the high frequency details across the temporal embedding. DDPM++ models the score network with parametrization of $\epsilon_\theta(z_t, t)$, which targets to estimate $-\sigma(t)\nabla_{z_t} \log p_t(z_t)$. We use the Transformer sinusoidal temporal embedding [54].

We use the U-Net [55] architecture for the score networks on both NCSN++ and DDPM++ based on [8]. We stack U-Net resblocks of up-and-down convolutions with skip connections that give input information to the output layer. Also, we follow Ho et al. [8] by applying the global attention at the resolution of $16 \times 16$. We use four U-Net resblocks with four feature map resolutions ($32 \times 32$ to $4 \times 4$). On CIFAR-10, we use four and eight resblocks for shallow and deep settings, respectively. The performances of shallow and deep models turn out to be insignificant, so we use four resblocks on CelebA. We provide the identical diffusion model structures to compare the baseline linear diffusion model and the INDM model in a fair setting.

**Flow Model** We build a normalizing flow model as follows. Ma et al. [24] uses the autoencoding structure of decouple the global information and the local representation. The compression encoder extracts the global information, and the invertible decoder is a conditional flow conditioned by the encoded latent representation. Ma et al. [24] utilizes a shallow network for the compressive encoder, and it applies Glow [56] for the invertible decoder. We empirically find that resnet-based flow network outperforms the Glow-based flow, so we replace Glow to ResFlow [23].

For the ResFlow, we drop three components from the original paper: 1) the activation normalization [56], 2) the batch normalization [57], and 3) the fully connected layers. For the activation function, we use the sine function [58] on quantitative comparisons in Section 7, and we use swish function [59] on qualitative analysis in otherwise sections including Section 6. With the sine activation, the training becomes more stable, and the FID performance is significantly improved while maintaining the NLL performance. For the multi-GPU training, we use the Neumann log-determinant gradient estimator, instead of the memory-efficient estimator [23].

## F.2 Experimental details

With the batch size of 128, we train the diffusion model with Exponential Moving Average (EMA) [28] rate of 0.9999, and we do not use EMA on our flow model. Using EMA on the flow model is advantageous on NLL at the expense of FID, and we build our model with emphasis on FID. We train the model by two step. The pre-training stage trains the diffusion model about five days with a flow model fixed as an identity function on four P40 GPUs with 96Gb GPU memory for all experiments. After the pre-training, we train both flow and diffusion networks about five days. In this stage, we apply the learning rate scheduling to boost the FID score. We initiate the learning rate after the sample generation performance is saturated. For the diffusion model, we drop the learning rate from $2 \times 10^{-4}$ to $10^{-5}$. For the flow model, we drop the learning rate from $10^{-3}$ to $10^{-5}$ for VPSDE and $5 \times 10^{-5}$ to $10^{-5}$ for VESDE.

VESDE and VPSDE have different training details. We apply INDM on VESDE with $\sigma_{min} = 10^{-2}$. Throughout the experiments, VESDE has $\sigma_{max} = 50$ on CIFAR-10 and $\sigma_{max} = 90$ on CelebA. On the other hand, VPSDE assumes $\beta(t) = \beta_{min} + (\beta_{max} - \beta_{min})t$ with $\beta_{min} = 0.1$ and $\beta_{max} = 20$. Both VESDE and VPSDE truncate the diffusion time on $[\epsilon, T]$ in order to stabilize the diffusion model [27], where $\epsilon = 10^{-5}$ and $T = 1$.

With all hyperparameters identical to Song et al. [11], however, we could not achieve the reported performance. Table 12 compares the reported performance and the model performance trained on out implementation, of which structure is heavily based on the released code of [1]. Due to the discrepancy between the reported and the regenerated performances, we compare our INDM to the regenerated performance as default to investigate the effect of nonlinear diffusion in a fair setting. Throughout the training, we used $4\times$ NVIDIA RTX 3090.

### F.2.1  Variance Reduction

**Flow Training** When we train the flow network with $\mathcal{L}\big(\{\mathbf{x}_t\}_{t=0}^T, g^2; \{\boldsymbol{\phi}, \boldsymbol{\theta}\}\big)$, this NELBO contains the integration of

$$\mathcal{L}\big(\{\mathbf{z}_t\}_{t=0}^T, g^2; \boldsymbol{\theta}\big) = \frac{1}{2} \int_0^T g^2(t) \mathbb{E}_{\mathbf{z}_0, \mathbf{z}_t}\big[\|\mathbf{s}_{\boldsymbol{\theta}}(\mathbf{z}_t, t)\|_2^2\big] \, \mathrm{d}t,$$

up to a constant. Suppose $\mathcal{L}_t\big(\{\mathbf{z}_t\}_{t=0}^T; \boldsymbol{\theta}\big)$ to be $\frac{1}{2}\mathbb{E}_{\mathbf{z}_0, \mathbf{z}_t}[\|\mathbf{s}_{\boldsymbol{\theta}}(\mathbf{z}_t, t) - \nabla_{\mathbf{z}_t} \log p_{0t}(\mathbf{z}_t|\mathbf{x}_0)\|_2^2]$. Previous works on diffusion models [60, 12, 11, 27] show that the estimation variance is largely reduced with the importance sampling, which could improve the model performance [11], and we apply this importance sampling throughout the experiments for NLL setting. Concretely, the importance sampling chooses an importance weight that is proportional to $\frac{g^2(t)}{\sigma^2(t)}$, and estimates the integration by $\mathcal{L}\big(\{\mathbf{z}_t\}_{t=0}^T, g^2; \boldsymbol{\theta}\big) = \int_0^T g^2(t)\mathcal{L}_t\big(\{\mathbf{z}_t\}_{t=0}^T; \boldsymbol{\theta}\big) \, \mathrm{d}t \approx \sum_{n=1}^N \sigma^2(t_n)\mathcal{L}_{t_n}\big(\{\mathbf{z}_t\}_{t=0}^T; \boldsymbol{\theta}\big)$, where $t_n$ is sampled from the importance distribution.

For VESDE, it satisfies $\beta(t) = 0$ and $g(t) = \sigma_{min}(\frac{\sigma_{max}}{\sigma_{min}})^t \sqrt{2\log\big(\frac{\sigma_{max}}{\sigma_{min}}\big)}$. The transition probability becomes $p_{-\infty t}(\mathbf{z}_t|\mathbf{z}_{-\infty}) = \mathcal{N}(\mathbf{z}_t; \mathbf{z}_{-\infty}, \sigma^2(t))$, where $\sigma^2(t) = \int_{-\infty}^t g^2(s) \, \mathrm{d}s = \sigma_{min}^2\big(\frac{\sigma_{max}}{\sigma_{min}}\big)^{2t}$. Since $\sigma^2(t)$ is proportional to $g^2(t)$ in VESDE, the importance weight follows the uniform distribution, and the importance sampling is equivalent with choosing the uniform $t$. This is why there is no experimetal setting of VE with NLL.

On the other hand, VPSDE satisfies $\beta(t) = \beta_{min} + (\beta_{max} - \beta_{min})t$ with $g(t) = \sqrt{\beta(t)}$. Then, the transition probability becomes $p_{0t}(\mathbf{z}_t|\mathbf{z}_0) = \mathcal{N}(\mathbf{z}_t; \mu(t)\mathbf{z}_t, \sigma^2(t)\mathbf{I})$, where $\mu(t) = e^{-\frac{1}{2}\int_0^t \beta(s) \, \mathrm{d}s}$ and $\sigma^2(t) = 1 - e^{-\int_0^t \beta(s) \, \mathrm{d}s}$. Thus, VPSDE has the importance weight of $\frac{g^2(t)}{\sigma^2(t)} = \frac{\beta(t)}{1 - e^{-\int_0^t \beta(s) \, \mathrm{d}s}}$.

The Monte-Carlo sample from this importance weight is the solution of the inverse Cumulative Distribution Function (CDF) of the importance distribution as

$$t = F^{-1}(u)$$
$$\Longleftrightarrow u = F(t) = \frac{1}{Z} \int_\epsilon^t \frac{g^2(s)}{\sigma^2(s)} \, \mathrm{d}s = \frac{1}{Z}\big(\mathcal{F}(t) - \mathcal{F}(\epsilon)\big), \tag{40}$$

where $u$ is a uniform sample from $[0, 1]$, $\mathcal{F}(t)$ is the antiderivative of the importance weight given by $\mathcal{F}(t) = \log\big(1 - e^{-0.5t^2(\beta_{max}-\beta_{min})-t\beta_{min}}\big) + 0.5t^2(\beta_{max} - \beta_{min}) + t\beta_{min}$, and $Z$ is the normalizing constant given by

$$\begin{aligned}
Z &= \int_\epsilon^T \frac{g^2(t)}{\sigma^2(t)} \, \mathrm{d}t \\
&= \Big[\log\big(1 - e^{-0.5t^2(\beta_{max}-\beta_{min})-t\beta_{min}}\big) + 0.5t^2(\beta_{max} - \beta_{min}) + t\beta_{min}\Big]_\epsilon^T \\
&= \log\big(1 - e^{-0.5T^2(\beta_{max}-\beta_{min})-T\beta_{min}}\big) - \log\big(1 - e^{-0.5\epsilon^2(\beta_{max}-\beta_{min})-\epsilon\beta_{min}}\big) \\
&\quad + 0.5(T^2 - \epsilon^2)(\beta_{max} - \beta_{min}) + (T - \epsilon)\beta_{min} \\
&\approx 23.86
\end{aligned}$$

Table 13: Ablation study on the stopping sampling time trained on DDPM++ (VP) in CIFAR-10.

| Model | FID ($t_{min} = 10^{-3}$) | FID ($t_{min} = 10^{-4}$) | FID ($t_{min} = 10^{-5}$) |
|---|---|---|---|
| INDM (deep, VP, NLL) | 5.94 | 5.74 | 5.71 |

Table 14: Ablation study on the SNR trained on NCSN++ (VE) in CIFAR-10. The performances are FID-5k scores.

| Model | Signal-to-Natio Ratio (SNR) | | | | |
|---|---|---|---|---|---|
| | 0.13 | 0.14 | 0.15 | 0.16 | 0.17 |
| INDM (VE, FID) | 7.24 | 7.12 | 7.20 | 7.25 | 7.34 |

for $T = 1$ and $\epsilon = 10^{-5}$. The solution for the inverse CDF in Eq. (40) becomes

$$e^{\int_0^t \beta(s)\,\mathrm{d}s} = 1 + \exp\left(Zu + \mathcal{F}(\epsilon)\right)$$

$$\iff \int_0^t \beta(s)\,\mathrm{d}s = \frac{1}{2}(\beta_{max} - \beta_{min})t^2 + \beta_{min}t = \log\left(1 + \exp\left(Zu + \mathcal{F}(\epsilon)\right)\right)$$

$$\iff t = \frac{-\beta_{min} + \sqrt{\beta_{min}^2 + 2(\beta_{max} - \beta_{min})\log\left(1 + \exp\left(Zu + \mathcal{F}(\epsilon)\right)\right)}}{\beta_{max} - \beta_{min}}.$$

The variation of the Monte-Carlo diffusion time depends on the uniform sample of $u$.

$$\int_0^t \beta(s)\,\mathrm{d}s = \log\left(\frac{1 - \sigma_{min}^2}{1 - \sigma_{min}^2\left(\frac{\sigma_{max}}{\sigma_{min}}\right)^t}\right) = \log\left(1 + \exp\left(Zu + \mathcal{F}(\epsilon)\right)\right)$$

$$\iff 1 - \sigma_{min}^2\left(\frac{\sigma_{max}}{\sigma_{min}}\right)^t = \frac{1 - \sigma_{min}^2}{1 + e^{Zu + \mathcal{F}(\epsilon)}}$$

$$\iff \sigma_{min}^2\left(\frac{\sigma_{max}^2}{\sigma_{min}^2}\right)^t = \frac{e^{Zu + \mathcal{F}(\epsilon)} + \sigma_{min}^2}{1 + e^{Zu + \mathcal{F}(\epsilon)}}$$

$$\iff t\log\frac{\sigma_{max}^2}{\sigma_{min}^2} = \log\left(e^{Zu + \mathcal{F}(\epsilon)} + \sigma_{min}^2\right) - \log\left(1 + e^{Zu + \mathcal{F}(\epsilon)}\right) - \log\sigma_{min}^2.$$

### F.2.2 Sampling Tricks to Improve FID

For ODE sampler, we use Runge Kutta 45 method [61] for the ODE solver. Since the score network was not trained beneath the truncation time, i.e., $s_\theta(z_t, t)$ has not been trained on $t \in [0, \epsilon)$, keep denoising up to the zero diffusion time would harm the sample fidelity. If $t_{min}$ is the stopping diffusion time of the ODE, one predictor step from $t_{min}$ to $0$ is applied to the noised sample, $z_{t_{min}}$, in order to eliminate the residual noise in $z_{t_{min}}$ to $z_0$ [62]. Table 13 searches the optimal stopping diffusion time, and it shows that the truncation time ($10^{-5}$) turns out to be the optimal stopping time. Throughout the paper, we report the FID (ODE) performance of our INDM with the training truncation time ($10^{-5}$). For VESDE, the ODE sampler fails to generate realistic images, so we do not report sample generation performance.

In PC sampler, for the predictor, we use the Reverse Diffusion Predictor for VESDE and the Euler-Maruyama Predictor for VPSDE. For the corrector, we use the Langevin dynamics [63] for VESDE, and we do not use any corrector for VPSDE. We use 1) Signal-to-Noise Ratio (SNR) scheduling, 2) temperature scheduling, 3) stopping time scheduling, and 4) data-adaptive prior than a fixed prior to improve FID. First, Table 14 presents that the optimal SNR is 0.14, which is slightly different from the optimal SNR of 0.16 in the linear diffusion [1]. We use SNR of 0.14 as default in our PC sampling.

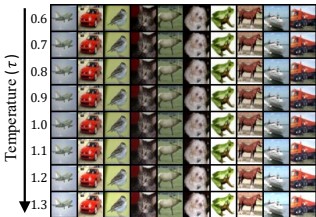

Figure 16: Ablation study for the flow temperature.

Table 15: Ablation study on the temperature for PC sampler trained on DDPM++ (VP) in CIFAR-10. The performances are FID scores. Contrary to Kingma and Dhariwal [56], the temperature bigger than 1 works the best.

| Model | Temperature | | | | | |
|---|---|---|---|---|---|---|
| | 1 | 1.03 | 1.04 | 1.05 | 1.1 | 1.2 |
| INDM (VP, FID) | 2.92 | 2.91 | 2.90 | 2.90 | 2.91 | 3.09 |

Table 16: Ablation study on the final denoising step trained on VESDE in CIFAR-10. The performances are FID scores.

| Model | FID $(\mathbf{x}_\epsilon)$ | FID $(\mathbf{x}_{-0.25})$ | FID $(\mathbf{x}_{-0.5})$ | FID $(\mathbf{x}_{-0.75})$ | FID $(\mathbf{x}_{-1})$ | FID $(\mathbf{x}_{-1.5})$ | FID $(\mathbf{x}_{-\infty})$ |
|---|---|---|---|---|---|---|---|
| NCSN++ (VE, FID) | 11.7 | 8.40 | 40.8 | 65.7 | 85.8 | 118 | 2.46 |
| INDM (VE, FID) | 2.40 | 2.33 | 2.31 | 2.29 | 2.29 | 2.34 | 2.37 |
| INDM (VE, deep, FID) | 2.35 | 2.29 | 2.28 | 2.29 | 2.29 | 2.36 | 2.33 |

Second, as introduced in Kingma and Dhariwal [56], we scale the generated latent, $\mathbf{z}_0^{\boldsymbol{\theta}}$, by multiplying the temperature. Table 15 presents that the optimal temperature for VPSDE is $1.04 \sim 1.05$ in terms of FID on INDM (VP, FID) setting. We use the temperature of 1 for the remaining settings except INDM (VP, FID). With temperature $\tau$, the normalizing flow puts its latent input scaled by $\tau$ to the flow network. In Figure 16, the image color with a higher temperature tends to be brighter, and we find that the optimal temperature depends on the experimental settings.

Third, the stopping time scheduling is a method that manipulate the final denoising step. To attain the variance of VESDE as $\sigma^2(t) = \sigma_{min}^2 (\frac{\sigma_{max}^2}{\sigma_{min}^2})^{2t}$, we should start the diffusion process of $d\mathbf{z}_t = \sigma^2(t)\,d\mathbf{w}_t$ at $t = -\infty$ because

$$\sigma^2(t) = \int_{t_0}^t g^2(s)\,ds = \sigma_{min}^2\left(\frac{\sigma_{max}^2}{\sigma_{min}^2}\right)^{2t} \tag{41}$$

implies $t_0 = -\infty$. If the generative SDE is $d\mathbf{z}_t = g^2(t)\mathbf{s}_{\boldsymbol{\theta}}(\mathbf{z}_t, t)\,d\bar{t} + \sigma^2(t)\,d\bar{\mathbf{w}}_t$, then the Euler-Maruyama discretization is

$$\mathbf{z}_{t_i} \leftarrow \mathbf{z}_{t_{i+1}} + g^2(t_{i+1})(t_i - t_{i+1})\mathbf{s}_{\boldsymbol{\theta}}(\mathbf{z}_{t_{i+1}}, t_{i+1}) + \sqrt{\sigma(t_{i+1})^2 - \sigma(t_i)^2}\boldsymbol{\epsilon}, \tag{42}$$

where $\boldsymbol{\epsilon} \sim \mathcal{N}(0, \mathbf{I})$. However, since the initial time of VESDE is $t = -\infty$, denoising the noised sample with the Euler-Maruyama discretization would incur arbitrary large error at the final step that denoises from $t = \epsilon$ to $t = -\infty$. Therefore, Song et al. [1] suggested the reverse diffusion predictor that denoises by

$$\mathbf{z}_{\sigma^{-1}(\sigma_i)} \leftarrow \mathbf{z}_{\sigma^{-1}(\sigma_{i+1})} + (\sigma_{i+1}^2 - \sigma_i^2)\mathbf{s}_{\boldsymbol{\theta}}(\mathbf{z}_{\sigma^{-1}(\sigma_{i+1})}, \sigma_{i+1}) + \sqrt{\sigma_{i+1}^2 - \sigma_i^2}\boldsymbol{\epsilon}, \tag{43}$$

which is equivalent to the Euler-Maruyama discretization if $t_i - t_{i+1}$ is small enough (because $\Delta\sigma^2(t) \approx g^2(t)\Delta t$ by Eq. (41)). The difference of Eqs. (42) and (43) is minor as long as we denoise on the range of $[\epsilon, T]$, but only Eq. (43) enables to denoise from $t = \epsilon$ to $t = -\infty$.

However, it turns out that the direction of the score network is not aligned to the direction of the data score near $t \approx 0$, so $\mathbf{s}_{\boldsymbol{\theta}}(\mathbf{z}_\epsilon, \sigma_{min})$ would not be accurate enough to the perturbed data score. Therefore, the final denoising step of

$$\mathbf{z}_{-\infty} \leftarrow \mathbf{z}_\epsilon + \sigma_{min}^2 \mathbf{s}_{\boldsymbol{\theta}}(\mathbf{z}_\epsilon, \sigma_{min}) + \sigma_{min}\boldsymbol{\epsilon},$$

might not be mostly effective. This leads us to try the final step as

$$\mathbf{z}_{\epsilon-\delta} = \mathbf{z}_\epsilon + \frac{1}{2}\Big(\sigma^2(\epsilon) - \sigma^2(\epsilon - \delta)\Big)\mathbf{s}_{\boldsymbol{\theta}}(\mathbf{z}_\epsilon, \sigma_{min}),$$

for various $\delta \geq 0$. After the denoising up to $\mathbf{z}_{\epsilon-\delta}$, we apply the inverse of the flow network to obtain $\mathbf{x}_{\epsilon-\delta} = \mathbf{h}_{\boldsymbol{\phi}}^{-1}(\mathbf{z}_{\epsilon-\delta})$, and Table 16 presents that there is a sweet spot $(\mathbf{x}_{-0.5} \sim \mathbf{x}_{-0.75})$ that works the best in terms of FID. We report the line searched FID performance for each of VESDE setting.

Lastly, we use $p_T^{\boldsymbol{\phi}}$ instead of $\pi$ to sample from INDM (VE). This data-adaptive prior is particularly beneficial on the experiment of VESDE. In INDM (VE), the data-adaptive prior reduces FID-5k

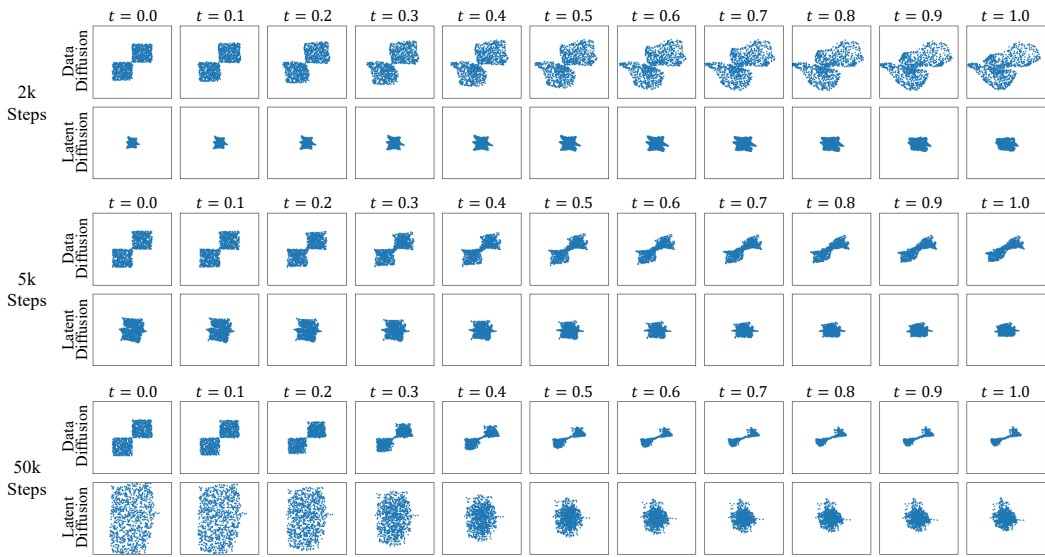

Figure 17: Comparison of data and latent diffusions by training steps of Checkerboard.

from 8.14 to 7.52, so we use this technique by default throughout out performance report. In INDM (VP), this technique is not effective, and we use the vanilla prior distribution. The reason why the data-adaptive prior is effective in VESDE is because the discrepancy of VESDE between $p_T^\phi$ and $\pi$ is significantly larger than VPSDE, see Figure 5 of Chen et al. [16].

We compute FID [18] for CIFAR-10 based on the statistics released by Song et al. [1][2], which used the modified Inception V1 network[3] in order to compare INDM to the baselines [1, 11] in a fair setting. On the other hand, for the CelebA dataset, we compute the clean-FID [64] that provides consistently antialiased performance.

### F.2.3 Interpolation Task

For the interpolation task, we provide the line-by-line algorithm in Algorithm 2. We train with the likelihood weighting as default for our experiment on the dataset interpolation. The interpolation loss of $\mathcal{L}_{int}$ consumes 0.2Gb of GPU memory, and the INDM loss of $\mathcal{L}_{INDM}$ takes 2.5Gb of GPU memory in the EMNIST-MNIST experiment.

---

**Algorithm 2** Data Interpolation of INDM

---

1: **repeat**
2:      Compute $\mathcal{L}_{INDM} = \mathcal{L}(\{\mathbf{x}_t\}_{t=0}^T, g^2; \{\phi, \theta\})$ for $\mathbf{x}_0 \sim p_r^{(1)}$
3:      Compute $\mathcal{L}_{int} = \mathbb{E}_{p_r^{(2)}}[-\log p_\phi(\mathbf{y})]$ for $\mathbf{y} \sim p_r^{(2)}$
4:      Compute $\mathcal{L}_{tot} = \mathcal{L}_{INDM} + \mathcal{L}_{int}$
5:      Update $\phi \leftarrow \phi - \frac{\partial \mathcal{L}_{tot}}{\partial \phi}$
6:      Update $\theta \leftarrow \theta - \frac{\partial \mathcal{L}_{tot}}{\partial \theta}$
7: **until** converged

---

### F.3 Effect of Pre-training

We find that training INDM with a pre-trained score network of linear diffusion models improves FID. Table 17 conducts the ablation study on the number of pre-training steps. We

Table 17: Ablation study on pre-training.

| # Pre-training Steps | 100k | 200k | 300k | 400k | 500k |
|---|---|---|---|---|---|
| NLL | 3.00 | 2.99 | 2.99 | 2.99 | 2.98 |
| FID | 7.39 | 7.31 | 6.80 | 6.65 | 6.22 |

[2]https://github.com/yang-song/score_sde_pytorch
[3]https://tfhub.dev/tensorflow/tfgan/eval/inception/1

Table 18: Elapsed time per a training step by discretization.

| Model | Complexity | $N = 100$ | $N = 1,000$ | $N = \infty$ |
|---|---|---|---|---|
| DDPM | $O(1)$ | 0.27 | 0.27 | 0.27 |
| SBP (w/o experience memory) | $O(N)$ | 2.83 | 23.3 | $\infty$ |
| SBP (w/ experience memory) | $O(N)$ | 0.52 | 2.39 | $\infty$ |
| DiffFlow | $O(N)$ | 18.45 | 180.88 | $\infty$ |
| INDM | $O(1)$ | 1.69 | 1.69 | 1.69 |

Table 19: Total training time in a single GPU days.

| Model | Total Training Time (GPU Days) | Training Steps | GPU Spec | #GPUs | NLL | FID |
|---|---|---|---|---|---|---|
| DDPM++ | 5 | 500k | P40 | 1 | 3.03 | 6.70 |
| LSGM | 44 | 450k | RTX 3090 | 8 | 2.87 | 6.89 |
| SBP | 3 | 260k | RTX 3090 | - | 2.98 | 3.18 |
| DiffFlow | 32 | 100k | RTX 2080 | 8 | 3.04 | 14.14 |
| INDM (including pre-training time) | 25 | 700k | P40 | 4 | 2.98 | 6.01 |
| INDM (w/o pre-training) | 60 | 600k | P40 | 4 | 2.98 | 8.49 |

pre-train the score network with DDPM++ (VP, NLL) for five variations of pre-training steps (100k/200k/300k/400k/500k), and we train flow+score networks for 350k steps further with NLL setting ($\lambda = g^2$). Table 17 empirically demonstrates that it is better to search the nonlinearity of the data process near the linear process. For this clear empirical advantage of pre-training, we report the quantitative performances in Section 7 with pre-training.

## F.4 Training Time

Table 18 presents the elapsed time per a training step by the number of discretization on CIFAR-10. In contrast to INDM which is invariant on the choice of $N$, the training time of SBP and DiffFlow is not scalable for their $O(N)$ complexities. The training time is measured under the identical computing resource (1x NVIDIA RTX 3090/Intel I7 3.8GHz) and the same batch size (32) to compare INDM with baselines in a fair setting.

Table 19 compares INDM with baselines with respect to a single GPU-time for the total training time on CIFAR-10. The remaining columns including training steps, GPU Spec, NLL, and FID are reported for the reference. For DiffFlow, we present the reported GPU days in the paper. For LSGM and SBP, we estimate the elapsed time with the released training configuration in their papers and GitHub repositories. For DDPM++ and INDM, we report the elapsed time from our own experiments. From the table, the overall training time of INDM/DiffFlow/LSGM remains at a similar scale. SBP is the fastest algorithm because of the experience replay memory technique. Note that a completely fair comparison between algorithms is infeasible because the training setup (e.g. #GPUs, training steps, network size ...) varies by algorithms. Also, P40 is strictly slower than RTX series GPUs.

## F.5 Visualization of Latent

### F.5.1 Visualization of 2d Latent Manifold

Figure 17 illustrates the data and latent manifolds of the 2d checkerboard dataset by training steps. The data manifold has the singularity at the origin, but this singularity disappears in the latent manifold after the training.

### F.5.2 Visualization of High-dimensional Latent Vector on Benchmark Datasets

Figure 18 illustrates the samples from (a) the data space and (b) the latent space. To visualize the latent vectors, we normalize the latent value into the $[0, 1]^d$ space.

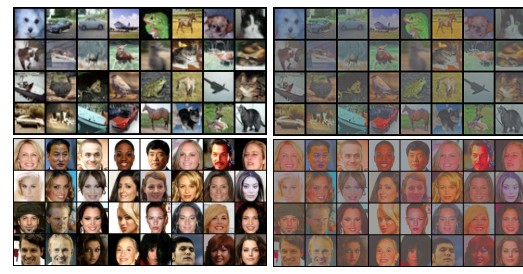

(a) Samples from $\mathbf{x}_0^{\phi,\theta}$     (b) Samples from $\mathbf{z}_0^{\theta}$

Figure 18: Samples from the data space and latent space on CIFAR-10 and CelebA.

### F.6 Nonlinear Diffusion Coefficient

INDM trains the volatility term, $\mathbf{G}_\phi$, which was fixed across previous research, except LSGM. The exact form of $\mathbf{G}_\phi$ in LSGM, however, is not derivable, so we exclude comparing LSGM in this section. As stated in Section 3, the noise distribution of a diffusion process is $\mathcal{N}(0, \mathbf{G}(\mathbf{x}_t, t)\mathbf{G}^T(\mathbf{x}_t, t))$, which is anisotropic by the input data, $\mathbf{x}_t$, and

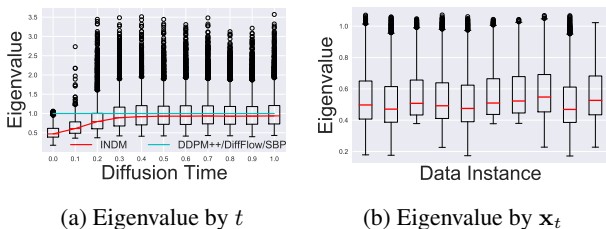

(a) Eigenvalue by $t$    (b) Eigenvalue by $\mathbf{x}_t$

Figure 19: Eigenvalue of $\mathbf{G}\mathbf{G}^T/g^2$ on CIFAR-10.

time, $t$. The influence of diffusion time on this covariance matrix is illustrated in Figure 19-(a). It presents the box plot of the eigenvalue distribution of the (normalized) covariance, $\mathbf{G}_\phi(\mathbf{x}_t, t)\mathbf{G}_\phi^T(\mathbf{x}_t, t)/g^2(t)$, on CIFAR-10, from $t = 0$ to $t = T$. All the eigenvalues of previous research collapse to one as they share the isotropic covariance matrix, $g^2(t)\mathbf{I}$. On the other hand, the eigenvalues of INDM is dispersive throughout the diffusion time. As the distribution becomes more dispersive, the covariance matrix becomes more unisotropic, and Figure 19-(a) implies that the learnt diffusion process is under a highly nonlinear noise perturbation in a range of large diffusion time. The covariance matrix also depends on the input data, and Figure 19-(b) illustrates the eigenvalue distribution of the covariance at distinctive data instances of $\mathbf{x}_t$ at $t = 0$. The eigenvalue distribution varies by instance, implying that data is diffused inhomogeneously by its location.

### F.7 Relative Energy

Each flow network parameter constructs a different latent trajectory, so training the flow network has the effect of shifting the diffusion bridge. To check if learning the flow network is helpful for the transportation cost or not, recall the Benamou-Brenier formula [65, 66], which is a dual formulation of the Wasserstein distance [66, 67] that the optimal transportation cost is the least kinetic energy out of all admissible transportation

Table 20: *Relative* Energy.

| Model | *Relative* Energy |
|---|---|
| DDPM++ | 1.60 |
| INDM | 1.23 |

plans: $W_2^2(p, q) = \inf_{\{p_t, \mathbf{v}_t\}_t} \left\{ \mathcal{K}(\{p_t, \mathbf{v}_t\}_t); \underbrace{\frac{\partial p_t}{\partial t} + \text{div}(p_t \mathbf{v}_t) = 0}_{\text{continuity equation}}, \underbrace{p_0 = p, p_T = q}_{\text{boundary conditions}} \right\}$, where

$\mathcal{K}(\{p_t, \mathbf{v}_t\}_t) := \int_0^T \int p_t(\mathbf{x}) \|\mathbf{v}_t(\mathbf{x})\|_2^2 \, d\mathbf{x} \, dt/T$ is the kinetic energy of the transportation. The continuity equation (that guarantees the conservation of mass along time transition [68]) and the boundary conditions determine the set of admissible trajectories, and the forward diffusion constructs an admissible trajectory. We *quantify* how much a trajectory is close to the optimal transport as the *relative* energy, given by $R(\phi) = \frac{\mathcal{K}(\{p_t^\phi, \mathbf{v}_t^\phi\}_t)}{W_2^2(p_0^\phi, p_T^\phi)}$. Table 20 shows that INDM's latent diffusion is more close to the optimal transport than DDPM++ on CIFAR-10.

### F.8 Full Quantitative Tables

Tables 21, 22, and 23 gives the full details of the quantitative comparisons to baseline models.

### F.9 Random samples

Figures 20 and 21 show the non cherry-picked random samples from INDM (VE, FID) on CIFAR-10 and INDM (VP, FID) on CelebA, respectively.

## G Proofs of Theorems and Propositions

**Theorem 1.** *Suppose that $p_{\phi, \theta}(\mathbf{x}_0)$ is the likelihood of a generative random variable $\mathbf{x}_0^{\phi, \theta}$. Then, the negative log-likelihood is bounded by*

$$\mathbb{E}_{p_r(\mathbf{x}_0)}\big[ -\log p_{\phi, \theta}(\mathbf{x}_0) \big] \leq \mathcal{L}\big(\{\mathbf{x}_t\}_{t=0}^T, g^2; \{\phi, \theta\}\big),$$

*where*

$$\mathcal{L}\big(\{\mathbf{x}_t\}_{t=0}^T, g^2; \{\boldsymbol{\phi}, \boldsymbol{\theta}\}\big) = -\mathbb{E}_{p_r(\mathbf{x}_0)}\big[\log\big|\det\big(\nabla_{\mathbf{x}_0}\mathbf{h}_{\boldsymbol{\phi}}\big)\big|\big]$$

$$+ \mathcal{L}\big(\{\mathbf{z}_t\}_{t=0}^T, g^2; \boldsymbol{\theta}\big) - \mathbb{E}_{\mathbf{z}_T}\big[\log\pi(\mathbf{z}_T)\big] + \frac{d}{2}\int_0^T \beta(t) - \frac{g^2(t)}{\sigma^2(t)}\, dt,$$

*with* $\mathcal{L}\big(\{\mathbf{z}_t\}_{t=0}^T, g^2; \boldsymbol{\theta}\big) := \frac{1}{2}\int_0^T g^2(t)\mathbb{E}_{\mathbf{z}_0, \mathbf{z}_t}\big[\|\mathbf{s}_{\boldsymbol{\theta}}(\mathbf{z}_t, t) - \nabla_{\mathbf{z}_t}\log p_{0t}(\mathbf{z}_t|\mathbf{z}_0)\|_2^2\big]\, dt$. *Here,* $p_{0t}(\mathbf{z}_t|\mathbf{z}_0)$ *is the transition probability of the forward linear diffusion process on latent space.*

*Proof of Theorem 1.* From the change of variable, the transformation of $\mathbf{z}_0 = \mathbf{h}_{\boldsymbol{\phi}}(\mathbf{x}_0)$ induces

$$p_r(\mathbf{x}_0) = \frac{p_0(\mathbf{z}_0)}{\big|\det\big(\frac{\partial\mathbf{h}_{\boldsymbol{\phi}}}{\partial\mathbf{x}_0}\big)\big|^{-1}},$$

and thus the entropy of the data distribution becomes

$$\begin{aligned}
\mathcal{H}(p_r) &= -\int p_r(\mathbf{x}_0)\log p_r(\mathbf{x}_0)\, d\mathbf{x}_0 \\
&= -\int p_0(\mathbf{z}_0)\log\frac{p_0(\mathbf{z}_0)}{\big|\det\big(\frac{\partial\mathbf{h}_{\boldsymbol{\phi}}}{\partial\mathbf{x}_0}\big)\big|^{-1}}\, d\mathbf{z}_0 \\
&= -\int p_r(\mathbf{x}_0)\log\big|\det\big(\frac{\partial\mathbf{h}_{\boldsymbol{\phi}}}{\partial\mathbf{x}_0}\big)\big|\, d\mathbf{x}_0 - \int p_0(\mathbf{z}_0)\log p_0(\mathbf{z}_0)\, d\mathbf{z}_0 \\
&= -\mathbb{E}_{p_r(\mathbf{x}_0)}\big[\log\big|\det\big(\frac{\partial\mathbf{h}_{\boldsymbol{\phi}}}{\partial\mathbf{x}_0}\big)\big|\big] - \int p_0(\mathbf{z}_0)\log p_0(\mathbf{z}_0)\, d\mathbf{z}_0 \\
&= -\mathbb{E}_{p_r(\mathbf{x}_0)}\big[\log\big|\det\big(\frac{\partial\mathbf{h}_{\boldsymbol{\phi}}}{\partial\mathbf{x}_0}\big)\big|\big] + \mathcal{H}(p_0).
\end{aligned}$$

From Theorem 4 of Song et al. [11], the entropy at $t = 0$ equals to

$$\mathcal{H}(p_0) = \mathcal{H}(p_T) - \frac{1}{2}\int_0^T \mathbb{E}_{p_t(\mathbf{z}_t)}\big[2\nabla_{\mathbf{z}_t}\cdot\mathbf{f}(\mathbf{z}_t, t) + g^2(t)\|\nabla_{\mathbf{z}_t}\log p_t(\mathbf{z}_t)\|_2^2\big]\, dt,$$

where $\mathbf{f}(\mathbf{z}_t, t)$ is a drift term of the diffusion for $\mathbf{z}_t$ and $p_t$ is the probability distribution of $\mathbf{z}_t$. Therefore, the negative log-likelihood becomes

$$\begin{aligned}
-\mathbb{E}_{p_r(\mathbf{x}_0)}\big[\log p_{\boldsymbol{\phi},\boldsymbol{\theta}}(\mathbf{x}_0)\big] &= D_{KL}(p_r\|p_{\boldsymbol{\phi},\boldsymbol{\theta}}) + \mathcal{H}(p_r) \\
&\leq D_{KL}(\boldsymbol{\mu}_{\boldsymbol{\phi}}(\{\mathbf{x}_t\})\|\boldsymbol{\nu}_{\boldsymbol{\phi},\boldsymbol{\theta}}(\{\mathbf{x}_t\})) + \mathcal{H}(p_r) \\
&= D_{KL}(\boldsymbol{\mu}_{\boldsymbol{\phi}}(\{\mathbf{x}_t\})\|\boldsymbol{\nu}_{\boldsymbol{\phi},\boldsymbol{\theta}}(\{\mathbf{x}_t\})) - \mathbb{E}_{p_r(\mathbf{x}_0)}\big[\log\big|\det\big(\frac{\partial\mathbf{h}_{\boldsymbol{\phi}}}{\partial\mathbf{x}_0}\big)\big|\big] + \mathcal{H}(p_0) \\
&= D_{KL}(\boldsymbol{\mu}_{\boldsymbol{\phi}}(\{\mathbf{z}_t\})\|\boldsymbol{\nu}_{\boldsymbol{\theta}}(\{\mathbf{z}_t\})) - \mathbb{E}_{p_r(\mathbf{x}_0)}\big[\log\big|\det\big(\frac{\partial\mathbf{h}_{\boldsymbol{\phi}}}{\partial\mathbf{x}_0}\big)\big|\big] + \mathcal{H}(p_T) \\
&\quad - \frac{1}{2}\int_0^T \mathbb{E}_{\mathbf{z}_t^{\boldsymbol{\phi}}}\big[-d\beta(t) + g^2(t)\|\nabla_{\mathbf{z}_t}\log p_t(\mathbf{z}_t)\|_2^2\big]\, dt.
\end{aligned}$$

Now, from Theorem 1 of [11], the KL-divergence between the path measures becomes

$$D_{KL}(\boldsymbol{\mu}_{\boldsymbol{\phi}}(\{\mathbf{z}_t\})\|\boldsymbol{\nu}_{\boldsymbol{\theta}}(\{\mathbf{z}_t\})) = D_{KL}(p_T(\mathbf{z}_T)\|\pi(\mathbf{z}_T)) \tag{44}$$

$$+ \frac{1}{2}\int_0^T g^2(t)\mathbb{E}_{p_t(\mathbf{z}_t)}\big[\|\mathbf{s}_{\boldsymbol{\theta}}(\mathbf{z}_t, t) - \nabla_{\mathbf{z}_t}\log p_t(\mathbf{z}_t)\|_2^2\big]\, dt, \tag{45}$$

so if we plug in this into the negative log-likelihood, we yield the following:

$$-\mathbb{E}_{p_r(\mathbf{x}_0)}\big[\log p_{\boldsymbol{\phi},\boldsymbol{\theta}}(\mathbf{x}_0)\big]$$

$$\leq -\mathbb{E}_{p_r(\mathbf{x}_0)}\big[\log\big|\det\big(\frac{\partial\mathbf{h}_{\boldsymbol{\phi}}}{\partial\mathbf{x}_0}\big)\big|\big] + \frac{1}{2}\int_0^T g^2(t)\mathbb{E}_{\mathbf{z}_t}\big[\|\mathbf{s}_{\boldsymbol{\theta}}(\mathbf{z}_t, t) - \nabla_{\mathbf{z}_t}\log p_t(\mathbf{z}_t)\|_2^2\big]\, dt$$

$$+ D_{KL}(p_T\|\pi) + \mathcal{H}(p_T) - \frac{1}{2}\int_0^T \mathbb{E}_{\mathbf{z}_t}\big[-d\beta(t) + g^2(t)\|\nabla_{\mathbf{z}_t}\log p_t(\mathbf{z}_t)\|_2^2\big]\,dt$$

$$= -\mathbb{E}_{p_r(\mathbf{x}_0)}\Big[\log\Big|\det\Big(\frac{\partial \mathbf{h}_\phi}{\partial \mathbf{x}_0}\Big)\Big|\Big] - \mathbb{E}_{\mathbf{z}_T}\big[\log\pi(\mathbf{z}_T)\big] + \frac{d}{2}\int_0^T \beta(t)\,dt$$

$$+ \frac{1}{2}\int_0^T g^2(t)\mathbb{E}_{\mathbf{z}_t}\big[\|\mathbf{s}_{\boldsymbol\theta}(\mathbf{z}_t,t) - \nabla_{\mathbf{z}_t}\log p_t(\mathbf{z}_t)\|_2^2 - \|\nabla_{\mathbf{z}_t}\log p_t(\mathbf{z}_t)\|_2^2\big]\,dt$$

Also, we have

$$\mathbb{E}_{\mathbf{z}_t}\big[\mathbf{s}_{\boldsymbol\theta}(\mathbf{z}_t,t)\cdot\nabla_{\mathbf{z}_t}\log p_t(\mathbf{z}_t)\big] = \int p_t(\mathbf{z}_t)\mathbf{s}_{\boldsymbol\theta}(\mathbf{z}_t,t)\cdot\nabla_{\mathbf{z}_t}\log p_t(\mathbf{z}_t)\,d\mathbf{z}_t$$

$$= \int \mathbf{s}_{\boldsymbol\theta}(\mathbf{z}_t,t)\cdot\nabla_{\mathbf{z}_t}p_t(\mathbf{z}_t)\,d\mathbf{z}_t$$

$$= \int \mathbf{s}_{\boldsymbol\theta}(\mathbf{z}_t,t)\cdot\int p_0(\mathbf{z}_0)\nabla_{\mathbf{z}_t}p_{0t}(\mathbf{z}_t|\mathbf{z}_0)\,d\mathbf{z}_0\,d\mathbf{z}_t$$

$$= \mathbb{E}_{\mathbf{z}_0}\mathbb{E}_{\mathbf{z}_t|\mathbf{z}_0}\big[\mathbf{s}_{\boldsymbol\theta}(\mathbf{z}_t,t)\cdot\nabla_{\mathbf{z}_t}\log p_{0t}(\mathbf{z}_t|\mathbf{z}_0)\big]$$

Therefore,

$$\frac{1}{2}\int_0^T g^2(t)\mathbb{E}_{\mathbf{z}_t}\big[\|\mathbf{s}_{\boldsymbol\theta}(\mathbf{z}_t,t) - \nabla_{\mathbf{z}_t}\log p_t(\mathbf{z}_t)\|_2^2 - \|\nabla_{\mathbf{z}_t}\log p_t(\mathbf{z}_t)\|_2^2\big]$$

$$= \int_0^T g^2(t)\mathbb{E}_{\mathbf{z}_t}\Big[\frac{1}{2}\|\mathbf{s}_{\boldsymbol\theta}(\mathbf{z}_t,t)\|_2^2 - \mathbf{s}_{\boldsymbol\theta}(\mathbf{z}_t,t)\cdot\nabla_{\mathbf{z}_t}\log p_t(\mathbf{z}_t)\Big]$$

$$= \int_0^T g^2(t)\mathbb{E}_{\mathbf{z}_0}\mathbb{E}_{\mathbf{z}_t|\mathbf{z}_0}\Big[\frac{1}{2}\|\mathbf{s}_{\boldsymbol\theta}(\mathbf{z}_t,t)\|_2^2 - \mathbf{s}_{\boldsymbol\theta}(\mathbf{z}_t,t)\cdot\nabla_{\mathbf{z}_t}\log p_{0t}(\mathbf{z}_t|\mathbf{z}_0)\Big]$$

$$= \frac{1}{2}\int_0^T g^2(t)\mathbb{E}_{\mathbf{z}_0}\mathbb{E}_{\mathbf{z}_t|\mathbf{z}_0}\Big[\|\mathbf{s}_{\boldsymbol\theta}(\mathbf{z}_t,t) - \nabla_{\mathbf{z}_t}\log p_{0t}(\mathbf{z}_t|\mathbf{z}_0)\|_2^2 - \|\nabla_{\mathbf{z}_t}\log p_{0t}(\mathbf{z}_t|\mathbf{z}_0)\|_2^2\Big].$$

Now, since $p_{0t}(\mathbf{z}_t|\mathbf{z}_0) = \mathcal{N}(\mathbf{z}_t; \mu(t)\mathbf{z}_t, \sigma^2(t)\mathbf{I})$ for $\mu(t)$ and $\sigma^2(t)$ determined by $\beta(t)$ and $g(t)$, we have

$$\mathbb{E}_{\mathbf{z}_t|\mathbf{z}_0}\big[\|\nabla_{\mathbf{z}_t}\log p_{0t}(\mathbf{z}_t|\mathbf{z}_0)\|_2^2\big] = \mathbb{E}_{\mathbf{z}_t|\mathbf{z}_0}\Big[\Big\|\frac{\mathbf{z}_t - \mu(t)\mathbf{z}_0}{\sigma^2(t)}\Big\|_2^2\Big] = \mathbb{E}_{\mathcal{N}(\mathbf{z};0,\mathbf{I})}\Big[\frac{\|\mathbf{z}\|_2^2}{\sigma^2(t)}\Big] = \frac{d}{\sigma^2(t)},$$

and we have the desired result. $\qquad\square$

**Proposition 1.** *Suppose $q_t^{\boldsymbol\theta}$ is the marginal distribution of $\nu_{\boldsymbol\theta}$ at t. The variational gap is*

$$\mathrm{Gap}\big(\boldsymbol\mu_\phi(\{\mathbf{x}_t\}), \nu_{\phi,\boldsymbol\theta}(\{\mathbf{x}_t\})\big) := D_{KL}\big(\boldsymbol\mu_\phi(\{\mathbf{x}_t\})\|\nu_{\phi,\boldsymbol\theta}(\{\mathbf{x}_t\})\big) - D_{KL}\big(p_0^\phi(\mathbf{x}_0)\|q_0^{\boldsymbol\theta}(\mathbf{x}_0)\big)$$

$$= \frac{1}{2}\int_0^T g^2(t)\mathbb{E}_{p_t^\phi(\mathbf{z}_t)}\big[\underbrace{\|\nabla\log q_t^{\boldsymbol\theta}(\mathbf{z}_t) - \mathbf{s}_{\boldsymbol\theta}(\mathbf{z}_t,t)\|_2^2}_{\text{Score-only error}}\big]\,dt.$$

***Proof of Proposition 1.*** Suppose $q_t^{\boldsymbol\theta}$ is a marginal distribution of the path measure of the generative SDE given by

$$d\mathbf{z}_t = \big[\mathbf{f}(\mathbf{z}_t,t) - g^2(t)\mathbf{s}_{\boldsymbol\theta}(\mathbf{z}_t,t)\big]\,d\bar{t} + g(t)\,d\bar{\mathbf{w}}_t. \tag{46}$$

The Fokker-Planck equation of the above generative SDE satisfies

$$\frac{\partial q_t^{\boldsymbol\theta}}{\partial t}(\mathbf{z}_t) = -\sum_{i=1}^d \frac{\partial}{\partial z_i}\big(\big[f_i(\mathbf{z}_t,t) - g^2(t)\big(\mathbf{s}_{\boldsymbol\theta}(\mathbf{z}_t,t)\big)_i\big]q_t^{\boldsymbol\theta}(\mathbf{z}_t)\big) - \frac{g^2(t)}{2}\sum_{i=1}^d \frac{\partial^2}{\partial z_i^2}\big[q_t^{\boldsymbol\theta}(\mathbf{z}_t)\big]$$

$$= \mathrm{div}\Big(\Big(-\mathbf{f}(\mathbf{z}_t,t) + g^2(t)\mathbf{s}_{\boldsymbol\theta}(\mathbf{z}_t,t) - \frac{g^2(t)}{2}\nabla\log q_t^{\boldsymbol\theta}(\mathbf{z}_t)\Big)q_t^{\boldsymbol\theta}(\mathbf{z}_t)\Big). \tag{47}$$

On the other hand, if $p_t^\phi$ is the marginal distribution of the path measure of the forward SDE given by

$$\mathrm{d}\mathbf{z}_t = \mathbf{f}(\mathbf{z}_t, t)\,\mathrm{d}t + g(t)\,\mathrm{d}\mathbf{w}_t,$$

then the corresponding Fokker-Planck equation becomes

$$\frac{\partial p_t^\phi}{\partial t}(\mathbf{z}_t) = \mathrm{div}\left(\left(-\mathbf{f}(\mathbf{z}_t, t) + \frac{g^2(t)}{2}\nabla \log p_t^\phi(\mathbf{z}_t, t)\right)p_t^\phi(\mathbf{z}_t)\right). \tag{48}$$

Combining Eq. (47) with Eq. (48) and using the integration by parts, the derivative of the KL divergence becomes

$$\begin{aligned}
\frac{\partial D_{KL}(p_t^\phi \| q_t^{\boldsymbol{\theta}})}{\partial t} &= \frac{\partial}{\partial t}\int p_t^\phi(\mathbf{z}_t)\log\frac{p_t^\phi(\mathbf{z}_t)}{q_t^{\boldsymbol{\theta}}(\mathbf{z}_t)}\,\mathrm{d}\mathbf{z}_t \\
&= \int \frac{\partial p_t^\phi}{\partial t}(\mathbf{z}_t)\log\frac{p_t^\phi(\mathbf{z}_t)}{q_t^{\boldsymbol{\theta}}(\mathbf{z}_t)}\,\mathrm{d}\mathbf{z}_t - \int \frac{\partial q_t^{\boldsymbol{\theta}}}{\partial t}(\mathbf{z}_t)\frac{p_t^\phi(\mathbf{z}_t)}{q_t^{\boldsymbol{\theta}}(\mathbf{z}_t)}\,\mathrm{d}\mathbf{z}_t \\
&= -\int p_t^\phi(\mathbf{z}_t)\left(-\mathbf{f}(\mathbf{z}_t, t) + \frac{g^2(t)}{2}\nabla\log p_t^\phi(\mathbf{z}_t)\right)^T\nabla\log\frac{p_t^\phi(\mathbf{z}_t)}{q_t^{\boldsymbol{\theta}}(\mathbf{z}_t)}\,\mathrm{d}\mathbf{z}_t \\
&\quad + \int p_t^\phi(\mathbf{z}_t)\left(-\mathbf{f}(\mathbf{z}_t, t) + g^2(t)\mathbf{s}_{\boldsymbol{\theta}}(\mathbf{z}_t, t) - \frac{g^2(t)}{2}\nabla\log q_t^{\boldsymbol{\theta}}(\mathbf{z}_t)\right)^T\nabla\log\frac{p_t^\phi(\mathbf{z}_t)}{q_t^{\boldsymbol{\theta}}(\mathbf{z}_t)}\,\mathrm{d}\mathbf{z}_t \\
&= \frac{g^2(t)}{2}\int p_t^\phi(\mathbf{z}_t)\left(\nabla\log\frac{p_t^\phi(\mathbf{z}_t)}{q_t^{\boldsymbol{\theta}}(\mathbf{z}_t)}\right)^T\left(2\mathbf{s}_{\boldsymbol{\theta}}(\mathbf{z}_t, t) - \nabla\log p_t^\phi(\mathbf{z}_t) - \nabla\log q_t^{\boldsymbol{\theta}}(\mathbf{z}_t)\right)\mathrm{d}\mathbf{z}_t.
\end{aligned}$$

Integrating the above derivative, we get the KL divergence of

$$D_{KL}\big(p_0^\phi(\mathbf{z}_0)\|q_0^{\boldsymbol{\theta}}(\mathbf{z}_0)\big) = -\int_0^T \frac{\partial D_{KL}(p_t^\phi\|q_t^{\boldsymbol{\theta}})}{\partial t}\,\mathrm{d}t + D_{KL}(p_T^\phi\|q_T^{\boldsymbol{\theta}}) \tag{49}$$

$$= \int_0^T \frac{g^2(t)}{2}\mathbb{E}_{\mathbf{z}_t^\phi}\left[(\nabla\log p_t^\phi - \nabla\log q_t^{\boldsymbol{\theta}})^T(\nabla\log p_t^\phi + \nabla\log q_t^{\boldsymbol{\theta}} - 2\mathbf{s}_{\boldsymbol{\theta}})\right]\mathrm{d}t + D_{KL}(p_T^\phi\|q_T^{\boldsymbol{\theta}}).$$

Also, from Eq. (44), we have

$$\begin{aligned}
&D_{KL}\big(\boldsymbol{\mu}_\phi(\{\mathbf{x}_t\})\|\boldsymbol{\nu}_{\phi,\boldsymbol{\theta}}(\{\mathbf{x}_t\})\big) = D_{KL}\big(\boldsymbol{\mu}_\phi(\{\mathbf{z}_t\})\|\boldsymbol{\nu}_{\boldsymbol{\theta}}(\{\mathbf{z}_t\})\big) \\
&\qquad = \int_0^T \frac{g^2(t)}{2}\mathbb{E}_{p_t^\phi(\mathbf{z}_t)}\big[\|\nabla\log p_t^\phi(\mathbf{z}_t) - \mathbf{s}_{\boldsymbol{\theta}}(\mathbf{z}_t, t)\|_2^2\big]\,\mathrm{d}t + D_{KL}(p_T^\phi\|q_T^{\boldsymbol{\theta}}).
\end{aligned} \tag{50}$$

By subtracting Eq. (49) from Eq. (50), we get the desired result:

$$\begin{aligned}
\mathrm{Gap}(\boldsymbol{\mu}_\phi, \boldsymbol{\nu}_{\phi,\boldsymbol{\theta}}) &= D_{KL}\big(\boldsymbol{\mu}_\phi(\{\mathbf{x}_t\})\|\boldsymbol{\nu}_{\phi,\boldsymbol{\theta}}(\{\mathbf{x}_t\})\big) - D_{KL}\big(p_r(\mathbf{x}_0)\|p_{\phi,\boldsymbol{\theta}}(\mathbf{x}_0)\big) \\
&= D_{KL}\big(\boldsymbol{\mu}_\phi(\{\mathbf{z}_t\})\|\boldsymbol{\nu}_{\boldsymbol{\theta}}(\{\mathbf{z}_t\})\big) - D_{KL}\big(p_0^\phi(\mathbf{z}_0)\|q_0^{\boldsymbol{\theta}}(\mathbf{z}_0)\big) \\
&= \int \frac{g^2(t)}{2}\mathbb{E}_{\mathbf{z}_t^\phi}\left[\|\nabla\log p_t^\phi - \mathbf{s}_{\boldsymbol{\theta}}\|_2^2 - (\nabla\log p_t^\phi - \nabla\log q_t^{\boldsymbol{\theta}})^T(\nabla\log p_t^\phi + \nabla\log q_t^{\boldsymbol{\theta}} - 2\mathbf{s}_{\boldsymbol{\theta}})\right]\mathrm{d}t \\
&= \int \frac{g^2(t)}{2}\mathbb{E}_{p_t^\phi(\mathbf{z}_t)}\left[\|\nabla\log q_t^{\boldsymbol{\theta}} - \mathbf{s}_{\boldsymbol{\theta}}\|_2^2\right]\mathrm{d}t.
\end{aligned}$$

$\square$

**Theorem 2.** $\mathrm{Gap}(\boldsymbol{\mu}_\phi, \boldsymbol{\nu}_{\phi,\boldsymbol{\theta}}) = 0$ *if and only if* $\mathbf{s}_{\boldsymbol{\theta}} \in \mathbf{S}_{sol}$.

***Proof of Theorem 2.*** ($\Rightarrow$) Suppose the variational gap is zero. Then, as the support of $p_t^\phi$ is the whole space of $\mathbb{R}^d$, Theorem 1 implies that $\mathbf{s}_{\boldsymbol{\theta}}(\mathbf{z}_t, t) = \nabla\log q_t^{\boldsymbol{\theta}}(\mathbf{z}_t)$ almost everywhere, for any $t > 0$. To check if $\mathbf{s}_{\boldsymbol{\theta}}(\mathbf{z}_0, 0) = \nabla\log q_0^{\boldsymbol{\theta}}(\mathbf{z}_0)$ at $t = 0$, suppose $\mathbf{s}_{\boldsymbol{\theta}}(\mathbf{z}_0, 0) \neq \nabla\log q_0^{\boldsymbol{\theta}}(\mathbf{z}_0)$ on a set of positive measure. Then, from the continuity of $\mathbf{s}_{\boldsymbol{\theta}}$ and $\nabla\log q_t^{\boldsymbol{\theta}}$, we have $\mathbf{s}_{\boldsymbol{\theta}}(\mathbf{z}_s, s) \neq \nabla\log q_s^{\boldsymbol{\theta}}(\mathbf{z}_s)$

on $s < t_0$ for some $t_0$. Therefore, for any $t \in [0, T]$, we conclude that $\mathbf{s}_{\boldsymbol{\theta}}(\mathbf{z}_t, t) = \nabla \log q_t^{\boldsymbol{\theta}}(\mathbf{z}_t)$ almost everywhere and Eq. (46) becomes

$$d\mathbf{z}_t = \left[\mathbf{f}(\mathbf{z}_t, t) - g^2(t)\nabla \log q_t^{\boldsymbol{\theta}}(\mathbf{z}_t)\right] d\bar{t} + g(t) \, d\bar{\mathbf{w}}_t. \tag{51}$$

As the Fokker-Planck equation of the SDE of Eq. (51) becomes

$$\frac{\partial q_t^{\boldsymbol{\theta}}}{\partial t}(\mathbf{z}_t) = \text{div}\left(\left(-\mathbf{f}(\mathbf{z}_t, t) + \frac{g^2(t)}{2}\nabla \log q_t^{\boldsymbol{\theta}}(\mathbf{z}_t)\right)q_t^{\boldsymbol{\theta}}(\mathbf{z}_t)\right),$$

which coincide with the Fokker-Planck equation of the forward SDE of $d\mathbf{z}_t = \mathbf{f}(\mathbf{z}_t, t) \, dt + g(t) \, d\mathbf{w}_t$, we conclude $\mathbf{s}_{\boldsymbol{\theta}} \in \mathbf{S}_{sol}$ by definition.

($\Leftarrow$) holds from Lemma 2. $\qquad\square$

**Theorem 3.** *For any fixed* $\mathbf{s}_{\bar{\boldsymbol{\theta}}} \in \mathbf{S}_{sol}$, *if* $\boldsymbol{\phi}^* \in \arg\min_{\boldsymbol{\phi}} D_{KL}(\boldsymbol{\mu}_{\boldsymbol{\phi}} \| \boldsymbol{\nu}_{\boldsymbol{\phi}, \bar{\boldsymbol{\theta}}})$, *then* $\mathbf{s}_{\boldsymbol{\phi}^*}(\mathbf{z}_t, t) = \nabla \log p_t^{\boldsymbol{\phi}^*}(\mathbf{z}_t) = \mathbf{s}_{\bar{\boldsymbol{\theta}}}(\mathbf{z}_t, t)$, *and* $D_{KL}(\boldsymbol{\mu}_{\boldsymbol{\phi}^*} \| \boldsymbol{\nu}_{\boldsymbol{\phi}^*, \bar{\boldsymbol{\theta}}}) = D_{KL}(p_r \| p_{\boldsymbol{\phi}^*, \bar{\boldsymbol{\theta}}}) = Gap(\boldsymbol{\mu}_{\boldsymbol{\phi}^*}, \boldsymbol{\nu}_{\boldsymbol{\phi}^*, \bar{\boldsymbol{\theta}}}) = 0$.

***Proof of Theorem 3.*** If $\mathbf{s}_{\bar{\boldsymbol{\theta}}} \in \mathbf{S}_{sol}$, there exists $q_0$ such that $\mathbf{s}_{\bar{\boldsymbol{\theta}}}(\mathbf{z}_t, t) = \nabla \log q_t(\mathbf{z}_t)$, where $\mathbf{z}_t \sim q_t$ is governed by $d\mathbf{z}_t = \mathbf{f}(\mathbf{z}_t, t) \, dt + g(t) \, d\mathbf{w}_t$ that starts from $\mathbf{z}_0 \sim q_0$. This implies that the generative path measure of $\boldsymbol{\nu}_{\boldsymbol{\phi}, \bar{\boldsymbol{\theta}}}$ coincides with some forward path measure. On the other hand, the forward latent diffusion is also governed by $d\mathbf{z}_t = \mathbf{f}(\mathbf{z}_t, t) \, dt + g(t) \, d\mathbf{w}_t$ that starts from $\mathbf{z}_0 \sim p_0^{\boldsymbol{\phi}}$. Therefore, if $p_0^{\boldsymbol{\phi}} = q_0$ almost everywhere, then the generative path measure of $\boldsymbol{\nu}_{\boldsymbol{\phi}, \bar{\boldsymbol{\theta}}}$ coincides with the forward path measure of $\boldsymbol{\mu}_{\boldsymbol{\phi}}$, and it holds that $D_{KL}(\boldsymbol{\mu}_{\boldsymbol{\phi}} \| \boldsymbol{\nu}_{\boldsymbol{\phi}, \bar{\boldsymbol{\theta}}}) = \int_0^T \frac{g^2(t)}{2}\mathbb{E}[\|\nabla \log p_t^{\boldsymbol{\phi}} - \nabla \log q_t\|_2^2] \, dt + D_{KL}(p_T^{\boldsymbol{\phi}} \| q_T) = 0$. If $p_0^{\boldsymbol{\phi}} \neq q_0$ on a set of positive measure $A$, then $D_{KL}(\boldsymbol{\mu}_{\boldsymbol{\phi}} \| \boldsymbol{\nu}_{\boldsymbol{\phi}, \bar{\boldsymbol{\theta}}}) = \int_0^T \frac{g^2(t)}{2}\mathbb{E}[\|\nabla \log p_t^{\boldsymbol{\phi}} - \nabla \log q_t\|_2^2] \, dt + D_{KL}(p_T^{\boldsymbol{\phi}} \| q_T)$ is strictly positive because $\|\nabla \log p_t^{\boldsymbol{\phi}} - \nabla \log q_t\|_2^2 > 0$ on $A$, for any $t$. This leads that if $\boldsymbol{\phi}^* \in \arg\min_{\boldsymbol{\phi}} D_{KL}(\boldsymbol{\mu}_{\boldsymbol{\phi}} \| \boldsymbol{\nu}_{\boldsymbol{\phi}, \bar{\boldsymbol{\theta}}})$, then $D_{KL}(\boldsymbol{\mu}_{\boldsymbol{\phi}^*} \| \boldsymbol{\nu}_{\boldsymbol{\phi}^*, \bar{\boldsymbol{\theta}}}) = 0$, and $p_0^{\boldsymbol{\phi}^*} = q_0$ almost everywhere. Therefore, we get the desired result because $0 = D_{KL}(\boldsymbol{\mu}_{\boldsymbol{\phi}^*} \| \boldsymbol{\nu}_{\boldsymbol{\phi}^*, \bar{\boldsymbol{\theta}}}) \geq D_{KL}(p_r \| p_{\boldsymbol{\phi}^*, \bar{\boldsymbol{\theta}}}) \geq 0$. $\qquad\square$

**Proposition 2.** $\mathbf{s}_{\boldsymbol{\theta}} \in \mathbf{S}_{div}$ *if and only if* $\nabla_{\mathbf{z}_t}\mathbf{s}_{\boldsymbol{\theta}}(\mathbf{z}_t, t)$ *is symmetric.*

***Proof of Proposition 2.*** If $\nabla_{\mathbf{z}_t}\mathbf{s}_{\boldsymbol{\theta}}(\mathbf{z}_t, t)$ is symmetric, then $\mathbf{s}_{\boldsymbol{\theta}}(\mathbf{z}_t, t)$ is a 1-form, and $\mathbf{s}_{\boldsymbol{\theta}} \in \mathbf{S}_{div}$. If $\mathbf{s}_{\boldsymbol{\theta}} \in \mathbf{S}_{div}$, then there exists $p_t$ such that $\mathbf{s}_{\boldsymbol{\theta}}(\mathbf{z}_t, t) = \nabla \log p_t(\mathbf{z}_t)$. Thus, $\nabla \mathbf{s}_{\boldsymbol{\theta}} = \nabla^2 \log p_t$, which is symmetric. $\qquad\square$

**Proposition 3.** *A matrix* $A \in \mathbb{R}^{d \times d}$ *is symmetric if and only if* $\mathbb{E}_{\boldsymbol{\epsilon}_1, \boldsymbol{\epsilon}_2 \sim \mathcal{N}(0, \mathbf{I})}\left[(\boldsymbol{\epsilon}_2^T(A - A^T)\boldsymbol{\epsilon}_1)^2\right] = 0$.

***Proof of Proposition 3.*** As

$$\mathbb{E}_{\boldsymbol{\epsilon}_1, \boldsymbol{\epsilon}_2 \sim \mathcal{N}(0, \mathbf{I})}\left[(\boldsymbol{\epsilon}_2^T A\boldsymbol{\epsilon}_1 - \boldsymbol{\epsilon}_1^T A\boldsymbol{\epsilon}_2)^2\right] = \mathbb{E}_{\boldsymbol{\epsilon}_1, \boldsymbol{\epsilon}_2 \sim \mathcal{N}(0, \mathbf{I})}\left[(\boldsymbol{\epsilon}_2^T(A - A^T)\boldsymbol{\epsilon}_1)^2\right],$$

$A$ is symmetric if and only if $\mathbb{E}_{\boldsymbol{\epsilon}_1, \boldsymbol{\epsilon}_2 \sim \mathcal{N}(0, \mathbf{I})}\left[(\boldsymbol{\epsilon}_2^T A\boldsymbol{\epsilon}_1 - \boldsymbol{\epsilon}_1^T A\boldsymbol{\epsilon}_2)^2\right] = 0$. $\qquad\square$

**Proposition 4.** *Let* $\boldsymbol{\epsilon}_1$ *and* $\boldsymbol{\epsilon}_2$ *be vectors of d independent samples from a random variable U with mean zero. Then*

$$\mathbb{E}_{\boldsymbol{\epsilon}_1, \boldsymbol{\epsilon}_2}[(\boldsymbol{\epsilon}_2^T(A - A^T)\boldsymbol{\epsilon}_1)^2] = \mathbb{E}_U[U^2]^2\|A - A^T\|_F^2$$

*and*

$$Var\Big((\boldsymbol{\epsilon}_2^T(A - A^T)\boldsymbol{\epsilon}_1)^2\Big) = Var(U^2)\Big(Var(U^2) + 2\big(Var(U) + \mathbb{E}_U[U]^2\big)^2\Big)\sum_{a,b}(\Delta A)_{ab}^4$$

$$+ 2\big(Var(U) + \mathbb{E}_U[U]^2\big)^2\Big(3Var(U^2) + 2\big(Var(U) + \mathbb{E}_U[U]^2\big)^2\Big)\sum_a \sum_{b \neq d}(\Delta A)_{ab}^2(\Delta A)_{ad}^2$$

$$+ 2\big(Var(U) + \mathbb{E}_U[U]^2\big)^4\Big(\sum_{a \neq c}\sum_{b \neq d}(\Delta A)_{ab}^2(\Delta A)_{cd}^2$$

$$+ 3\sum_{a \neq c}\sum_{b \neq d}(\Delta A)_{ab}(\Delta A)_{ad}(\Delta A)_{cb}(\Delta A)_{cd}\Big),$$

*where* $(\Delta A)_{ab} := A_{ab} - A_{ba}$.

*Proof of Proposition 4.*

$$\mathbb{E}_{\boldsymbol{\epsilon}_1,\boldsymbol{\epsilon}_2}\left[\left(\boldsymbol{\epsilon}_2^T(A-A^T)\boldsymbol{\epsilon}_1\right)^2\right] = \mathbb{E}_{\boldsymbol{\epsilon}_1,\boldsymbol{\epsilon}_2}\left[\left(\sum_{i,j}\epsilon_{1,i}\epsilon_{2,j}(A_{ij}-A_{ji})^2\right)^2\right]$$

$$= \mathbb{E}_{\boldsymbol{\epsilon}_1,\boldsymbol{\epsilon}_2}\left[\sum_{i,j,r,s}\epsilon_{1,i}\epsilon_{2,j}\epsilon_{1,r}\epsilon_{2,s}(A_{ij}-A_{ji})(A_{rs}-A_{sr})\right]$$

$$= \mathbb{E}_{\boldsymbol{\epsilon}_1,\boldsymbol{\epsilon}_2}\left[\sum_{i,j}\epsilon_{1,i}^2\epsilon_{2,j}^2(A_{ij}-A_{ji})^2\right]$$

$$= \mathbb{E}_U[U^2]^2\sum_{i,j}(A_{ij}-A_{ji})^2$$

$$= \mathbb{E}_U[U^2]^2\|A-A^T\|_F^2.$$

Also, if $B := A - A^T$, then

$$\mathbb{E}_{\boldsymbol{\epsilon}_1,\boldsymbol{\epsilon}_2}\left[\left(\boldsymbol{\epsilon}_2^T(A-A^T)\boldsymbol{\epsilon}_1\right)^4\right]$$

$$= \mathbb{E}_{\boldsymbol{\epsilon}_1,\boldsymbol{\epsilon}_2}\left[\sum_{a,b,c,d,e,f,g,h}\epsilon_{1,a}\epsilon_{2,b}\epsilon_{1,c}\epsilon_{2,d}\epsilon_{1,e}\epsilon_{2,f}\epsilon_{1,g}\epsilon_{2,h}B_{ab}B_{cd}B_{ef}B_{gh}\right]$$

$$= \mathbb{E}_{\boldsymbol{\epsilon}_1,\boldsymbol{\epsilon}_2}\left[\sum_{b,d,f,h}\epsilon_{2,b}\epsilon_{2,d}\epsilon_{2,f}\epsilon_{2,h}\sum_{a,c}\epsilon_{1,a}\epsilon_{1,c}\sum_{e,g}\epsilon_{1,e}\epsilon_{1,g}B_{ab}B_{cd}B_{ef}B_{gh}\right]$$

$$= \mathbb{E}_{\boldsymbol{\epsilon}_1,\boldsymbol{\epsilon}_2}\left[\sum_{b,d,f,h}\epsilon_{2,b}\epsilon_{2,d}\epsilon_{2,f}\epsilon_{2,h}\sum_{a}\epsilon_{1,a}^2\sum_{e,g}\epsilon_{1,e}\epsilon_{1,g}B_{ab}B_{ad}B_{ef}B_{gh}\right]$$

$$+ \mathbb{E}_{\boldsymbol{\epsilon}_1,\boldsymbol{\epsilon}_2}\left[\sum_{b,d,f,h}\epsilon_{2,b}\epsilon_{2,d}\epsilon_{2,f}\epsilon_{2,h}\sum_{a\neq c}\epsilon_{1,a}\epsilon_{1,c}\sum_{e,g}\epsilon_{1,e}\epsilon_{1,g}B_{ab}B_{cd}B_{ef}B_{gh}\right]$$

$$= \mathbb{E}_{\boldsymbol{\epsilon}_1,\boldsymbol{\epsilon}_2}\left[\sum_{b,d,f,h}\epsilon_{2,b}\epsilon_{2,d}\epsilon_{2,f}\epsilon_{2,h}\sum_{a}\epsilon_{1,a}^2\sum_{e}\epsilon_{1,e}^2B_{ab}B_{ad}B_{ef}B_{eh}\right]$$

$$+ \mathbb{E}_{\boldsymbol{\epsilon}_1,\boldsymbol{\epsilon}_2}\left[\sum_{b,d,f,h}\epsilon_{2,b}\epsilon_{2,d}\epsilon_{2,f}\epsilon_{2,h}\sum_{a\neq c}\epsilon_{1,a}^2\epsilon_{1,c}^2B_{ab}B_{cd}(B_{af}B_{ch}+B_{cf}B_{ah})\right]$$

$$= \mathbb{E}_{\boldsymbol{\epsilon}_1,\boldsymbol{\epsilon}_2}\left[\sum_{b,d,f,h}\epsilon_{2,b}\epsilon_{2,d}\epsilon_{2,f}\epsilon_{2,h}\sum_{a}\epsilon_{1,a}^4B_{ab}B_{ad}B_{af}B_{ah}\right]$$

$$+ 3\mathbb{E}_{\boldsymbol{\epsilon}_1,\boldsymbol{\epsilon}_2}\left[\sum_{b,d,f,h}\epsilon_{2,b}\epsilon_{2,d}\epsilon_{2,f}\epsilon_{2,h}\sum_{a\neq c}\epsilon_{1,a}^2\epsilon_{1,c}^2B_{ab}B_{af}B_{cd}B_{ch}\right]$$

$$= \mathbb{E}_U[U^4]\sum_{a}\mathbb{E}_{\boldsymbol{\epsilon}_2}\left[\sum_{b,d,f,h}\epsilon_{2,b}\epsilon_{2,d}\epsilon_{2,f}\epsilon_{2,h}B_{ab}B_{ad}B_{af}B_{ah}\right]$$

$$+ 3\mathbb{E}_U[U^2]^2\sum_{a\neq c}\mathbb{E}_{\boldsymbol{\epsilon}_2}\left[\sum_{b,d,f,h}\epsilon_{2,b}\epsilon_{2,d}\epsilon_{2,f}\epsilon_{2,h}B_{ab}B_{af}B_{cd}B_{ch}\right]$$

$$= \mathbb{E}_U[U^4]\sum_{a}\mathbb{E}_{\boldsymbol{\epsilon}_2}\left[\sum_{b}\epsilon_{2,b}^4B_{ab}^4+3\sum_{b\neq d}\epsilon_{2,b}^2\epsilon_{2,d}^2B_{ab}^2B_{ad}^2\right]$$

$$+ 3\mathbb{E}_U[U^2]^2\sum_{a\neq c}\mathbb{E}_{\boldsymbol{\epsilon}_2}\left[\sum_{b}\epsilon_{2,b}^4B_{ab}^2B_{cb}^2+\sum_{b\neq d}\epsilon_{2,b}^2\epsilon_{2,d}^2(B_{ab}^2B_{cd}^2+2B_{ab}B_{ad}B_{cb}B_{cd})\right]$$

$$= \mathbb{E}_U[U^4]^2\sum_{a,b}B_{ab}^4+3\mathbb{E}_U[U^2]^2\mathbb{E}_U[U^4]\left[\left(\sum_{a}\sum_{b\neq d}B_{ab}^2B_{ad}^2+\sum_{b}\sum_{a\neq c}B_{ab}^2B_{cb}^2\right)\right]$$

$$+ 3\mathbb{E}_U[U^2]^4\sum_{a\neq c}\sum_{b\neq d}\left(B_{ab}^2B_{cd}^2+2B_{ab}B_{ad}B_{cb}B_{cd}\right)$$

$$= \mathbb{E}_U[U^4]^2\sum_{a,b}B_{ab}^4+6\mathbb{E}_U[U^2]^2\mathbb{E}_U[U^4]\left[\sum_{a}\sum_{b\neq d}B_{ab}^2B_{ad}^2\right]$$

$$+3\mathbb{E}_U[U^2]^4 \sum_{a\neq c}\sum_{b\neq d}\left(B_{ab}^2 B_{cd}^2 + 2B_{ab}B_{ad}B_{cb}B_{cd}\right)$$

Also,

$$\mathbb{E}_{\boldsymbol{\epsilon}_1,\boldsymbol{\epsilon}_2}\left[\left(\boldsymbol{\epsilon}_2^T(A-A^T)\boldsymbol{\epsilon}_1\right)^2\right]^2 = \left(\mathbb{E}_U[U^2]^2\sum_{i,j}B_{ij}^2\right)^2$$

$$= \mathbb{E}_U[U^2]^4\sum_{i,j,r,s}B_{ij}^2 B_{rs}^2$$

$$= \mathbb{E}_U[U^2]^4\left(\sum_{i,j}\sum_s B_{ij}^2 B_{is}^2 + \sum_{j,s}\sum_{i\neq r}B_{ij}^2 B_{rs}^2\right)$$

$$= \mathbb{E}_U[U^2]^4\left(\sum_{i,j}\sum_s B_{ij}^2 B_{is}^2 + \sum_j \sum_{i\neq r}B_{ij}^2 B_{rj}^2 + \sum_{i\neq r}\sum_{j\neq s}B_{ij}^2 B_{rs}^2\right)$$

$$= \mathbb{E}_U[U^2]^4\left(\sum_{i,j}B_{ij}^4 + \sum_i \sum_{j\neq s}B_{ij}^2 B_{is}^2 + \sum_j \sum_{i\neq r}B_{ij}^2 B_{rj}^2 + \sum_{i\neq r}\sum_{j\neq s}B_{ij}^2 B_{rs}^2\right)$$

Therefore,

$$\mathrm{Var}\left(\left(\boldsymbol{\epsilon}_2^T(A-A^T)\boldsymbol{\epsilon}_1\right)^2\right) = \mathbb{E}_{\boldsymbol{\epsilon}_1,\boldsymbol{\epsilon}_2}\left[\left(\boldsymbol{\epsilon}_2^T(A-A^T)\boldsymbol{\epsilon}_1\right)^4\right] - \mathbb{E}_{\boldsymbol{\epsilon}_1,\boldsymbol{\epsilon}_2}\left[\left(\boldsymbol{\epsilon}_2^T(A-A^T)\boldsymbol{\epsilon}_1\right)^2\right]^2$$

$$= \left(\mathbb{E}_U[U^4]^2 - \mathbb{E}_U[U^2]^4\right)\sum_{a,b}B_{ab}^4 + 2\mathbb{E}_U[U^2]^2\left(3\mathbb{E}_U[U^4] - \mathbb{E}_U[U^2]^2\right)\sum_a\sum_{b\neq d}B_{ab}^2 B_{ad}^2$$

$$+2\mathbb{E}_U[U^2]^4\left(\sum_{a\neq c}\sum_{b\neq d}B_{ab}^2 B_{cd}^2 + 3\sum_{a\neq c}\sum_{b\neq d}B_{ab}B_{ad}B_{cb}B_{cd}\right)$$

$$= \mathrm{Var}(U^2)\left(\mathrm{Var}(U^2) + 2\left(\mathrm{Var}(U) + \mathbb{E}_U[U]^2\right)^2\right)\sum_{a,b}B_{ab}^4$$

$$+2\left(\mathrm{Var}(U) + \mathbb{E}_U[U]^2\right)^2\left(3\mathrm{Var}(U^2) + 2\left(\mathrm{Var}(U) + \mathbb{E}_U[U]^2\right)^2\right)\sum_a\sum_{b\neq d}B_{ab}^2 B_{ad}^2$$

$$+2\left(\mathrm{Var}(U) + \mathbb{E}_U[U]^2\right)^4\left(\sum_{a\neq c}\sum_{b\neq d}B_{ab}^2 B_{cd}^2 + 3\sum_{a\neq c}\sum_{b\neq d}B_{ab}B_{ad}B_{cb}B_{cd}\right)$$

$\square$

**Proposition 5.** *Let $U$ be the discrete random variable which takes the values $1, -1$ each with probability $1/2$. Then $(\boldsymbol{\epsilon}_2^T(A-A^T)\boldsymbol{\epsilon}_1)^2$ is the unbiased estimator of $\|A - A^T\|_F^2$. Moreover, $U$ is the unique random variable amongst zero-mean random variables for which the estimator is an unbiased estimator, and attains a minimum variance.*

***Proof of Proposition 5.*** A random variable $U^2$ has strictly positive variance if $U^2$ attains more than two values on a nonzero measure. To make $\mathrm{Var}(U^2) = 0$, the random variable should be a discrete variable which takes the values 1, -1 each with probability 1/2. $\square$

**Theorem 4** (De Bortoli et al. [15] and Guth et al. [22]). *Assume that there exists $M \geq 0$ such that for any $t \in [0, T]$ and $\mathbf{z} \in \mathbb{R}^d$, the score estimation is close enough to the forward score by $M$, $\|\mathbf{s}_{\boldsymbol{\theta}}(\mathbf{x}, t) - \nabla\log p_t^{\boldsymbol{\phi}}(\mathbf{x})\| \leq M$, with $\mathbf{s}_{\boldsymbol{\theta}} \in C([0, T] \times \mathbb{R}^d, \mathbb{R}^d)$. Assume that $\nabla\log p_t^{\boldsymbol{\phi}}(\mathbf{z})$ is $C^2$ in both $t$ and $\mathbf{z}$, and that $\sup_{\mathbf{z},t}\|\nabla^2\log p_t^{\boldsymbol{\phi}}(\mathbf{z})\| \leq K$ and $\|\frac{\partial}{\partial t}\nabla\log p_t^{\boldsymbol{\phi}}(\mathbf{z})\| \leq Me^{-\alpha t}\|\mathbf{z}\|$ for some $K, M, \alpha > 0$. Suppose $(\mathbf{h}_{\boldsymbol{\phi}}^{-1})_\#$ s a push-forward map. Then $\|p_r - (\mathbf{h}_{\boldsymbol{\phi}}^{-1})_\# p_{0,N}^{\boldsymbol{\theta}}\|_{TV} \leq E_{pri}(\boldsymbol{\phi}) + E_{dis}(\boldsymbol{\phi}) + E_{est}(\boldsymbol{\phi}, \boldsymbol{\theta})$, where $E_{pri}(\boldsymbol{\phi}) = \sqrt{2}e^{-T}D_{KL}(p_T^{\boldsymbol{\phi}}\|\pi)^{1/2}$ is the error originating from the prior mismatch; $E_{dis}(\boldsymbol{\phi}) = 6\sqrt{\delta}(1 + \mathbb{E}_{p_0^{\boldsymbol{\phi}}(\mathbf{z})}[\|\mathbf{z}\|^4]^{1/4})(1 + K + M(1 + \frac{1}{\sqrt{2\alpha}}))$ is the discretization error with $\delta = \frac{\max\gamma_k^2}{\min\gamma_k}$; $E_{est}(\boldsymbol{\phi}, \boldsymbol{\theta}) = 2TM^2$ is the score estimation error.*

*Remark 5.* Although the proof is based on the standard form of the Ornstein-Uhlenbeck process, the direct extension of the theorem holds for generic VPSDE if there exists $\bar{\beta} > 0$ such that $\frac{1}{\bar{\beta}} \leq \beta(t) \leq \bar{\beta}$. See De Bortoli [69].

**Lemma 4** (Lemma S11 of De Bortoli et al. [15]). *Let* $(\mathsf{E}, \mathcal{E})$ *and* $(\mathsf{F}, \mathcal{F})$ *be two measurable spaces and* $K : \mathsf{E} \times \mathcal{F} \to [0, 1]$ *be a Markov kernel. Then for any* $\mu_0, \mu_1 \in \mathcal{P}(\mathsf{E})$ *we have*

$$\|\mu_0 K - \mu_1 K\|_{TV} \leq \|\mu_0 - \mu_1\|_{TV}.$$

*In addition, for any* $\varphi : \mathsf{E} \to \mathsf{F}$ *measurable we get that*

$$\|\varphi_\# \mu_0 - \varphi_\# \mu_1\|_{TV} \leq \|\mu_0 - \mu_1\|_{TV},$$

*with equality if* $\varphi$ *is injective.*

**Proof of Theorem 4.** For any $k \in \{1, ..., N\}$, denote $R_k$ the Markov kernel such that for any $\mathbf{z} \in \mathbb{R}^d$, $\mathsf{A} \in \mathcal{B}(\mathbb{R}^d)$ and $k \in \{0, ..., N-1\}$ we have

$$R^{\boldsymbol{\theta}}_{k+1}(\mathbf{z}, \mathsf{A}) = (4\pi\gamma_{k+1})^{-d/2} \int_{\mathsf{A}} \exp\left[ -\frac{\|\tilde{\mathbf{z}} - \mathcal{T}^{\boldsymbol{\theta}}_{k+1}(\mathbf{z})\|^2}{4\gamma_{k+1}} \right] \mathrm{d}\tilde{\mathbf{z}},$$

where for any $\mathbf{z} \in \mathbb{R}^d$, $\mathcal{T}^{\boldsymbol{\theta}}_{k+1}(\mathbf{z}) = \mathbf{z} + \gamma_{k+1}\{\mathbf{z} + 2\mathbf{s}_{\boldsymbol{\theta}}(\mathbf{z}, t_k)\}$, where $t_k = \sum_l^{k-1} \gamma_l$. Define $Q^{\boldsymbol{\theta}}_N = \prod_{l=1}^N R^{\boldsymbol{\theta}}_l$. Analogously, let us define

$$R^{\boldsymbol{\phi}}_{k+1}(\mathbf{z}, \mathsf{A}) = (4\pi\gamma_{k+1})^{-d/2} \int_{\mathsf{A}} \exp\left[ -\frac{\|\tilde{\mathbf{z}} - \mathcal{T}^{\boldsymbol{\phi}}_{k+1}(\mathbf{z})\|^2}{4\gamma_{k+1}} \right] \mathrm{d}\tilde{\mathbf{z}},$$

for $\mathcal{T}^{\boldsymbol{\phi}}_{k+1}(\mathbf{z}) = \mathbf{z} + \gamma_{k+1}\{\mathbf{z} + 2\nabla \log p^{\boldsymbol{\phi}}_t(\mathbf{z}, t_k)\}$ and $Q^{\boldsymbol{\phi}}_N = \prod_{l=1}^N R^{\boldsymbol{\phi}}_l$.

Suppose $\mathbb{P}_{T|0}$ is the transition kernel from time zero to $T$ and $\mathbb{P}^R$ is the reverse-time measure, i.e., for any $\mathsf{A} \in \mathcal{B}(\mathcal{C})$ we have $\mathbb{P}^R(\mathsf{A}) = \mathbb{P}(\mathsf{A}^R)$ with $\mathsf{A}^R = \{t \mapsto \omega(T - t) : \omega \in \mathsf{A}\}$. Then,

$$p^{\boldsymbol{\phi}}_0 \mathbb{P}_{T|0} \mathbb{P}^R_{T|0}(\mathsf{A}) = \mathbb{P}_T \mathbb{P}^R_{T|0}(\mathsf{A}) = \mathbb{P}^R_0 \mathbb{P}^R_{T|0}(\mathsf{A}) = \mathbb{P}^R_T(\mathsf{A}) = p^{\boldsymbol{\phi}}_0(\mathsf{A}). \tag{52}$$

Combining Eq. (52) with Lemma 4, we have

$$\begin{aligned}
\|p^{\boldsymbol{\phi}}_0 - p^{\boldsymbol{\theta}}_{0,N}\|_{TV} &= \|p^{\boldsymbol{\phi}}_0 \mathbb{P}_{T|0} \mathbb{P}^R_{T|0} - \pi Q^{\boldsymbol{\theta}}_N\|_{TV} \\
&\leq \|p^{\boldsymbol{\phi}}_0 \mathbb{P}_{T|0} \mathbb{P}^R_{T|0} - \pi \mathbb{P}^R_{T|0}\|_{TV} + \|\pi \mathbb{P}^R_{T|0} - \pi Q^{\boldsymbol{\phi}}_N\|_{TV} + \|\pi Q^{\boldsymbol{\phi}}_N - \pi Q^{\boldsymbol{\theta}}_N\|_{TV} \\
&\leq \underbrace{\|p^{\boldsymbol{\phi}}_0 \mathbb{P}_{T|0} - \pi\|_{TV}}_{E_{pri}} + \underbrace{\|\pi \mathbb{P}^R_{T|0} - \pi Q^{\boldsymbol{\phi}}_N\|_{TV}}_{E_{dis}} + \underbrace{\|\pi Q^{\boldsymbol{\phi}}_N - \pi Q^{\boldsymbol{\theta}}_N\|_{TV}}_{E_{est}} .
\end{aligned}$$

The first two terms, $E_{pri}(\boldsymbol{\phi}) + E_{dis}(\boldsymbol{\phi})$, are those terms derived in Theorem 2 of Guth et al. [22]. By Lemma S13 of De Bortoli et al. [15], the last term, $E_{est}(\boldsymbol{\phi}, \boldsymbol{\theta})$, is bounded by

$$\|\pi Q^{\boldsymbol{\phi}}_N - \pi Q^{\boldsymbol{\theta}}_N\|^2_{TV} \leq \frac{1}{2} \int_0^T \mathbb{E}\big[\|b_{\boldsymbol{\phi}}(\{\mathbf{z}_t\}^T_{t=0}, t) - b_{\boldsymbol{\theta}}(\{\mathbf{z}_t\}^T_{t=0}, t)\|^2\big] \mathrm{d}t,$$

where $b_{\boldsymbol{\phi}}(\{\mathbf{z}_t\}^T_{t=0}, t) = \sum_{k=0}^{N-1} 1_{[t_k, t_{k+1})}(t)\{\mathbf{z}_{t_k} + 2\log p^{\boldsymbol{\phi}}_t(\mathbf{z}_{t_k})\}$ and $b_{\boldsymbol{\theta}}(\{\mathbf{z}_t\}^T_{t=0}, t) = \sum_{k=0}^{N-1} 1_{[t_k, t_{k+1})}(t)\{\mathbf{z}_{t_k} + 2\mathbf{s}_{\boldsymbol{\theta}}(\mathbf{z}_{t_k}, t_k)\}$ are the drift terms of piecewise generative processes, given by

$$\mathrm{d}\mathbf{z}_t = \left[ -\mathbf{z}_t - 2\nabla \log p^{\boldsymbol{\phi}}_{t_k}(\mathbf{z}_{t_k}) \right] \mathrm{d}\bar{t} + g(t) \, \mathrm{d}\bar{\mathbf{w}}_t$$

and

$$\mathrm{d}\mathbf{z}_t = \left[ -\mathbf{z}_t - 2\mathbf{s}_{\boldsymbol{\theta}}(\mathbf{z}_{t_k}, t_k) \right] \mathrm{d}\bar{t} + g(t) \, \mathrm{d}\bar{\mathbf{w}}_t$$

defined each of the interval $[t_k, t_{k+1}]$ for $k = 0, ..., N-1$, respectively. Therefore, $E_{est}(\boldsymbol{\phi}, \boldsymbol{\theta})$ is bounded by

$$\begin{aligned}
E_{est}(\boldsymbol{\phi}, \boldsymbol{\theta}) &\leq \frac{1}{2} \int_0^T \mathbb{E}\big[\|b_{\boldsymbol{\phi}}(\{\mathbf{z}_t\}^T_{t=0}, t) - b_{\boldsymbol{\theta}}(\{\mathbf{z}_t\}^T_{t=0}, t)\|^2\big] \mathrm{d}t \\
&= 2 \sum_{k=0}^{N-1} \int_{t_k}^{t_{k+1}} \mathbb{E}\big[\|\nabla \log p^{\boldsymbol{\phi}}_{t_k}(\mathbf{z}_{t_k}) - \mathbf{s}_{\boldsymbol{\theta}}(\mathbf{z}_{t_k}, t_k)\|^2\big] \mathrm{d}t
\end{aligned}$$

Table 21: Performance comparison to linear/nonlinear diffusion models on CIFAR-10. We report both before/after correction of density estimation performances. We report the baseline performances of linear diffusions by training our PyTorch implementation based on Song et al. [1, 11] with identical hyperparameters and networks on both linear/nonlinear diffusions in order to quantify the effect of nonlinearity in a fair setting. Boldface numbers represent the best performance in a column, and underlined numbers represent the second best.

| SDE | Model | Nonlinear Data Diffusion | # Params | NLL ($\downarrow$) after correction | NLL ($\downarrow$) before correction | NELBO ($\downarrow$) w/ residual (after) | NELBO ($\downarrow$) w/o residual (before) | Gap ($\downarrow$) (=NELBO-NLL) after | Gap ($\downarrow$) (=NELBO-NLL) before | FID ($\downarrow$) ODE | FID ($\downarrow$) PC |
|---|---|---|---|---|---|---|---|---|---|---|---|
| VE | NCSN++ (FID) | ✗ | 63M | 4.86 | 3.66 | 4.89 | 4.45 | 0.03 | 0.79 | - | 2.38 |
|  | INDM (FID) | ✓ | 76M | 3.22 | 3.13 | 3.28 | 3.24 | 0.06 | 0.11 | - | 2.29 |
|  | NCSN++ (deep, FID) | ✗ | 108M | 4.85 | 3.45 | 4.86 | 4.43 | 0.01 | 0.98 | - | **2.20** |
|  | INDM (deep, FID) | ✓ | 118M | 3.13 | 3.03 | 3.14 | 3.10 | 0.01 | 0.07 | - | 2.28 |
| VP | DDPM++ (FID) | ✗ | 62M | 3.21 | 3.16 | 3.34 | 3.32 | 0.13 | 0.16 | 3.90 | 2.89 |
|  | INDM (FID) | ✓ | 75M | 3.17 | 3.11 | 3.23 | 3.18 | 0.06 | 0.07 | **3.61** | 2.90 |
|  | DDPM++ (deep, FID) | ✗ | 108M | 3.19 | 3.13 | 3.32 | 3.29 | 0.13 | 0.16 | 3.69 | 2.64 |
|  | INDM (deep, FID) | ✓ | 121M | 3.09 | 3.02 | 3.13 | 3.08 | 0.04 | 0.06 | 3.67 | 3.15 |
|  | DDPM++ (NLL) | ✗ | 62M | 3.03 | 2.97 | 3.13 | 3.11 | 0.10 | 0.14 | 6.70 | 5.17 |
|  | INDM (NLL) | ✓ | 75M | 2.98 | 2.95 | 2.98 | 2.97 | 0.00 | 0.02 | 6.01 | 5.30 |
|  | INDM (NLL, ST) | ✓ | 75M | 3.01 | 2.98 | 3.02 | 3.01 | 0.01 | 0.03 | 3.88 | 3.25 |
|  | DDPM++ (deep, NLL) | ✗ | 108M | 3.01 | 2.95 | 3.11 | 3.09 | 0.10 | 0.14 | 6.43 | 4.88 |
|  | INDM (deep, NLL) | ✓ | 121M | **2.97** | **2.94** | **2.97** | **2.96** | **0.00** | **0.02** | 5.71 | 4.79 |

Table 22: Performance comparison on CIFAR-10.

| Class | SDE | Type | Model | NLL ($\downarrow$) after correction | NLL ($\downarrow$) before correction | NELBO ($\downarrow$) w/ residual (after) | NELBO ($\downarrow$) w/o residual (before) | Gap ($\downarrow$) (=NELBO-NLL) after | Gap ($\downarrow$) (=NELBO-NLL) before | FID ($\downarrow$) ODE | FID ($\downarrow$) PC |
|---|---|---|---|---|---|---|---|---|---|---|---|
| GAN | | | StyleGAN2 + ADA [70] | - | - | - | - | - | - | 2.92 | |
|  | | | StyleFormer [32] | - | - | - | - | - | - | 2.82 | |
|  | | | SNGAN + DGflow [71] | - | - | - | - | - | - | 9.62 | |
|  | | | TransGAN [72] | - | - | - | - | - | - | 9.26 | |
| Autoregressive | | | PixelCNN [73] | 3.14 | - | - | - | - | - | 65.9 | |
|  | | | PixelRNN [73] | 3.00 | - | - | - | - | - | - | |
|  | | | Sparse Transformer [74] | 2.80 | - | - | - | - | - | - | |
| Flow | | | Glow [56] | 3.35 | - | - | - | - | - | 48.9 | |
|  | | | Residual Flow [23] | 3.28 | - | - | - | - | - | 46.4 | |
|  | | | Flow++ [26] | 3.28 | - | - | - | - | - | 46.4 | |
|  | | | Wolf [24] | 3.27 | - | - | - | - | - | 37.5 | |
|  | | | VFlow [75] | 2.98 | - | - | - | - | - | - | |
|  | | | DenseFlow-74-10 [4] | 2.98 | - | - | - | - | - | 34.9 | |
| VAE | | | NVAE [6] | - | - | 2.91 | - | - | - | 23.5 | |
|  | | | Very Deep VAE [76] | - | - | 2.87 | - | - | - | - | |
|  | | | δ-VAE [77] | - | - | 2.83 | - | - | - | - | |
|  | | | DCVAE [78] | - | - | - | - | - | - | 17.9 | |
|  | | | CR-NVAE [31] | - | - | - | - | - | - | 2.51 | |
| Diffusion | | Linear | DDPM [8] | - | - | 3.75 | - | - | - | 3.17 | |
|  | | | NCSNv2 [7] | - | - | - | - | - | - | 10.87 | |
|  | | | DDIM [79] | - | - | - | - | - | - | 4.04 | |
|  | | | IDDPM [60] | 3.37 | - | - | - | - | - | 2.90 | |
|  | | | VDM [28] | **2.65** | - | - | - | - | - | 7.41 | |
|  | | | NCSN++ (FID) [1] | 4.85 | 3.45 | 4.86 | 4.43 | 0.01 | 0.98 | - | **2.20** |
|  | | | DDPM++ (FID) [1] | 3.19 | 3.13 | 3.32 | 3.29 | 0.13 | 0.16 | 3.69 | 2.64 |
|  | | | DDPM++ (NLL) [11] | 3.01 | 2.95 | 3.11 | 3.09 | 0.10 | 0.14 | 6.43 | 4.88 |
|  | | | CLD-SGM [20] | - | - | - | 3.31 | - | - | 2.25 | - |
|  | | SBP | SB-FBSDE [16] | - | 2.98 | - | - | - | - | - | 3.18 |
|  | Nonlinear | VAE-based | LSGM (FID) [9] | - | - | 3.45 | 3.43 | - | - | **2.10** | - |
|  | | | LSGM (NLL)-269M | - | - | - | 2.97 | - | - | 6.15 | - |
|  | | | LSGM (NLL) | - | - | **2.87** | **2.87** | - | - | 6.89 | - |
|  | | | LSGM (balanced)-109M | - | - | - | 2.96 | - | - | 4.60 | - |
|  | | | LSGM (balanced) | - | - | 2.98 | 2.95 | - | - | 2.17 | - |
|  | | Flow-based | DiffFlow (FID) [13] | - | - | 3.04 | - | - | - | - | 14.14 |
|  | | | INDM (FID) | 3.13 | 3.03 | 3.14 | 3.10 | 0.01 | 0.07 | - | 2.28 |
|  | | | INDM (NLL) | 2.97 | **2.94** | 2.97 | 2.96 | **0.00** | **0.02** | 5.71 | 4.79 |
|  | | | INDM (ST) | 3.01 | 2.98 | 3.02 | 3.01 | 0.01 | 0.03 | 3.88 | 3.25 |

$$\leq 2TM^2.$$

Now, from Lemma 4 and the invertibility of the flow transformation, we have

$$\|p_r - (\mathbf{h}_\phi^{-1})_\# \circ p_{0,N}^\theta\|_{TV} = \|(\mathbf{h}_\phi)_\# \circ p_r - p_{0,N}^\theta\|_{TV} = \|p_0^\phi - p_{0,N}^\theta\|_{TV},$$

which completes the proof. □

Table 23: Performance comparison on CelebA $64 \times 64$.

| Model | NLL ($\downarrow$) | | NELBO ($\downarrow$) | | Gap ($\downarrow$) | | FID ($\downarrow$) | |
| | after | before | w/ res- | w/o res- | after | before | ODE | PC |
|---|---|---|---|---|---|---|---|---|
| UNCSN++ [27] | - | **1.93** | - | - | - | - | - | **1.92** |
| DDGM [29] | - | - | - | - | - | - | - | 2.92 |
| Efficient-VDVAE [30] | - | **1.83** | - | - | | | | |
| CR-NVAE [31] | - | - | 1.86 | - | - | - | - | - |
| DenseFlow-74-10 [4] | 1.99 | - | - | - | - | - | - | - |
| StyleFormer [32] | - | - | - | - | - | - | 3.66 | |
| NCSN++ (VE) | 3.41 | 2.37 | 3.42 | 3.96 | 0.01 | 1.59 | - | 3.95 |
| INDM (VE, FID) | 2.31 | 1.95 | 2.33 | 2.17 | 0.02 | 0.22 | - | 2.54 |
| DDPM++ (VP, FID) | 2.14 | 2.07 | 2.21 | 2.22 | 0.06 | 0.14 | 2.32 | 3.03 |
| INDM (VP, FID) | 2.27 | 2.13 | 2.31 | 2.20 | 0.04 | 0.07 | **1.75** | 2.32 |
| DDPM++ (VP, NLL) | 2.00 | 1.93 | 2.09 | 2.09 | 0.09 | 0.16 | 3.95 | 5.31 |
| INDM (VP, NLL) | 2.05 | 1.97 | 2.05 | 2.00 | 0.00 | 0.03 | 3.06 | 5.14 |

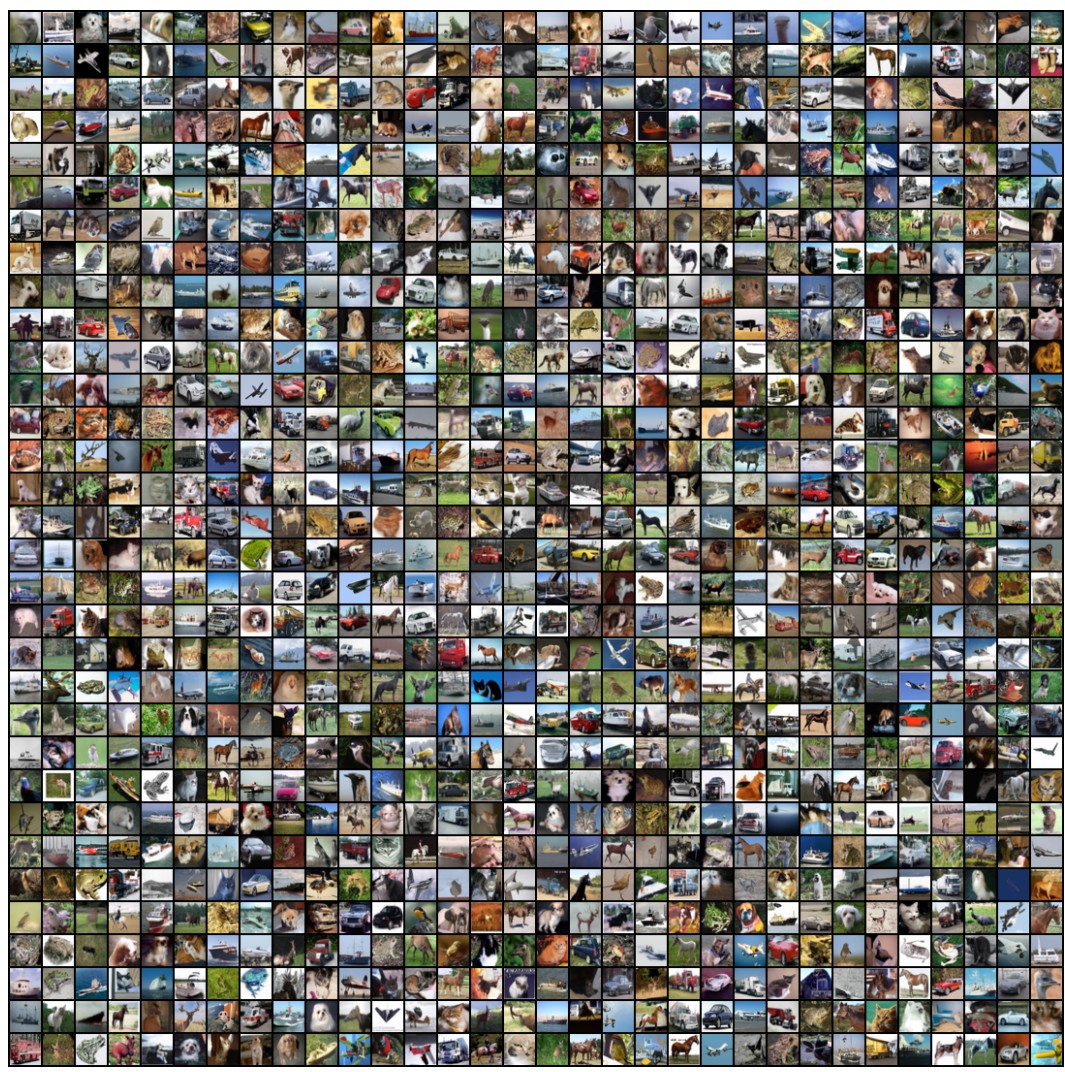

Figure 20: Non cherry-picked random samples from CIFAR-10 trained on INDM (VE, deep, FID).

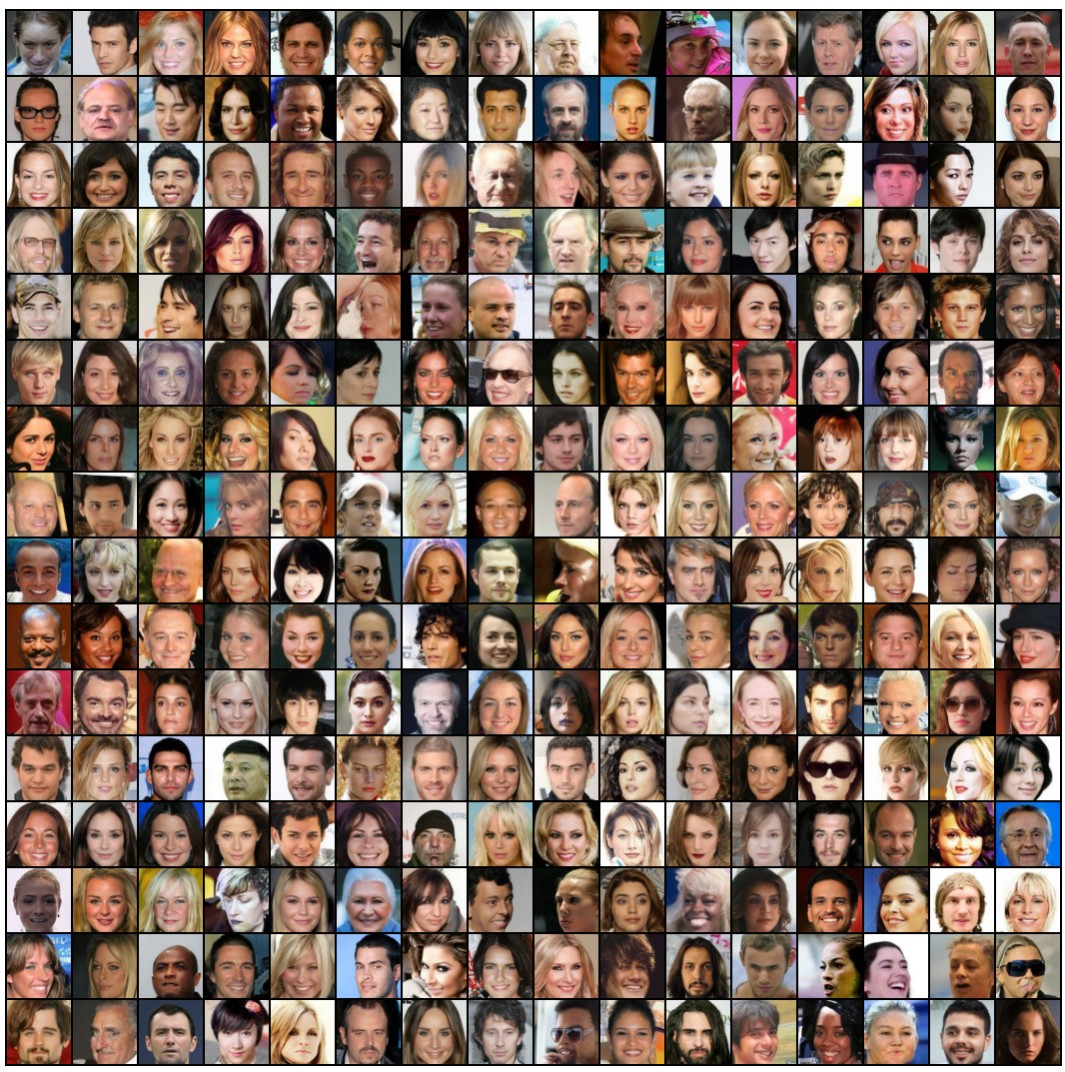

Figure 21: Non cherry-picked random samples fr om CelebA trained on INDM (VP, FID).