# OpenReview forum: "Maximum Likelihood Training of Implicit Nonlinear Diffusion Model"
_NeurIPS.cc/2022/Conference — NeurIPS 2022 Accept_

### Official Review · Reviewer_19AU · 2022-07-09

**Rating:** 6
**Confidence:** 4
**Soundness:** 3 good
**Presentation:** 3 good
**Contribution:** 3 good

**Summary:**

This paper proposes to combine normalizing flows and denoising diffusion models in which the data distribution is first transformed into a latent space by normalizing flows and then the latent distribution is modeled by generative diffusion models. By leveraging Ito's theorem, it is proved that the resulting diffusion process is non-linear in the original data space. Therefore, the authors call it implicit nonlinear diffusion models (INDM). Experimentally, the proposed INDM achieves zero variational gap and state-of-the-art FID on the CelebA dataset.

**Questions:**

I like figure 9. It tells me to some extent what is learned in the latent space. Could you provide more cases like that, say the results on the datasets like two moons, pinwheel, checkerboard ... Also, an illustration of Cifar and CelebA is also important. From figure 3, it seems that the flow models just learn some color transformation. In this regard, the flow models make tiny contributions to data fitting. Could you show more cases on that?

**Limitations:**


- The model archiecture is similar to LSGM and the benefit of flows is unclear.

- The results look convincing enough. Though, the onerous discussion make me hesitate about its practical value. The authors should explain their approach in simple and concise language.


**Strengths And Weaknesses:**

**Strengths**

The application of Ito's theorem is exactly interesting. It shows a strong connection between diffusion models and normalizing flows. And I think this connection is intuitive since the diffusion models can be considered a special case of flow models. Moreover, the reduction of the variational gap is also intriguing by applying normalizing flows.


**Weaknesses**

- The benefit of normalizing flows is unclear. Although the authors try to explain by the means of optimal transport and relative energy, it still lacks an intuitive understanding of why flows+diffusion makes sense, say does the transformed latent space make it easier for diffusion models to learn? One virtue of normalizing flows is to transform complex data distribution into a simple form, then one could learn a model on the simple manifold. However, it is unclear what is learned by flow models in INDM and how flow models increase the efficiency of learning and sampling.

- It seems that the authors try to hind something. I think the main contribution of this paper is to show that the combination of normalizing flows can improve the expressivity of diffusion models. However, the authors do not discuss the benefit of normalizing flows thoroughly, but use a lot of vague discussions to make readers suck.

    - Sec6.1 shows a better converge on NLL and NELBO. But as mentioned before, it's unclear why the normalizing flow benefits this. The author tries to contribute this to the "bidirectional attraction". But after reading the main paper, I can't understand how the flow models can be used to reduce the variational gaps, say NLL=NELBO, and make the learning curve of INDM more MLE.

    - Sec6.2 shows the diffusion process of INDM is nonlinear. However, I don't think this illustration deserves to be shown in the main paper because Ito's theorem has told us everything. The appearance of sec6.2 just traps the reader in the details of the eigenvalues of the covariance matrix.

    - Sec6.3 extremely distracts my focus. I don't think the connection of optimal transport makes good contribute to the INDM. I must admit that I simply ran out of energy and gave up on close reading by the time that I got to the description of optimal transport in section 6.3. It is okay to show this connection, although it is still intuitive as I understand. But it is worth appearing in the appendix. Understanding this part in the main paper just makes readers feel distracted.

    - It seems that the relative energy is a good way to show the benefit of normalizing flows. This part deserves further discussion. However, I find the discussion on it is also quite vague. To be honest, I cannot fully understand smaller relative energy contributes to faster sampling. I think the authors should explain this in simple and concise language and give only the necessary supporting evidence for their claims.

- INDM seems an incremental improvement of LSGM. The diffusion and generative processes are similar. The only difference lies in that LSGM is based on the car but INDM is based on the normalizing flows. Just push back, the benefit of learning diffusion models on the latent space transported by flows is unclear, considering that the benefit of learning on latent space of LSGM is apparent, which learns an expressive prior for VAE and simple latent space for diffusion models.

---

> ### Author Response · Authors · 2022-08-02
> **Continued**
>
> **Q6. [Comparison with LSGM]** *INDM seems an incremental improvement of LSGM. The diffusion and generative processes are similar. The only difference lies in that LSGM is based on the car but INDM is based on the normalizing flows. Just push back, the benefit of learning diffusion models on the latent space transported by flows is unclear, considering that the benefit of learning on latent space of LSGM is apparent, which learns an expressive prior for VAE and simple latent space for diffusion models.*
> \
> \
> **A.**
> The key aspect of IDNM is enabling the nonlinear diffusion with closed-form drift/diffusion terms, which becomes only feasible by flow because of its exact invertibility property. This closed-form nonlinear diffusion categorizes INDM as one of the nonlinear diffusions, contrastive to LSGM which is not an algorithm of nonlinear diffusions. Moreover, this closed-form nonlinear diffusion becomes the key source of the success of the dataset interpolation task in Section 7.1 of the revised manuscript. Given the inability of dataset interpolation by LSGM and the ability of dataset interpolation by INDM, INDM is not an incremental improvement of LSGM.
>
> To support the above argument, given the loss function of line 333 in the originally submitted main paper, we construct a loss function of LSGM for dataset interpolation by replacing 1) IDNM NELBO into LSGM NELBO, and 2) Interpolation loss into VAE NELBO. Figure 9 of the revised manuscript empirically shows that LSGM is suboptimal to INDM, and we attribute this to the closed-ness of INDM. To be concrete, INDM builds a diffusion bridge between two datasets, but LSGM has no such bridge in general because of the broken invertibility.
>
> Also, the reviewer points out that the benefit of LSGM is an expressive prior for VAE. This is correct because the diffusion model in LSGM models the prior distribution of VAE. However, this is exactly the same in INDM because our diffusion model is modeling the prior distribution of flow. In fact, if we compare the required latent dimension, the below table indicates that LSGM requires 15x more latent dimension than INDM on CIFAR-10.
>
> | Dataset | Dataset Dimension | Latent Dimension of INDM | Latent Dimension of LSGM |
> |--------------------|-----|-----|-----|
> | MNIST | 784 | 784 | 2,560 |
> | (Binarized) OMNIGLOT | 11,025 | 11,025 | 15,360 |
> | CIFAR-10 | 3,072 | 3,072 | 46,080 |
> | CelebA-HQ 256 | 196,608 | 196,608 | 819,200 |
>
> \
> \
> \
> **Q7. [Toy Dataset Illustration]** *I like figure 9. It tells me to some extent what is learned in the latent space. Could you provide more cases like that, say the results on the datasets like two moons, pinwheel, checkerboard ...*
> \
> \
> **A.**
> It is very interesting to visualize the latent diffusion process in 2d toy examples. We provide the result on the toy 2d checkerboard dataset in Figure 15 of the revised Appendix. The checkerboard dataset is particularly interesting because the data manifold has a singularity at the origin. Figure 15 describes the diffusion processes of both data and latent manifolds by training steps. We could observe that the data singularity disappears at the latent manifold. Also, as training proceed, the latent manifold is inflated, which is observed in CIFAR-10, as well (in Table 6 of the original/revised Appendix). This inflated latent helps the sampling robustness by discretization, as we described by (Reasoning #1)-paragraph of **Q1** (Figure 12 to get a quick intuition). Also, we additionally analyze additional aspects on the diffusion trajectory with a toy 2d case in Figure 13 and lines 227 - 246 of the revised Appendix.
>
> \
> \
> \
> **Q8. [Role of Flow]** *Also, an illustration of Cifar and CelebA is also important. From figure 3, it seems that the flow models just learn some color transformation. In this regard, the flow models make tiny contributions to data fitting. Could you show more cases on that?*
> \
> \
> **A.**
> As the image ignores the values smaller than 0 and larger than 1, we have normalized the latent vector into the feasible space, $[0,1]^{d}$, in order to visualize the latent vector in Figure 3. This normalization possibly hinders the true effect of flow transformation. Table 6 of the originally submitted Appendix B.4 presents the statistics of the unnormalized latent space, and this means that the latent manifold is pushed outside of the prior manifold by the flow transformation (Figure 12 of the revision). From this property, INDM becomes more robust on time discretization in sampling, which is explained in **Q1** with detailed arguments. Moreover, the flow transforms a data manifold into a smoother latent manifold up to the second derivatives in Figure 14 of the revision, and this could not be seen from the visualization in Figure 3. We clarify that Figure 3 is visualizing a *normalized* latent in our revision, and we provide additional illustrations of this normalized latent in Figure 16 of the revision.

---

> > ### Comment · Reviewer_19AU · 2022-08-07
> > **Answer to the comments from the authors**
> >
> > Thank you for the detailed answers. There are some points still confusing me.
> >
> > > **In your answer to Q6** This closed-form nonlinear diffusion categorizes INDM as one of the nonlinear diffusions, contrastive to LSGM which is not an algorithm of nonlinear diffusions.
> >
> > I am concerned about whether this statement is true or not. It is alright that INDM is a nonlinear diffusion with closed forms. But I think LSGM is also a nonlinear diffusion, although there is no closed form, as Ito's theorem doesn't require the invertible transformation. Is it true?
> >
> > Furthermore, I am curious about your sumarization of the latent dimension of LSGM. It seems that the latent dimension in LSGM is larger than the original input images.  However, in the [table 7 of the LSGM paper](https://arxiv.org/pdf/2106.05931.pdf), the dimension of the latent space is smaller than that of the input images. Do I miss something?

---

> > > ### Author Response · Authors · 2022-08-07
> > > **Answer to the Reviewer's comments**
> > >
> > > **Q9. [Nonlinear Data Diffusion?]** *But I think LSGM is also a nonlinear diffusion, although there is no closed form, as Ito's theorem doesn't require the invertible transformation. Is it true?*
> > > \
> > > \
> > > **A.** We agree with the reviewer's point. LSGM applies the diffusion process in latent space, where the data is transported to latent via the encoder map of VAE. Having that this encoder map is trained highly nonlinearly, there clearly exists nonlinearity in the data diffusion if we transport the latent diffusion to the data space. We acknowledge and respect the reviewer's point.
> > >
> > > Our original intention could be understood if we observe the diffusion process in a relatively strict view. By a strict (mathematical) definition, a diffusion process is a sequence of random variables which is connected via a Markov chain (or an SDE in a continuous diffusion). According to this strict view, one needs to satisfy two requirements to call it a diffusion process:
> > >
> > > > (Condition #1) there are multiple (possibly infinite) number of random variables;
> > >
> > > > (Condition #2) the random variables are connected via a Markov chain.
> > >
> > > To further analyze, let's enumerate all the random variables in LSGM. First, there is a data variable, $x_{0}$. Second, through the encoder map, there is a latent variable, $z_{0}:=E(x_{0})$. Third, the latent forward diffusion constructs a sequence of random variables, $z_{t}$, for all $t\in[0,T]$. Fourth, the latent generative diffusion builds the reverse random variables, $z_{t}^{-}$, for all $t\in[0,T]$. Finally, the decoder constructs the reconstructed variable, $x_{rec}=(D\circ E)(x_{0})$.
> > >
> > > Now, in a strict view, could we build multiple random variables that construct the *forward* process? Unfortunately, Lemma 3 of the initially submitted Appendix claims that the answer is negative. According to Lemma 3, there is *no inverse function* of the encoder map as long as the data dimension and the latent dimension are different, and the sequence of latent variables cannot be transported to the data space. Therefore, from this aspect, in LSGM, we cannot build a forward data diffusion process because (Condition #1) is not satisfied.
> > >
> > > On the other hand, in INDM, we could build multiple random variables in the data space through the inverse of the flow map (see lines 102 - 103 of the initial submission). Satisfying (Condition #1) is the original motivation for using the invertible normalizing flow. After constructing these random variables, we could apply Ito's Lemma to derive the governing SDE of the derived data variables in lines 104 - 106.
> > >
> > > Instead of the encoder map, one could construct the sequence of data variables with the decoder map, $x_{t}^{D}=D(z_{t})$. With this construction, this sequence of random variables is a data diffusion process. However, the initial variable of this process follows the reconstructed variable, $x_{0}^{D}=D(z_{0})=D(E(x_{0}))=(D\circ E)(x_{0})=x_{rec}$, not the original data variable. Therefore, in our strict view, this is not the forward data diffusion process.
> > >
> > > We acknowledge and respect the reviewer's comment. As the reviewer mentioned, LSGM transforms the data variable nonlinearly to the initial latent variable, which is diffused linearly. Therefore, it is possible to conclude that the data is conceptually diffused nonlinearly. We agree with the reviewer's view and would revise our manuscript by mentioning the reviewer's view upon our' view. We sincerely thank the reviewer's comment.
> > >
> > > \
> > > \
> > > \
> > > **Q10. [Latent Dimension of LSGM]** *Furthermore, I am curious about your sumarization of the latent dimension of LSGM. It seems that the latent dimension in LSGM is larger than the original input images. However, in the table 7 of the LSGM paper, the dimension of the latent space is smaller than that of the input images. Do I miss something?*
> > > \
> > > \
> > > **A.** Like the reviewer, we also had a hard time figuring out the latent dimension of LSGM. The above table is based on the released code and checkpoints of LSGM. The latent dimension of the above table differs from the *spatial dims. of $z$ in each scale*, and we find that this is because LSGM applies their diffusion process on every groups and channels. In other words, the formula for the LSGM's latent dimension is "(*# groups in each scale*) $\times$ (*spatial dims. of $z$ in each scale*) $\times$ (*# channel in $z$*)". For instance, in CIFAR-10 (best FID), the latent dimension is $20\times 16^{2}\times 9=46,080$. In CIFAR-10 (best NLL), the latent dimension is $4\times 16^{2}\times 45=46,080$.
> > >
> > > For CelebA-HQ, LSGM applies the diffusion process only on the top-scale. There are a pair of scales in CelebA-HQ (best qualitative) with spatial dims of $128^{2}$ and $64^{2}$. Among these top/bottom-scales, LSGM only models the top-scale latent with the diffusion model, and leaves the bottom-scale latent unchanged from its backbone VAE structure. Therefore, in this case, the latent dimension should be $10\times 64^{2}\times 20=819,200$.

---

> > > > ### Comment · Reviewer_19AU · 2022-08-08
> > > > **Thanks for your response**
> > > >
> > > > Thanks for your detailed response. I raise my score to weak accept since some of my concerns have been addressed. However, as the reviewer befw said, some clarifications are not clear and seem misleading. I can't fully confirm whether all claims are correct.

---

> > > > > ### Author Response · Authors · 2022-08-09
> > > > > **Thank you for the affirmative evaluation**
> > > > >
> > > > > We hugely thank the reviewer for such a constructive review. We would reflect all the reviewer's comments on the final revision.

---

> ### Author Response · Authors · 2022-08-02
> **Continued**
>
> **Q3. [Unnecessary Illustration]** *Sec6.2 shows the diffusion process of INDM is nonlinear. However, I don't think this illustration deserves to be shown in the main paper because Ito's theorem has told us everything. The appearance of sec6.2 just traps the reader in the details of the eigenvalues of the covariance matrix.*
> \
> \
> **A.**
> We agree that Section 6.2 has no further information on the design of INDM. However, still readers may question whether the nonlinear diffusion would be meaningfully trained, or not, in the benchmark datasets. Section 6.2 serves this utilization-aspect explanation with some observed data. If the reviewer insists, we are willing to remove Section 6.2 to describe more details on the model training, instead.
>
> \
> \
> \
> **Q4. [Distracting Argument]** *Sec6.3 extremely distracts my focus. I don't think the connection of optimal transport makes good contribute to the INDM. I must admit that I simply ran out of energy and gave up on close reading by the time that I got to the description of optimal transport in section 6.3. It is okay to show this connection, although it is still intuitive as I understand. But it is worth appearing in the appendix. Understanding this part in the main paper just makes readers feel distracted.*
> \
> \
> **A.**
> The original intention of Section 6.3 is to provide the empirical reasoning for why INDM is robust on the sampling step size. As the reviewer is wondering, regardless of whether the transportation is optimal or not, the straightness of the diffusion trajectory is what governs the sampling robustness. Thank you for your review, and we revised Section 6.3 to exclude the connection of optimal transport with INDM. Our revised version observes that the sampling is largely robust on the tolerance, and we delve into this robustness with respect to the straightness of diffusion trajectory, of which we believe to be an important characteristic of INDM.
>
> \
> \
> \
> **Q5. [Relative Energy]** *It seems that the relative energy is a good way to show the benefit of normalizing flows. This part deserves further discussion. However, I find the discussion on it is also quite vague. To be honest, I cannot fully understand smaller relative energy contributes to faster sampling. I think the authors should explain this in simple and concise language and give only the necessary supporting evidence for their claims.*
> \
> \
> **A.**
> The shorter the relative energy, the shorter the path. A shorter path in the Euclidean space is more likely to be a straight line, but as the reviewer pointed out, currently, the evidence does not support the connection between relative energy and straight-ness in a concrete manner. In our paper revision, we focus more on the straight-ness of INDM’s diffusion trajectory, than on the optimal transport.

---

> ### Author Response · Authors · 2022-08-02
> **Continued**
>
> **Q2. [Why MLE Training?]** *Sec6.1 shows a better converge on NLL and NELBO. But as mentioned before, it's unclear why the normalizing flow benefits this. The author tries to contribute this to the "bidirectional attraction". But after reading the main paper, I can't understand how the flow models can be used to reduce the variational gaps, say NLL=NELBO, and make the learning curve of INDM more MLE.*
> \
> \
> **A.**
> The reviewer points out that it is unclear if the flow training indeed helps reduce the variational gap (NELBO-NLL). Unveiling why INDM training is nearly MLE is a difficult problem because a concrete theoretic analysis of deep learning is intractable in general. Thus, instead of discovering any concrete theorem on MLE training, we limit ourselves to partially explain the MLE training from the perspective of our loss design, which is presented in Appendix B.2 of the original submission. To begin with, it is important to note that any score network ($s_{\theta}$) is decomposed into two parts, according to the Helmholtz Theorem (Lemma 1 of Appendix B.2): a score function part ($\nabla\log{p_{t}^{\theta}}$) and a remaining part ($u_{t}^{\theta}$), i.e., $s_{\theta}=\nabla\log{p_{t}^{\theta}}+u_{t}^{\theta}$. The remaining residual part is nothing to do with the score function, and it ($u_{t}^{\theta}$) should collapse to zero if the score network perfectly matches the data score (lines 83 - 85 in Appendix of the initial submission). The non-zero remaining part induces a strictly positive variational gap as described in Eq. (15) of the Appendix.
>
> The estimation target of DDPM++ is the data score, i.e., $s_{\theta}=\nabla\log{p_{t}^{\theta}}+u_{t}^{\theta}\rightarrow \nabla\log{p_{t}^{\phi=id}}$. This data score is fixed throughout the training procedure. On the other hand, according to NELBO given by Eq. (14) of the Appendix, the target of the INDM score network interactively moves by the flow training. The essence is that the flow training and the score training move closer to each other, and they meet at the intermediate point. For that, the NELBO ($=\Vert s_\theta-\nabla\log{p_{t}^{\phi}}\Vert_{2}^{2}=\Vert \nabla\log{p_{t}^{\theta}}+u_{t}^{\theta}-\nabla\log{p_{t}^{\phi}}\Vert_{2}^{2}$) is approximately decomposed into two terms: 1) the difference between the forward score and the reverse score; 2) the remaining residual term. In other words, the first term is related with $\nabla\log{p_{t}^{\phi}}-\nabla\log{p_{t}^{\theta}}$, and the second term is the $u_{t}^{\theta}$-only part. As the forward score ($\nabla\log{p_{t}^{\phi}}$) is trained to meet the reverse score ($\nabla\log{p_{t}^{\phi}}$) at middle, the first term diminishes in INDM faster than DDPM++. Consequently, the second $u_{t}^{\phi}$-term also converges to zero faster (lines 137 - 141 of the Appendix) as the NELBO training could focus more on reducing this second term. This leads the variational gap in Eq. (15) to converge to zero faster in INDM, compared to DDPM++. In fact, in NLL computation, we solve the probability flow ODE given by Eq. (44) of the originally submitted Appendix, and here, when computing the right-hand-side of Eq. (44), observe that $\text{tr}(\nabla s_{\theta})=\text{div}(s_{\theta})=\text{div}(\nabla\log{p_{t}^{\theta}}+u_{t}^{\theta})=\text{div}(\nabla\log{p_{t}^{\theta}})$ because $\text{div}(u_{t}^{\theta})\equiv 0$ for any $\theta$ (by the Helmholtz theorem). This means that *NLL would be invariant of $u_{t}^{\theta}$, and $u_{t}^{\theta}$ only influences to the variational gap; so the fast convergence of $u_{t}^{\theta}$ to zero in INDM over DDPM++ is a clear advantage*. This is the mechanism of nearly MLE-training in INDM derived from the loss design, and one can quickly understand the intuition from the illustration of Figure 11 of the original Appendix. This was abstracted as the term "*bidirectional attraction*" in the main paper due to the space constraint, and we would like to explain this further in the camera-ready version. Please refer to Appendix B.2 for further analysis of INDM.

---

> ### Author Response · Authors · 2022-08-02
> **Continued**
>
> **[Continued Answer to Q1]** (Reasoning #1) Figure 8-(b) of the original manuscript presents that the direction of INDM diffusion path is more linear than DDPM++. Underlying dynamics of this figure is additionally described in Appendix B.4. Figure 13 of the originally submitted Appendix B.4 illustrates the diffusion geometries of DDPM++ and INDM. The path of DDPM++ in the green line moves highly nonlinearly, while the path of INDM in the blue line is relatively linear. This means that a data sample is diffused (forwardly) in a highly nonlinear way, whereas a latent sample is diffused in a linear way, towards the prior distribution, which is depicted as a thin shell (most of the mass of a Gaussian distribution in high dimension is concentrated in a thin shell of square radius to be its dimension). Eq. (17) of the original Appendix theoretically backup Figure 13. According to Eq. (17), once the flow network pushes the data points toward the outside of the prior distribution (lines 221 – 233 of Appendix B.4), the intermediate data is gradually and linearly contracting towards the origin at the expense of random noise (see more linear diffusion path in toy cases with larger-sized initial distribution in Figure 12 for an illustration). Therefore, the question is, which factor of INDM pushes the latent manifold outside of the prior manifold? As described above, the log-determinant term is maximized in the training process, which means that the volume of the latent manifold is the volume of the data manifold times the determinant by the change-of-variable ($Vol(h_{\phi}(A))= Vol(A)\vert det\vert$ for $A\subset\mathbb{R}^{d}$). Therefore, the flow mapping pushes the data point outwardly in $\mathbb{R}^{d}$ to make the latent volume larger. This explains that the log-determinant is key to construct the latent diffusion more linear than the data diffusion. This phenomenon was consistently observed in our experiments.
>
> (Reasoning #2) For a more theoretic-grounded analysis, Theorem 1 of the DSB paper [17] might explain why the sample robustness arises in INDM. Theorem 1 of [17] decomposes the sampling error (measured by the TV norm) with two terms: 1) the first term is proportional to the distance between the diffused ending variable and the prior distribution; 2) the second term is the error originating from the sampling discretization. By using Pinsker's inequality to the empirical observation of $D_{KL}(p_{T}\Vert\pi)\approx 10^{-5}$ in bpd scale [8], the first term is insignificant, and it remains the second term that dominates the entire error. Theorem 1 of [17] derives that the sensitivity of the sampling error by discretization is determined by a constant, which is proportional to $A_{1}^{data}:=\sup{\frac{\Vert\nabla\log{p_{r}(x)}\Vert}{1+\Vert x\Vert}}$. In other words, it indicates that a (trained) diffusion model is robust on the discretization step size if $A_{1}^{data}$ is small. Not only on the data diffusion, Theorem 1 of [17] is directly applicable to the latent diffusion, so $A_{1}^{latent}:=\sup{\frac{\Vert\nabla\log{p_{0}^{\phi}(z)}\Vert}{1+\Vert z\Vert}}$ characterizes the sensitivity of the discretization error in INDM. We add an experiment of these quantities in Figure 7-(c) of the revised manuscript, and it illustrates that $A_{1}^{latent}$ of INDM is 12x times smaller than $A_{1}^{data}$ of DDPM++ (because the latent manifold is smoother than the data manifold), implying that the latent diffusion of INDM is more robust on the discretization step size than the data diffusion of DDPM++. Therefore, Figure 7-(c) combined with Theorem 1 of [17] elucidates the mechanism of the sample robustness of INDM. Please see lines 248 - 262 of the revised main paper and the revised Appendix B.4 for details on this extent.

---

> ### Author Response · Authors · 2022-08-02
> **Thank you for the integrated feedback**
>
> **Q1. [Unclear Effect of Flow]** *The benefit of normalizing flows is unclear. Although the authors try to explain by the means of optimal transport and relative energy, it still lacks an intuitive understanding of why flows+diffusion makes sense, say does the transformed latent space make it easier for diffusion models to learn? One virtue of normalizing flows is to transform complex data distribution into a simple form, then one could learn a model on the simple manifold. However, it is unclear what is learned by flow models in INDM and how flow models increase the efficiency of learning and sampling.*
> \
> \
> **A.**
> (Empirical Effect) The below table shows the ablation study of flow training. We first train the score network, and after the training saturates, we train both the score and flow networks. The score pretraining achieves NLL of 3.03 and FID of 6.70, but the fine-tuning stage of flow(+score) training improves NLL to 2.98 and FID to 6.01. This is the empirical effect of flow training in view of the model performance. This performance gain could potentially originate from the MLE training of INDM, which is induced by the loss design (see **Q2**).
>
> | Model | NLL | FID |
> |--------------------|-------|----|
> | DDPM++ (score training) | 3.03 | 6.70 |
> | + INDM training (score+flow training) | 2.98 | 6.01 |
>
> (Smoothed Latent Manifold) In contrast to the data diffusion that fixes the smoothness of diffused variable, INDM's flow training is potentially beneficial on constructing a smoother latent manifold. Figure 14 of the revised Appendix introduces the approximate first and second derivatives of the log probability. INDM's latent manifold becomes empirically smoother than DDPM's data manifold up to the second-order derivative. Specifically, Figure 14-(a) depicts the score norm, which is the approximation of the score function that represents for the first-order smoothness. Also, each dot of Figure 14-(b) represents $\frac{\Vert\nabla\log{p_{t}(\mathbf{x})-\nabla\log{p_{t}(\mathbf{y})}\Vert}}{\Vert\mathbf{x}-\mathbf{y}\Vert}$ for arbitrary $\mathbf{x},\mathbf{y}\sim p_{t}(\mathbf{x}_{t})$, of which limit is the second derivative of the log probability. Figure 14 empirically demonstrates that the flow training in INDM pushes the unsmooth data distribution into a smoothed latent distribution. For an intuitive visualization, Figure 15 of the revised Appendix illustrates that the singularity of the data manifold disappears in the latent manifold.
>
> This behavior comes from the loss design. When we observe the NELBO in Theorem 1, it includes the flow loss, which is the log-determinant of the Jacobian. By minimizing this NELBO, the log-determinant is maximized. Here, from $\log{p_{r}(x)}=\log{p_{0}^{\phi}(h_{\phi}(x))}+\log{det}$, if the log-determinant is increased, then the initial latent probability at $h_{\phi}(x)$ is decreased because the left-hand-side is fixed by $\phi$. As the initial latent distribution is a probability distribution (sum to 1), this means that the flow transformation makes the latent distribution flattened (and smoothed) when the log-determinant is maximized. In our experiments, we observe that the log-determinant value increases from 0 (=DDPM++) to 3.65 (bpd scale) after the training of INDM on CIFAR-10. Given that, we conjecture the score training in INDM becomes relatively easier than DDPM++ because the estimation target (i.e., the latent/data scores) becomes simpler in INDM, as the reviewer pointed out.
>
> (Reasonings of Sample Robustness) It turns out that the latent diffusion is beneficial for the sampling process throughout our experiments. We provide two reasonings for sample robustness. To summarize, the first is that the latent diffusion trajectory is more linear than the data diffusion trajectory; and the second is that the latent manifold is smoother than the data manifold. As the sample robustness is related to time discretization, we compare the data diffusion in DDPM++ and the latent diffusion in INDM. We add the below arguments in our revised main paper of lines 239 – 262.

---

### Official Review · Reviewer_Aizs · 2022-07-09

**Rating:** 7
**Confidence:** 3
**Soundness:** 4 excellent
**Presentation:** 4 excellent
**Contribution:** 4 excellent

**Summary:**

The paper presents a trainable non-linear diffusion process as a generative models. This work is an expansion of previous successful works that use linear diffusion processes and showed state-of-the-art results on sample generation.
This non-linearity is applied implicitly, helping avoid the alternative of an explicit non-linearity in the drift and diffusion coefficients, which leads to an intractable probability term that causes long training time.
The key idea of the presented method is to transform each data point to a latent variable using a non-linear invertible transformation, and applying a linear diffusion process on that latent variable, where in each point in time one can retrieve the relevant diffused image from a diffused latent variable.
The authors showed that their method also constructs a generative process that is easy to sample from.

**Questions:**

See my comments in the Weaknesses section.


**Limitations:**

The authors have addressed their method's limitations.
The described work has no potential negative societal impact.

**Strengths And Weaknesses:**

Strength:
	- The problem is known to be valid and challenging, and the proposed idea is novel.
	- The method and the idea behind it are well explained in the paper.
	- The authors provide a motivation for non-linear diffusion process by presenting very nice figures of diffused data and its vector field during diffusion, that shows a clear advantage of non-linear diffusion over the linear version.

Weaknesses:
	- The quantitative results are not the best in all , but the gaps between them and the SOTA results are not too big, especially when considering the number of parameters required for the training.
It would be nice to see qualitative results on the CelebA dataset, as the gap in FID seems to be significant.

---

> ### Author Response · Authors · 2022-08-02
> **Thank you for the integrated feedback**
>
> **Q1. [CelebA]** *The quantitative results are not the best in all, but the gaps between them and the SOTA results are not too big, especially when considering the number of parameters required for the training. It would be nice to see qualitative results on the CelebA dataset, as the gap in FID seems to be significant.*
> \
> \
> **A.**
> Thank you for the reviewer's affirmative feedback. Quantitatively, as the reviewer notes, INDM is not the best in CIFAR-10. On the other hand, INDM achieves the SOTA FID in CelebA dataset by improving the previously best FID (1.92) of UNCSN++ to a newly best FID (1.75). This is particularly impressive as the linear counterpart of the SOTA training configuration performs 2.32 in Table 5 of the initial submission, which means that the performance gain purely comes from the nonlinear diffusion modeling.
>
> For the qualitative results on CelebA, Figure 16 of the originally submitted Appendix shows non-cherry picked samples from our SOTA checkpoint.

---

> > ### Comment · Reviewer_Aizs · 2022-08-09
> > **I keep my original rating.**
> >
> > I think that the extension to non-linear diffusion process is necessary and contributing.
> > I agree that the FID improvement in CelebA is impressive enough.
> > I keep my original rating for this paper.

---

> > > ### Author Response · Authors · 2022-08-09
> > > **Thank you for the affirmative evaluation**
> > >
> > > We greatly thank the reviewer for the affirmative review. Thank you so much.

---

### Official Review · Reviewer_befw · 2022-07-22

**Rating:** 6
**Confidence:** 3
**Soundness:** 2 fair
**Presentation:** 2 fair
**Contribution:** 3 good

**Summary:**

The authors propose a variant of latent diffusion models using invertible flows, this enables likelihood-based training. By training a linear diffusion model in the latent space, the corresponding diffusion model in the data space will be nonlinear. Training the flow allows one to learn both drift and diffusion volatility in the forward data-space diffusion.

The authors demonstrate excellent generative performance on standard datasets, as measured by FID scores and show how the method may be adapted for interpolation.

**Questions:**

- It is not clear what figure 7 is showing, shouldn't the end points lie along the same support for each method in data space if the diffusion model is working?
- Does the interpolation loss require taking a gradient through time in the diffusion simulation? If so what is the memory requirement, does this break for larger images?
- See questions on SB in weaknesses above
- Is the extensive pretraining regime required?
- What is the total train time including pretraining vs other baselines (without pretraining)?
- The latent is the same dimension as the data, is there any computational savings?

**Limitations:**

The authors briefly comment on training time in the final section, however, do not detail the extensive pretraining regime required.

It would benefit the paper to address some issues in the discussion regarding OT, and move this to the appendices, then instead spend more time discussing the training procedure, provide an algorithm and discuss the importance of the pretraining regime which will be of significant importance to practitioners.

**Strengths And Weaknesses:**


Strengths:
- The authors propose an elegant method to train latent diffusion models using flows as encoder/ decoder, this permits efficient training of nonlinear diffusions via linear diffusion in the latent space and gives an explicit nonlinear diffusion in the data space
- The method exhibits excellent generative performance measured by FID scores

Weaknesses:
- The paper would benefit from an algorithm detailing the exact training procedure
- In the appendices, line 599, it is stated that the diffusion model is trained without a flow (for 5 days), then is fine-tuned with the flow. This is a very important detail and should be included in the main text. Does the proposed method not work without pretraining? The authors should include the performance for the diffusion without the flow first to indicate what the benefit is. Similarly, given 5 days is a significant amount of time would this not mean the training cost is a lot more than regular diffusions? Hence Table 1 may be misleading.
- OT section is not clear and seems misleading:
    - It appears there is no theoretical justification for being close to OT, am I right in thinking this is purely an empirical observation?
    - Line 275 references Appendices C.4, however, there is no Appendix C.4 in the supplementary material
    - Line 264, is this the Wasserstein 2 distance being compared? How is it computed? Hence is the ground cost the squared Euclidean distance? It is important to note that DDPM uses a discretized Ornstein Uhlenbeck process (VPSDE) which will result in an approximate OT distance with different ground cost to the squared Euclidean, one would need to use a Brownian motion (VESDE) for fair comparison, see e.g. section 3.1 of [1]. Hence the OT trajectory for different ground costs may not be a straight line, see e.g. [2], hence the diagrams and discussion may be misleading e.g. line 247.
    - FBSDE, as advocated in [18]'s code, trains a forward and backward network to completion and then fine tunes with a small number of steps. If not training each network to completion, the final networks will not result in an SB approximation as it will not be following the IPF. In order to get a reasonable OT computation, one needs to train the network to completion at each IPF iteration.
    - Perhaps comparing OT with closed form solutions such as between 2 Gaussian measures might be easier to justify, see e.g. Figure 2 of [1] (arxiv version)
- Figure 4 uses a checkpoint from FBSDE [18], is this where the first iteration is using score-matching with a long time horizon and the forward diffusion in the first IPF iteration, then fine-tuned in subsequent IPF iterations? If so then visually looking at the diffusion time for non linearity may be misleading if most of the diffusion is close to Gaussian, see e.g. [2] the double well experiment for non-linear trajectories


- Minor
    - Line 179, "Few works [18] in SBP detour the issue on the slow training by constructing an experience replay memory ", this technique for Diffusion SB was first introduced in [1], see page 53 Technique 3, and Algorithm 3. This paper was around 5 months before the reference cited.
- Line 229, again [1] describes continuous time SB in section 3.5, and implements a discretized version


Summary:

The paper proposes some nice ideas, and the method demonstrates excellent performance. The idea to use flows to convert a diffusion to latent is a good contribution.

Unfortunately the authors focus too much on connections to OT (which appears theoretically ungrounded, not explained well, with some apparent misunderstandings) and not enough on explaining the actual method, algorithm, and training procedure, omitting key details from the main text such as the pretraining regime.


[1] Diffusion Schrödinger Bridge with Applications to Score-Based Generative Modeling
Valentin De Bortoli, James Thornton, Jeremy Heng, Arnaud Doucet, https://arxiv.org/abs/2106.01357
[2] Solving Schrödinger Bridges via Maximum Likelihood, Vargas, https://arxiv.org/abs/2106.02081

---

> ### Author Response · Authors · 2022-08-02
> **Continued**
>
> **Q13. [Pretraining Needed?]** *Is the extensive pretraining regime required?*
> \
> \
> **A.**
> The table below gives the ablation study's result on the number of pretraining steps. We observe that NLL is identical with/without the pretraining (at **Q2**). Thus, we investigate the sample performance by the pretraining steps in this answer. There are 5 variants of experimental settings. First, we pretrain DDPM++ with 100/200/300/400/500k steps. Afterward, we fine-tune INDM for 150k more steps on each of pretrained checkpoint, and we report the final FIDs of INDM. According to our ablation study, the pretraining phase is critical to the sample performance, but we respectively emphasize that the experiments of 200k and 500k pretraining steps differ in FID by less than 1.
>
> | Pretraining Steps | 100k | 200k | 300k | 400k | 500k |
> |--------------------|-----|-------|-------|-------|----|
> | +150k INDM Training | 8.77 | 7.56 | 7.31 | 6.92 | 6.67 |
>
>
> \
> \
> \
> **Q14. [Training Time]** *What is the total train time including pretraining vs other baselines (without pretraining)?*
> \
> \
> **A.**
> We add the below table in Appendix E.7 of the revised version. The below table compares INDM with baselines with respect to a single GPU-time for the total training time on CIFAR-10. The remaining columns including training steps, GPU Spec, NLL, and FID are reported for reference. For DiffFlow, we present the reported GPU days in the paper. For LSGM and SBP, we estimate the elapsed time with the released training configuration in their papers and GitHub repositories. For DDPM++ and INDM, we report the elapsed time from our own experiments. From the below table, the overall training time of INDM/DiffFlow/LSGM remains at a similar scale. SBP is the fastest algorithm because of the experience replay memory technique. Note that a completely fair comparison between algorithms is infeasible because the training setup (e.g. #GPUs, training steps, network size …) varies by algorithms. Also, P40 is strictly slower than RTX series GPUs.
>
> | Model | Total Training Time (GPU Days) | Training Steps | GPU Spec | #GPUs | NLL | FID |
> |--------------------|-----|-------|-------|-------|----|----|
> | DDPM++ [1] | 5 | 500k | P40 | 1 | 3.03 | 6.70 |
> | LSGM [14] | 44 | 450k | RTX 3090 | 8 | 2.87 | 6.89 |
> | SBP [18] | 3 | 260k | RTX 3090 | - | 2.98 | 3.18 |
> | DiffFlow [15] | 32 | 100k | RTX 2080 | 8 | 3.04 | 14.14 |
> | INDM (including pretraining time) | 25 | 700k | P40 | 4 | 2.98 | 6.01 |
> | INDM (w/o pretraining) | 60 | 600k | P40 | 4 | 2.98 | 8.49 |
>
>
> \
> \
> \
> **Q15. [Computation By Dimension]** *The latent is the same dimension as the data, is there any computational savings?*
> \
> \
> **A.**
> The below table compares the latent dimension of LSGM and INDM. We compute the latent dimension of LSGM, according to their paper and released checkpoint. Contrary to the dimensional reduction property, VAE in LSGM maps data into a latent space of a much higher dimension than the data dimension. LSGM is known to perform well, but having observed 15x higher latent dimension than the data dimension on CIFAR-10, the good performance was not gained for free. On the other hand, INDM always retains the same dimension to the data, while keeping the invertibility.
>
> | Dataset | Data Dimension | Latent Dimension of INDM | Latent Dimension of LSGM [14] |
> |--------------------|-----|-----|-----|
> | MNIST | 784 | 784 | 2,560 |
> | (Binarized) OMNIGLOT | 11,025 | 11,025 | 15,360 |
> | CIFAR-10 | 3,072 | 3,072 | 46,080 |
> | CelebA-HQ 256 | 196,608 | 196,608 | 819,200 |
>
>
> \
> \
> \
> **Q16. [Paper Revision]** *It would benefit the paper to address some issues in the discussion regarding OT, and move this to the appendices, then instead spend more time discussing the training procedure, provide an algorithm and discuss the importance of the pretraining regime which will be of significant importance to practitioners.*
> \
> \
> **A.**
> We greatly thank for the helpful review. We revise the paper as the reviewer suggested. We move the discussion regarding OT to the Appendix, and instead, we clarify our training details, especially focusing on the training procedure and experimental details. Please see the revised paper.

---

> ### Author Response · Authors · 2022-08-02
> **Continued**
>
> **Q8. [FBSDE]** *Figure 4 uses a checkpoint from FBSDE [18], is this where the first iteration is using score-matching with a long time horizon and the forward diffusion in the first IPF iteration, then fine-tuned in subsequent IPF iterations? If so then visually looking at the diffusion time for non linearity may be misleading if most of the diffusion is close to Gaussian, see e.g. [2] the double well experiment for non-linear trajectories.*
> \
> \
> **A.**
> The reviewer is correct on the training procedure of FBSDE. Though there is a potential risk of the non-convergence to SBP of the FBSDE algorithm, under the situation that no other paper has released any trained checkpoint on a benchmark real-world dataset for SBP, using FBSDE is the best option to elucidate the behavior of SBP.
>
> Following the double well experiment [2], we acknowledge that SBP with the reference measure of a Brownian motion and a potential energy-driven reference measure could behave differently in their nonlinearity. However, as the destined variable of VPSDE is much closer to the prior than VESDE (see Figure 5 of FBSDE paper), the reference measure of VPSDE is likely to be closer to $\mathcal{P}(p_{r},\pi)$ in VPSDE than VESDE. As the solution of SBP is the closest path measure on $\mathcal{P}(p_{r},\pi)$ to the reference measure, SBP with VPSDE is expected to attain a more linear forward SDE than VESDE. Given that, we believe that Figure 4 is enough to think that SBP is semi-linear under the family of linear reference measures. Please let us know if any checkpoint trained with VPSDE is released in the family of SBP papers. We will measure the nonlinearity of VPSDE, if any exists.
>
> \
> \
> \
> **Q9. [Citation Correction]** *Line 179, "Few works [18] in SBP detour the issue on the slow training by constructing an experience replay memory ", this technique for Diffusion SB was first introduced in [1], see page 53 Technique 3, and Algorithm 3. This paper was around 5 months before the reference cited.*
> \
> \
> **A.**
> We have revised the citation as the reviewer points out.
>
> \
> \
> \
> **Q10. [Clarification of SBP]** *Line 229, again [1] describes continuous time SB in section 3.5, and implements a discretized version.*
> \
> \
> **A.**
> We acknowledge that SBP is designed and analyzed in continuous time with a discretized implementation. In Table 1 of the revised paper, we clarify Table 1 column name to be "Implemented Data Diffusion" instead of "Data Diffusion".
>
> \
> \
> \
> **Q11. [Figure 7]** *It is not clear what figure 7 is showing, shouldn't the end points lie along the same support for each method in data space if the diffusion model is working?*
> \
> \
> **A.**
> Not necessarily the end points should lie along the same support. The latent diffusion ends at a Gaussian prior in INDM, but the data diffusion in INDM does not have to end at this prior. Rather, the flow network transforms the latent ending variable into the data ending variable, so it depends on how normalizing flow is trained. We have no knowledge of this at the current state, but it turns out that the data ending variable is visually a pure noise in all benchmark real data experiments as depicted in Figure 3 of the original submission.
>
> \
> \
> \
> **Q12. [Memory Usage]** *Does the interpolation loss require taking a gradient through time in the diffusion simulation? If so what is the memory requirement, does this break for larger images?*
> \
> \
> **A.**
> We suspect that the reviewer is asking if a network (either flow-or-score) evaluation is required in the forward step of the diffusion process in INDM, particularly in calculating the interpolation loss. The answer is negative. The interpolation loss simply optimizes $E_{p_{r}^{(2)}(\mathbf{y})}[-\log{p_{\phi}(\mathbf{y})}]$, where $\log{p_{\phi}(\mathbf{y})}$ is computed from the a single feed-forward flow evaluation. Thus, the interpolation loss does not require a forward diffusion, and it only needs one time of flow evaluation. Therefore, the interpolation loss takes 0.2GB for its memory usage, out of 2.5GB of the total memory. In higher dimensional cases, the memory proportion (=0.2/2.5) of the interpolation loss out of total memory will not significantly increase. For clarity, we add the line-by-line algorithm of the interpolation task in Algorithm 2 of the revised Appendix.

---

> > ### Comment · Reviewer_befw · 2022-08-08
> > **Clarity on interpolation loss**
> >
> >  **For clarity, we add the line-by-line algorithm of the interpolation task in Algorithm 2 of the revised Appendix**
> >
> > Can you please be explicit exactly how this is calculated, currently in line 3 of algo 2 of the Appendix, it just say **Compute Lint = Ep[− log pφ(y)] for y ∼ p**
> >
> > How is $pφ$ evaluated? And what is this exactly? Earlier in the paper $p\thetaφ$ is defined as the density of the generated data. Typically in diffusion models, the likelihood is computed via solving an ODE flow to a Gaussian?

---

> > > ### Author Response · Authors · 2022-08-09
> > > **Clarification on interpolation loss**
> > >
> > > $p_{\phi}(y)$ is obtained by feed-forwarding the flow network, once. Concretely, it is computed by $\log{p_{\phi}(y)}=\log{p(h_{\phi}(z))}+\log{det}$, where $p(h_{\phi}(z))$ is the probability of prior distribution at $h_{\phi}(z)$. There is no additional diffusion-related computational burden for the interpolation task. We would revise our manuscript to note how we compute each probability clearly. The essence of the interpolation task is to transform the initial latent distribution into the first data distribution; and the fully diffused latent distribution into the second data distribution.

---

> > > > ### Comment · Reviewer_befw · 2022-08-09
> > > > **Understand**
> > > >
> > > > Sorry for not understanding at first but I assumed this setup only has access to samples of the dataset.
> > > >
> > > > I think I understand what is happening. In this case the flow is just chosen/ trained so that the push forward from Gaussian is the second dataset?

---

> > > > > ### Author Response · Authors · 2022-08-09
> > > > > **Thank you for the prompt feedback**
> > > > >
> > > > > The reviewer is correctly understanding the interpolation task. The flow is trained so that the push forward from a Gaussian (or prior of latent diffusion) is the second data distribution. Also, here, the push forward from the initial latent distribution is the first data distribution. In this sense, we call this a diffusion bridge that interpolates two datasets.

---

> ### Author Response · Authors · 2022-08-02
> **Continued**
>
> **[Continued Answer to Q5]** A possible approach to analyze such sampling robustness is using Theorem 1 of DSB paper [17]. Theorem 1 of [17] decomposes the sampling error (measured by the TV norm) with two terms: 1) the first term is proportional to the distance between the diffused ending variable and the prior distribution; 2) the second term is the error originating from the sampling discretization. By using Pinsker's inequality to the empirical observation of $D_{KL}(p_{T}\Vert\pi)\approx 10^{-5}$ in bpd scale [8], the first term is insignificant, and it remains the second term that dominates the entire error. Theorem 1 of [17] derives that the sensitivity of the sampling error by discretization is determined by a constant, which is proportional to $A_{1}^{data}:=\sup{\frac{\Vert\nabla\log{p_{r}(x)}\Vert_{2}}{1+\Vert x\Vert_{2}}}$. In other words, it indicates that a (trained) diffusion model is robust on the discretization step size if $A_{1}^{data}$ is small. Not only on the data diffusion, Theorem 1 of [17] is directly applicable to the latent diffusion, so $A_{1}^{latent}:=\sup{\frac{\Vert\nabla\log{p_{0}^{\phi}(z)}\Vert_{2}}{1+\Vert z\Vert_{2}}}$ characterizes the sensitivity of the discretization error in INDM. We add an experiment of these quantities in Figure 7-(c) of the revised manuscript, and it illustrates that $A_{1}^{latent}$ of INDM is 12x times smaller than $A_{1}^{data}$ of DDPM++, implying that the latent diffusion of INDM is more robust on the discretization step size than the data diffusion of DDPM++. Therefore, Figure 7-(c) combined with Theorem 1 of [17] partially elucidates the mechanism of the sample robustness of INDM. Please see lines 248 - 262 of the revised main paper and the revised Appendix B.4 for details on this extent.
> \
> \
> [Chen2014On] Chen, Yongxin, Tryphon T. Georgiou, and Michele Pavon. "On the relation between optimal transport and Schrödinger bridges: A stochastic control viewpoint." Journal of Optimization Theory and Applications 169.2 (2016): 671-691.
>
> [Khrulkov22Understanding] Khrulkov, Valentin, and Ivan Oseledets. "Understanding ddpm latent codes through optimal transport." arXiv preprint arXiv:2202.07477 (2022).
>
> \
> \
> \
> **Q6. [FBSDE]** *FBSDE, as advocated in [18]'s code, trains a forward and backward network to completion and then fine tunes with a small number of steps. If not training each network to completion, the final networks will not result in an SB approximation as it will not be following the IPF. In order to get a reasonable OT computation, one needs to train the network to completion at each IPF iteration.*
> \
> \
> **A.**
> The reviewer is correct on the training procedure of FBSDE [18]: it trains the score matching at the initial stage for a long-time, and then fine-tunes each of half-bridge problem with either initial-value or final-value constraints, iteratively. As the reviewer points out, each iterative fine-tuning stage might end with immature optimization, and this nature could bring a possibility of the non-convergence of FBSDE to the SBP. We believe that DSB [17] could better converge to the solution of SBP, but unfortunately, DSB paper does not release their trained checkpoints. Therefore, we have experimented based on the released checkpoint of FBSDE paper at the expense of insufficient IPF iteration for each stage. In the revision, we exclude the result of FBSDE in Table 2 because FBSDE uses VESDE, while DDPM++ and INDM in Table 2 use VPSDE.
>
> \
> \
> \
> **Q7. [Closed-form Toy Experiment]** *Perhaps comparing OT with closed form solutions such as between 2 Gaussian measures might be easier to justify, see e.g. Figure 2 of [1] (arxiv version).*
> \
> \
> **A.**
> As the reviewer pointed out, we have experimented on a 2d toy dataset, the two moons dataset. We have computed the optimal Monge map using the Python Optimal Transport (POT) library. Figure 13 of the revised Appendix compares the latent diffusion trajectory and the optimal transport trajectory at the beginning and end of the training. We use the probability flow ODE to draw the deterministic diffusion trajectory. Figure 13 describes that the flow training pushes the latent manifold outwardly as training proceeds, and this push-forwarded latent manifold diffuses to the prior manifold quite linearly because VPSDE has the drift term going inwardly, see Eq. (16). Therefore, we conclude that 1) VPSDE is nearly the optimal trajectory from [Khrulkov22Understanding], 2) the time-continuous trajectory itself becomes linearized in the toy 2d case. Combining these properties, a latent diffusion could potentially reduce the relative energy, which leads the diffusion trajectory closer to the optimal transport.

---

> ### Author Response · Authors · 2022-08-02
> **Continued**
>
> **Q5. [OT]** *Line 264, is this the Wasserstein 2 distance being compared? How is it computed? Hence is the ground cost the squared Euclidean distance? It is important to note that DDPM uses a discretized Ornstein Uhlenbeck process (VPSDE) which will result in an approximate OT distance with different ground cost to the squared Euclidean, one would need to use a Brownian motion (VESDE) for fair comparison, see e.g. section 3.1 of [1]. Hence the OT trajectory for different ground costs may not be a straight line, see e.g. [2], hence the diagrams and discussion may be misleading e.g. line 247.*
> \
> \
> **A.**
> The default cost function is the squared Euclidean distance. We agree with the reviewer that the connection of OT and the straight line is misleading. Our original intention was to explain the sampling robustness with respect to the straightness of the diffusion trajectory. In concise language, Figure 8-(a) of the original manuscript illustrates that INDM is more robust on the ODE tolerance, and Figure 8-(b) partially explains the mechanism of this robustness in view of the cosine similarity of the diffusion trajectory. Please refer to Appendix B.4 of the original submission for the detailed analysis of the straightness of diffusion trajectory. We think it is much better to omit the argument of OT in Section 6.3 for a clear understanding of the sample robustness, so we have removed OT part in our revised paper, and separated the "relative energy" subsection into Section 6.4.
>
> As the reviewer pointed out, a Brownian motion (VESDE) could be approximately considered an entropy-regularized OT with the squared Euclidean cost function (Eq. 54-56 of [Chen2014On]). Also, we acknowledge that the fluid dynamic formulation, i.e., Benamou-Brenier style formulation, of the Ornstein Uhlenbeck process (VPSDE) is not an entropy-regularized OT problem (Eq. 59 of [Chen2014On]). However, we emphasize that a recent paper [Khrulkov22Understanding] on VPSDE claims that the encoder map of VPSDE is nearly the optimal transport with the squared Euclidean cost function. Here, the encoder map of VPSDE is the mapping from the initial point to the final point passed through the probability flow ODE of VPSDE. Specifically, [Khrulkov22Understanding] proves that the encoder map is exactly the optimal transport if the initial distribution follows a Gaussian distribution; and [Khrulkov22Understanding] empirically shows that the encoder map of real distribution is nearly the optimal transport. Two papers seem to contradict each other, but it is not because [Chen2014On] proves that the solution of minimizing the reverse KL divergence under the VPSDE reference measure does not equivalent to the Benamou-Brenier formulation anymore, whereas [Khrulkov22Understanding] shows that the *encoder map* of VPSDE is itself a nearly optimal transport without specifying any loss function. Our starting point of the discussion in Section 6.3 is that even though VPSDE is nearly an optimal transport, the continuous-time transport plan of VPSDE (DDPM++) is too curvy for a robust sampling, which is in fact naturally anticipated by the continuous-time analysis of [Chen2014On]. Section 6.3 (and Appendix B.4 of the original submission) was devoted to empirically show that the transport plan of INDM is less curvy than DDPM++. Since this argument is nothing to do with optimal transport plan, we exclude the argument of OT in Section 6.3. About a short summary of Appendix B.4, please see (Reasoning #1)-paragraph of the answer for **Q1** of reviewer #4.

---

> ### Author Response · Authors · 2022-08-02
> **Thank you for the integrated feedback**
>
> **Q1. [Algorithm Pseudocode]** *The paper would benefit from an algorithm detailing the exact training procedure.*
> \
> \
> **A.**
> Thank you for the sincere review. As the reviewer pointed out, we spend more space discussing the actual method, algorithm, and training procedure. We put the training procedure in Algorithm 1 of the revised manuscript. Also, we add Algorithm 2 of the dataset interpolation task in the Appendix.
>
> \
> \
> \
> **Q2. [Pretraining]** *In the appendices, line 599, it is stated that the diffusion model is trained without a flow (for 5 days), then is fine-tuned with the flow. This is a very important detail and should be included in the main text. Does the proposed method not work without pretraining? The authors should include the performance for the diffusion without the flow first to indicate what the benefit is. Similarly, given 5 days is a significant amount of time would this not mean the training cost is a lot more than regular diffusions? Hence Table 1 may be misleading.*
> \
> \
> **A.**
> As the reviewer pointed out, pretraining is a very important detail that should be included in the main text. We add Table 4 (or just see the below table) and subsequent details of the pretraining in lines 301 - 311 in the revised paper. The below table shows that NLL is identical with or without pretaining, but FID is slightly worsened if we train INDM from the scratch. Also, the DDPM++/NCSN++ performances in the below table and Table 3 of the original/revised main paper already indicate *the performance for the diffusion without the flow* on CIFAR-10. For instance, in the below table, DDPM++ performs with NLL of 3.03, and this is the performance for the diffusion model with identity flow. Starting from this pretrained checkpoint, further INDM training of score+flow networks successfully improves NLL from 3.03 to 2.98 and FID from 6.70 to 6.01. Note that the performance gain of INDM purely originates from our design. The performance of DDPM++ does not improve after reaching 3.03 of NLL, but at the fine-tuning stage, NLL immediately begins to decrease down to 2.98. We would additionally discuss ablation study of the pretraining iterations before the end of the rebuttal period.
>
> | Model | NLL | FID |
> |--------------------|-------|----|
> | DDPM++ | 3.03 | 6.70 |
> | Reported INDM With Pretraining | 2.98 | 6.01 |
> | INDM Without Pretraining | 2.98 | 8.49 |
>
> We agree that 5 days of additional training is a significant amount of time. In the revised version, we change the column name from "Training Cost" to "Training Complexity" in order to correctly clarify our original intention that INDM reduces the time complexity from $O(N)$ (SBP and DiffFlow) to $O(1)$, where $N$ is the number of discretizations (Here, DSB [17] or SB-FBSDE [18] detour this complexity issue by using the periodically refreshed experience replay memory, but this does not come from the structural design). Please refer to the answer of **Q14** for the total training time.
>
> The below table measures the training time per training step for various numbers of discretizations $N$ on CIFAR-10. In contrast to INDM which is invariant on the choice of $N$, SBP and DiffFlow are not scalable by $N$. The training time is measured under the identical computing resource (1x NVIDIA RTX 3090/Intel I7 3.8GHz) and the same batch size (32) to compare INDM with baselines in a fair setting. We add this table and associated argument in Appendix E.7 of the revision.
>
> | Model | $N=100$ | $N=1000$ | $N=\infty$ |
> |--------------------|-------|-------|-------|
> | DDPM [1] | 0.27 | 0.27 | 0.27 |
> | SBP (w/o experience replay) [18] | 2.83 | 23.3 | $\infty$ |
> | SBP (w/ experience replay, including refreshing time) [18] | 0.52 | 2.39 | $\infty$ |
> | DiffFlow [15] | 18.45 | 180.88 | $\infty$ |
> | INDM | 1.69 | 1.69 | 1.69 |
>
> \
> \
> \
> **Q3. [Theoretic Justification]** *It appears there is no theoretical justification for being close to OT, am I right in thinking this is purely an empirical observation?*
> \
> \
> **A.**
> The reviewer is correct that there is no theoretical justification in the submitted version. We agree with the review's comment on the OT section, and we move the argument of the optimal transport to the Appendix as the reviewer recommended. Please refer to **Q5** for a detailed explanation on this.
>
> \
> \
> \
> **Q4. [Supplementary]** *Line 275 references Appendices C.4, however, there is no Appendix C.4 in the supplementary material.*
> \
> \
> **A.**
> We are sorry for the supplementary issue. From the Fokker-Planck equation, a corresponding PDE of the SDE of $\mathrm{d}x_t=f(x_t,t)\mathrm{d}t+g(t)\mathrm{d}w_t$ is $\frac{\partial}{\partial t}p_{t}=-\text{div}\big((f-\frac{1}{2}g^{2}\nabla\log{p_{t}})p_{t}\big)$ (line 498 of the original submitted Appendix). This leads that the diffusion bridge *between $p_{0}^{\phi}$ and $p_{T}^{\phi}$* satisfies the continuity equation with $v_{t}=f-\frac{1}{2}g^{2}\nabla \log{p_{t}}$. Please see the revised Appendix B.5.

---

> > ### Comment · Reviewer_befw · 2022-08-08
> > **Thank you**
> >
> > Thank you for the clarifications.
> >
> > The paper is a lot clearer with additional clarity on the pretraining, and moving the unnecessary content regarding connection to OT to the appendices. But I would need more time to read and evaluate this properly.
> >
> > If the FID performance is better using a standard DDPM without the flow than DDPM + flow (without pretraining), then isn't the main benefit of the proposed method essentially being fine-tuning approach to DDPM? **This should be made clear**. Finetuning is a valid contribution as one could take any pretrained diffusion and improve it. However it is very hard to tell from the paper that this is the contribution.
> >
> > Rather than in LSGMs where the encoder is pretrained followed by a diffusion, the proposed method is pretraining an existing diffusion then fine tuning with a flow. Although the LSGM dimension is higher in total, which I did not know, on further inspection the spatial dimension is smaller, which is the key to efficiency (most diffusion models increase the number of channels significantly in the Unet anyway).
> >
> > After further review, other than fine-tuning, it is still not clear from the paper what the benefit of the proposed approach is over LSGM. It has an explicit form of nonlinear diffusion which is elegant, but I still do not understand the benefit of this. The authors mentioned in response to Reviewer KAbZ that the benefit is in the interpolation, then this should be expanded on in the paper. Indeed Figure 8 even suggests that the interpolation is not very meaningful if the different colours are completely mixed at T=1 (in the data space).
> >
> > Therefore, although improved, I believe this paper still needs more work in terms of clarity of contribution and benefits, and would need another round of reviews.

---

> > > ### Author Response · Authors · 2022-08-09
> > > **Thank you for the review**
> > >
> > > We kindly request the reviewer to consider the number of parameters used. Table 5 of the revised paper clarifies that LSGM-269M with the NLL setting reports $(NLL,FID)=(2.97,6.15)$, whereas INDM-121M reports $(NLL,FID)=(2.97,4.79)$. Also, LSGM-109M with the balanced setting reports $(NLL,FID)=(2.96,4.60)$, but INDM-75M reports $(NLL,FID)=(3.01,3.25)$. These results under a comparable number of parameters (Indeed, LSGM has larger #params than INDM in both cases) strongly support the quantitative efficacy of INDM, compared to LSGM. We believe this parameter-efficient result is significant.
> > >
> > > Moreover, we respectively argue about the training stability. In any of training configurations (with/without pretraining, choice of diffusion weighting functions, etc), the training was stable in INDM. However, as repeatedly reported [13,30], LSGM is under the training instability in FID setting ($\lambda=\sigma^{2}$). This is why a small-sized LSGM (best FID) is not reported [30], and this might block lightening the size of LSGM. We think this training stability is a major contribution of INDM over LSGM.
> > >
> > > Furthermore, we want to emphasize that INDM is consistently faster than LSGM in its sampling speed. We compare this in Figure 7-(a), and we analyze why the sampling in INDM is robust in Appendix B.3 and B.4. This analysis is the first systematic analysis of the sampling procedure with respect to the diffusion manifold and its geometry. With the lack of such analysis in LSGM, we think this is another merit of INDM over LSGM.
> > >
> > > Lastly, we would like to clarify on pretraining. We agree with the reviewer that developing a fine-tuning approach of DDPM is clearly a good research direction that is worth to be investigated further. However, we point that the baseline models including LSGM and SBP also use pretraining in their model training. With this use of pretraining, however, the baseline models are evaluated by their own modeling methods as well as their pretraining techniques.
> > >
> > > We request the reviewer to reconsider our main contribution that non-linearizes the forward diffusion process in a strict view (see **Q9** of Reviewer 19AU to understand what a strict view is). Under this framework, pretraining can be interpreted merely as a technique that searches for the optimal diffusion nonlinearity near a linear diffusion. We feel it is a bit unfair to evaluate this paper only from the perspective of pretraining.
> > >
> > > Also, LSGM is currently performing only on a larger dimensional latent (about 15x larger in CIFAR-10) and no work has successfully trained a latent diffusion on a smaller-or-equivalent dimension. In contrast, our INDM shows the efficacy of latent diffusions on an equivalent-dimensional space to the data dimension.
> > >
> > > Overall, we believe INDM could improve the community of diffusion models for its clear empirical strength, training stability, and abundant analysis. We respect the reviewer’s comment, and we would faithfully focus more on the comparison of INDM with LSGM and the specification of the pretraining method in our camera-ready version if accepted.

---

> > > > ### Comment · Reviewer_befw · 2022-08-09
> > > > **Clarifications**
> > > >
> > > > Thank you for the clarifications.
> > > >
> > > >  Whilst the response is clear , this discussion is missing from the main paper. I do not see any discussion regarding latent dimension comparisons to lsgm or comparison on training stability.
> > > >
> > > > I do see the table on number of parameters and this discussion.
> > > >
> > > > I will raise my score to reflect this but I strongly encourage the authors to make it clear in the paper what the benefits and contribution are over lsgm.
> > > >
> > > > Having a closed form non linear diffusion is nice on paper and in principle but it needs to be clear what the benefits are.
> > > >
> > > > The interpolation does not seem like a strong justification as  figure 8 suggests the interpolation in this method is not very meaningful unlike SB solving the OT problem.

---

> > > > > ### Author Response · Authors · 2022-08-09
> > > > > **Thank you for the affirmative evaluation**
> > > > >
> > > > > Thank you for raising the score. As the reviewer points out, in the final revision, we would include the following points: 1) a discussion regarding latent dimension comparisons to LSGM; 2) a discussion regarding the training stability; 3) a discussion on the effect of pretraining; 4) a discussion on training specifications including the image generation task and the interpolation task. We greatly thank the reviewer for such a helpful and healthy review. We think an endeavor with respect to the interpolation task would be an excellent research topic in the future.

---

### Official Review · Reviewer_KAbZ · 2022-07-23

**Rating:** 6
**Confidence:** 2
**Soundness:** 3 good
**Presentation:** 3 good
**Contribution:** 2 fair

**Summary:**

The authors expend the linear diffusions of SDEs to a data-adaptive trainable nonlinear diffusion. They propose a new model, implicit Nonlinear Diffusion Models (INDM) to train its forward diffusion SDE. The proposed model constructs nonlinearity of data diffusion by transforming a linear diffusion process on the latent space back to the data space, where the transformation is implemented with a normalizing flow.

**Questions:**

Are there other generative models that integrate normalizing flow to be considered as baselines in the experiments?

**Limitations:**

1. The motivations of integrating normalizing flow into the diffusion model is not convinced. Can we integrate other component rather than normalizing flow into the diffusion model?
2. The motivations of why invertibility is important is not clear, although the authors provide a figure and tried to illustrate this.
3. Normalizing flow has been widely used in generative model such as normalizing flow-based VAE. Thus, the idea of integrating normalizing flow into diffusion model is not novel, as diffusion model is categorized into generative model.


**Strengths And Weaknesses:**

1. Nonlinear diffusion is taken into account in diffusion models.
2. The proposed diffusion model is able to learn forward diffusion process out of the variational family of inference path measures, and the corresponding optimization is able to provide MLE training and a good diffusion bridge.
3. Experiments are carefully conducted and analyzed.

---

> ### Author Response · Authors · 2022-08-02
> **Continued**
>
> **Q4. [Other Baselines]** *Are there other generative models that integrate normalizing flow to be considered as baselines in the experiments?*
> \
> \
> **A.**
> To the best of our knowledge, DiffFlow is the only model that integrates normalizing flows with diffusion models, so far. DiffFlow models the drift term, $\mathbf{f}_{\phi}$, with a flow network, parametrized by $\phi$. There are several functional advantages of INDM over DiffFlow. First, unlike INDM which models both drift and diffusion terms nonlinearly, DiffFlow only models the nonlinear drift term, and it omits to nonlinearize the diffusion term (lines 149 - 150 of the original submission). The diffusion term of DiffFlow, $g$, *cannot* be nonlinearized with the suggested modeling approach of DiffFlow (lines 397-419 of Appendix C.2).
>
> Second, the variance of DiffFlow loss is strictly larger than the variance of INDM (Table 8 and lines 430-442 of Appendix C.2). This higher variance of DiffFlow could potentially lead to a suboptimal training: DiffFlow performs 14.14 in FID, whereas INDM performs 2.28 in FID (Table 4). Third, as described in **Q3**, the training of DiffFlow is slower compared to INDM by far.

---

> ### Author Response · Authors · 2022-08-02
> **Continued**
>
> **Q3. [Contribution Claim]** *Normalizing flow has been widely used in generative model such as normalizing flow-based VAE. Thus, the idea of integrating normalizing flow into diffusion model is not novel, as diffusion model is categorized into generative model.*
> \
> \
> **A.**
> As the reviewer pointed out, combining different kinds of generative models might be an old idea. We agree that works with a straightforward and incremental extension should not be valued in the machine learning community, but we kindly ask the reviewer to reconsider our contribution. Each subfield of generative models has its own difficulties and strengths, and the recently introduced diffusion model is worth investigating further as there is only a few systematic analysis of diffusion models, particularly on the diffusion strategy. Having that a diffusion model is a generalization of Hierarchical VAE with an infinite number of latent variables, the major difference between a diffusion model and VAE lies in their inference procedures. In contrast to VAE which trains inference through an encoder network, a diffusion model fixes its inference part because the diffusion strategy of the forward process is fixed throughout the training. We believe that enabling inference training in diffusion models is a significant and nontrivial direction worth investigating. This is explained in lines 25 - 29 of the originally submitted main paper.
>
> In the paper, we find that *all* existing works on this extent (such as DiffFlow or SBP) require $O(N)$ complexity where $N$ is the number of discretizations, compared to the linear diffusion model with $O(1)$ complexity. Our INDM design naturally remains the complexity to be $O(1)$. Structurally, INDM could be viewed as merely an integration of two generative models. However, we believe this simplicity should be counted as an advantage, rather than a disadvantage because this simple structural change is the source of all improvements to our model. We think this is a nontrivial improvement, and we revise Table 1 to clarify this property; please see the revised version.
>
> The below table summarizes our argument. We measure the wall-clock time of the training for each algorithm on CIFAR-10 with a shared batch size (32) and a shared $N=1,000$. Here, our INDM is originally suggested as a continuous time diffusion model, but we compare the elapsed time of discretized INDM below to clarify the advantage of INDM in a fair setting. We use the reported hyperparameters (such as network structure) other than the batch size and the discretization steps ($N$). We use an identical computing resource of 1x NVIDIA RTX 3090 and Intel I7 3.8GHz.
>
> Below, DiffFlow seems extremely slow if we take $N=1,000$, so DiffFlow selects less number of latent variables ($N\sim O(10)$) for their training, which could be a potential reason for their suboptimal performance of 14.14 in FID. If $N=30$, it takes 5.57sec/iter in DiffFlow. SBP detours $O(N)$ issue by using the experience replay memory, but still, the training time is proportional to $N$: it takes 0.52sec/iter for $N=100$, and 2.39sec/iter for $N=1,000$. Experience replay is one potential reason for the suboptimal performance of SBP. Excluding the experience replay, it takes 2.83sec/iter for $N=100$, and 23.3sec/iter for $N=1,000$ in SBP. Last, we emphasize that the training time of DDPM and INDM is invariant to the choice of $N$. We put a new section on the wall-clock training time in the revised Appendix E.7.
>
> | Model | Inference Training | Model Nonlinearity | Time Complexity | Wall-Clock Time |
> |--------------------|-------|----|----------|----------|
> | DDPM [1] | No | Linear | $O(1)$ | 0.27sec/iter |
> | DiffFlow [15] | Yes | Nonlinear | $O(N)$ | 180.88sec/iter |
> | SBP [18] | Yes | Semi-Linear | $O(N)$ | 2.39sec/iter |
> | INDM | Yes | Nonlinear | $O(1)$ | 1.69sec/iter |

---

> ### Author Response · Authors · 2022-08-02
> **Continued**
>
> **Q2. [Motivation of Normalizing Flow]** *The motivations of integrating normalizing flow into the diffusion model is not convinced. Can we integrate other component rather than normalizing flow into the diffusion model?*
> \
> \
> **A.**
> The major motivation for using normalizing flows is to guarantee the exact invertibility, and we refer to **Q1** for its theoretic and empirical importance. Also, other types of generative models, such as VAE-or-GAN, has been integrated into diffusion models, previously. Below, we enumerate the non-exhaustive list of such combined models. First, LSGM is the first to apply diffusion models to latent space where VAE maps from data to latent, back and forth. Although LSGM enjoys its flexible training, as explained in **Q1**, LSGM lacks its theoretical ground for positioning it in the category of nonlinear diffusion models, which was our primary motivation for developing INDM.
>
> One potential advantage of INDM is that it could improve the performance of LSGM by jointly combining LSGM with INDM. LSGM is modeling the prior distribution of latent space using a linear diffusion, but if we focus on the property of INDM that it is a nonlinear diffusion model, switching a *linear* diffusion in the prior modeling of LSGM with INDM (which is a *nonlinear* diffusion) could maximally expand the model expressiveness.
>
> Second, TDPM [Truncated22Zheng] integrates diffusion models with GAN. In its modeling, TDPM uses the diffusion model to deform the data variable into a *diffused* data variable, and subsequently, TDPM applies GAN for modeling this diffused data variable. In sample generation, TDPM creates diffused fake data by feed-forwarding the GAN generator, and afterward, TDPM solves the generative diffusion process to remove the remaining noise from the diffused fake data. The primary purpose of TDPM is to obtain a diffusion model with faster sample generation, and TDPM cannot be cast as one of the nonlinear diffusion models. Thus, though TDPM is another option for combining diffusion models with GAN, the contribution of TDPM and INDM is orthogonal.
> \
> \
> \
> [Truncated22Zheng] Zheng, Huangjie, et al. "Truncated diffusion probabilistic models." arXiv preprint arXiv:2202.09671 (2022).

---

> ### Author Response · Authors · 2022-08-02
> **Thank you for the integrated feedback**
>
> **Q1. [Why Invertibility Needed]** *The motivations of why invertibility is important is not clear, although the authors provide a figure and tried to illustrate this.*
> \
> \
> **A.**
> The importance of invertibility could be explained in both theoretical and empirical ways. For the theoretical side, in Figure 3 of the originally submitted manuscript, the green path that represents the "Nonlinear Forward Path" from the data variable to the noise variable is potentially not existing if the transformation between data and latent is not invertible (e.g., LSGM). This means that the Ito's lemma is applicable bidirectionally (data $\leftrightarrow$ latent) only if the mapping function is invertible. Please see **Q9** and **Q6** of Reviewer #4 (19AU) for a detailed analysis of this extent. Also, Lines 137 – 144 explain the detail of the argument on LSGM that has no exact invertibility, and Appendix C.1 gives a concrete derivation of our claims.
>
> Aside from the theory, invertibility is useful for the image-to-image translation task. We have additionally experimented with the image-to-image translation task with LSGM, and we illustrate the result in Figure 9 of the revised manuscript. In Figure 9, INDM achieves a successful image-to-image translation, while LSGM is not successful on the translation task. Such a training failure of LSGM can be explained as follows. INDM always attains a (nonlinear) forward data process starting from EMNIST, and it remains the flow training to adjust the destined variable of the forward process toward MNIST. On the other hand, LSGM has no forward data diffusion starting from EMNIST by the non-invertibility, and so a data diffusion process in LSGM is potentially limited to connect between EMNIST and MNIST. This non-existence of the forward diffusion in LSGM leads the generative process on the data space to be drifted away.
>
> Also, not only on the interpolation task, LSGM is known for its training instability on image generation tasks. Table 7 of the originally submitted Appendix presents the training failure of LSGM. LSGM is repeatedly reported [17,19] for its training instability if the training starts from scratch with a certain weighting function ($\lambda=\sigma^{2}$), whereas INDM training is stable for any training configuration. Structurally, LSGM and INDM differ only on the invertibility between data and latent, and we attribute the training stability of INDM to this invertibility.

---

### Meta-Review · Area_Chair_XXM3 · 2022-08-25

**Recommendation:** Accept
**Confidence:** Certain

**Metareview:**

The paper presents a novel non-linear diffusion model that transforms a linear diffusion in a latent space using normalizing flows. After the discussion period, three reviewers rated the paper a weak accept and one as accept. Overall, reviewers found the main idea of training flows in this latent space to be novel and interesting, and felt that the method was supported by experiments showing very good performance. Beyond that, there was a lot of discussion between reviewers and authors. Some of this focused on specific details about the training method; while quite a lot focused on other characterizations and discussions presented by the authors in the paper, which the reviewer’s found confusing or distracting (especially about optimal transport). The author response clarified several points, which prompted two reviewers to raise their scores. The authors promised to remove the discussion about optimal transport. Overall, there is support to accept the paper; the authors are encouraged to take the reviewer feedback and discussions into account to clarify the final version of the paper.


**Award:**

No

---

### Decision · Program_Chairs · 2022-09-14

Accept